# COMPETITIVE FAIR SCHEDULING WITH PREDICTIONS

**Tianming Zhao**
School of Computer Science
The University of Sydney

**Chunqiu Xia**
School of Computer Science
The University of Sydney

**Xiaomin Chang**
School of Computer Science
The University of Sydney

**Chunhao Li**
School of Computer Science
The University of Sydney

**Wei Li**
School of Computer Science
The University of Sydney

**Albert Y. Zomaya**
School of Computer Science
The University of Sydney

## ABSTRACT

Beyond the worst-case analysis of algorithms, the learning-augmented framework considers that an algorithm can leverage possibly imperfect predictions about the unknown variables to have guarantees tied to the prediction quality. We consider online non-clairvoyant scheduling to minimize the max-stretch under this framework, where the scheduler can access job size predictions. We present a family of algorithms: Relaxed-Greedy (RG) with an $O(\eta^3 \cdot \sqrt{P})$ competitive ratio, where $\eta$ denotes the prediction error for job sizes and $P$ the maximum job size ratio; Adaptive Relaxed-Greedy with an $O(\lambda^{0.5} \cdot \eta^{2.5} \cdot \sqrt{P})$ competitive ratio, where $\lambda$ denotes the error for the minimum job size; Predictive Relaxed-Greedy with an $O(\lambda^{0.5} \cdot \varphi^{0.5} \cdot \eta \cdot \max\{\eta, \varphi\} \cdot \sqrt{P})$ competitive ratio, where $\varphi$ denotes the error for the maximum job size. We also present $\mathsf{RG}^x$, an algorithm that represents a trade-off between consistency and smoothness, with an $O(\eta^{2+2x} \cdot P^{1-x})$ competitive ratio. We introduce a general method using resource augmentation to bound robustness, resulting in RR-augmented RG, with a $(1 + \epsilon)$-speed $O(\min\{\eta^3\sqrt{P}, \frac{n}{\epsilon}\})$ competitive ratio. Finally, we conduct simulations on synthetic and real-world datasets to evaluate the practical performance of these algorithms.

## 1 INTRODUCTION

In classic online scheduling, algorithms are designed to have worst-case guarantees under incomplete information. However, this conservative approach often results in suboptimal performance, especially compared to where full information is available. Given the rapid advancements in Machine Learning (ML) predictive models, the need for worst-case guarantees under incomplete information may be too pessimistic. Beyond worst-case analysis, recent researchers have explored learning-augmented algorithms to leverage predictions to improve decision-making. With ML models providing predictions of unknown variables, these algorithms are robust and have an improved performance.

Scheduling jobs that arrive over time (preemption allowed) is a fundamental challenge in computing resource management, particularly when minimizing the maximum stretch. The *stretch* of a job $J_j$ is defined as the ratio of its response time to job size, given by $\frac{C_j - r_j}{p_j^*}$, where $C_j$ denotes the completion time, $r_j$ release time, and $p_j^*$ job size (or processing time). The maximum stretch, termed *max-stretch*, measures *fairness* of the resource allocation: a max-stretch of $k$ implies that each job receives at least a $\frac{1}{k}$-speed machine. In contrast, other metrics like total completion time ($\sum C_j$) and total response time ($\sum C_j - r_j$) emphasize *efficiency* and will unfairly delay large jobs, even those released early. Studying max-stretch scheduling offers a general insight into fair resource allocation.

Previous work has considered offline max-stretch scheduling and online clairvoyant scheduling; both assume precisely known job sizes (Bender et al., 1998; 2002; Benoit et al., 2021). However, assuming perfect knowledge is overly optimistic in practice. Job sizes can be estimated using ML models (Amiri and Mohammad-Khanli, 2017) but are inherently subject to error. This raises a key question: *how can we make fair scheduling decisions with imperfect information?* Our objective is learning-augmented algorithms with performance guarantees tied to the prediction quality and robust

against poor predictions. To our knowledge, this paper is the first to explore learning-augmented algorithms for fair scheduling. We define several terms before presenting our contributions.

**Prediction errors.**   We consider several types of predictions: job size predictions per job at the job's arrival, an upfront prediction for the minimum job size, and an upfront prediction for the maximum job size. Prediction errors are defined as the multiplicative difference between predicted and actual job sizes. Let $\eta$ denote the prediction error, defined as $\eta = \max_{1 \le j \le n} \eta_j = \max_{1 \le j \le n} \max\{\frac{p_j^*}{p_j}, \frac{p_j}{p_j^*}\}$, where $p_j$ is the job size prediction for $J_j$ (Zhao et al., 2022; Azar et al., 2021). We define $P$ as the ratio of the largest and smallest job sizes: $P = \frac{p_{\max}^*}{p_{\min}^*}$, where $p_{\max}^*$ and $p_{\min}^*$ represent the maximum and minimum job sizes. For the predicted minimum job size, the error is denoted by $\lambda = \max\{\frac{p_{\min}^e}{p_{\min}^*}, \frac{p_{\min}^*}{p_{\min}^e}\}$, where $p_{\min}^e$ is the predicted minimum job size. The error for the predicted maximum job size is $\varphi = \max\{\frac{p_{\max}^e}{p_{\max}^*}, \frac{p_{\max}^*}{p_{\max}^e}\}$, where $p_{\max}^e$ is the predicted maximum job size.

**Competitive framework, consistency, smoothness, and robustness.**   An online algorithm $A$ will be compared against an offline optimal algorithm $A^*$ that knows all jobs' release times and sizes in advance. We say $A$ has a competitive ratio $c$ or is $c$-competitive, if $S^A(I) \le c \cdot S^{A^*}(I)$ for any problem instance $I$, where $S^A(I)$ and $S^{A^*}(I)$ denote the max-stretch achieved by $A$ and $A^*$, respectively. *Consistency* measures the algorithm performance under perfect predictions; *smoothness* under any prediction; *robustness* under worst-case predictions. Formally, an algorithm is said to have consistency $\tau$ (or be $\tau$-consistent) if it is $\tau$-competitive when $\eta = 1$; smoothness $\rho$ (or be $\rho$-smooth) if it is $\rho$-competitive where $\rho$ is a function of $\eta$; and robustness $\upsilon$ (or be $\upsilon$-robust) if it is $\upsilon$-competitive for any $\eta$, where $\upsilon$ is independent of $\eta$. The relationship between smoothness and robustness can be expressed as robustness $= \lim_{\eta \to \infty}$ smoothness.

**Resource augmentation.**   An algorithm $A$ is $(1 + \epsilon)$-speed $c$-competitive if $S_{1+\epsilon}^A(I) \le c \cdot S_1^{A^*}(I)$ for any problem instance $I$, where $S_{1+\epsilon}^A(I)$ and $S_1^{A^*}(I)$ denote the max-stretch achieved by $A$ and $A^*$, respectively, and the subscripts denote the speed of the processor used by the corresponding algorithm and $A^*$ is an offline optimal algorithm.

CONTRIBUTIONS

1. **Improved fairness in scheduling:** We present an exact offline algorithm with a runtime of $O(n^2 \log n)$, where $n$ is the number of jobs. We develop a family of Relaxed-Greedy (RG) algorithms under various prediction settings, achieving an $O(\sqrt{P})$ competitive ratio matching the best-known clairvoyant result under perfect predictions. These algorithms require different types of predictions, including job size and minimum/maximum job size predictions, with varying guarantees depending on the quality and availability of the predictions. These algorithms leverage ML predictions for job sizes to optimize fairness in resource allocation. They extend the algorithmic idea proposed by Bender et al. (2002), but with predictions, they no longer require prior knowledge of exact or extreme job sizes, making them applicable to a broader range of scenarios.

2. **Consistency-smoothness trade-off:** We explore the trade-off between consistency and smoothness by introducing $\mathsf{RG}^x$ with a competitive ratio of $O(\eta^{2+2x} \cdot P^{1-x})$. This algorithm is parameterized, allowing users to adjust the guarantees as a function of the prediction error $\eta$, based on the confidence in the prediction quality.

3. **Enhanced robustness via resource augmentation:** To address the robustness limitations of $O(n \cdot P)$ in the RG family, we propose $\mathsf{RG}^+$, round-robin-augmented version of RG, which achieves optimal robustness of $O(n)$ with a $(1 + \epsilon)$-speed and a competitive ratio of $O(\min\{\eta^3 \sqrt{P}, \frac{n}{\epsilon}\})$. We demonstrate how asymptotically optimal robustness can be achieved via resource augmentation when the classic technique of preferential round robin fails (Purohit et al., 2018).

4. **Extensive experimental validation:** We conduct experiments on synthetic and real-world datasets to evaluate the performance of these algorithms. The experimental results validate our theoretical analysis and demonstrate that $\mathsf{RG}^+$ consistently outperforms other algorithms in practical scenarios, showing its robustness and effectiveness.

The family of the RG algorithms is somewhat reminiscent of the $O(\sqrt{P})$-competitive algorithm proposed by Bender et al. (2002). The existing algorithms work in clairvoyant scheduling. They assume the minimum job size is 1 and a known maximum job size. The assumption is unrealistic, particularly when the extreme job sizes vary from time to time. Our algorithms extend the existing work to non-clairvoyant scheduling and overcome these limitations via predictions. The proposed algorithms allow variable minimum job size and do not require the exact maximum job size.

## 2 PRELIMINARIES

**Problem definition.** Consider a job system with a single unit-speed machine and $n$ independent jobs. The jobs arrive at the machine for processing over time. We use $J_j$ to denote the $j$-th arrived job (tie break arbitrarily). Job $J_j$ has a release time $r_j$ and a job size $p_j^*$, i.e., it takes (in total) $p_j^*$ time to complete processing job $J_j$. Jobs are preemptive, which means they can be preempted during their processing. A preempted job can resume processing later. Let $C_j$ denote the completion time for job $J_j$. The *stretch* $S_j$ for job $J_j$ is the ratio of the response time and job size, i.e., $S_j = \frac{C_j - r_j}{p_j^*}$. Our objective is to minimize the maximum stretch, known as *max-stretch*:

max-stretch $= \max_{1 \le j \le n} S_j = \max_{1 \le j \le n} \frac{C_j - r_j}{p_j^*}$.

**Related work.** We review recent advances in competitive online scheduling with predictions. For single-machine static scheduling to minimize total completion time, an $O(\min\{\frac{1}{\lambda}(1 + \frac{2\eta}{n}), \frac{2}{1-\lambda}\})$-competitive algorithm exists (Purohit et al., 2018), where $\eta$ is the additive error defined as $\eta = \sum_{j=1}^{n} |p_j - p_j^*|$, and $\lambda$ is a user-defined parameter. Follow-up works analyze robustness-consistency trade-offs (Wei and Zhang, 2020) and extend the results to dynamic scheduling and parallel machines (Bampis et al., 2022). For unrelated machine scheduling to minimize makespan, a deterministic $O(\min\{\frac{\log \eta \log m}{\log \log m}, \log m\})$-competitive algorithm is developed (Li and Xian, 2021), building on (Lattanzi et al., 2020). Here, the predictions are for machine loads, $\eta$ is the multiplicative error of the predictions, and $m$ is the number of machines. For single-machine scheduling to minimize weighted response time, an $O(\min\{\mu^3 \log(\mu P), \mu^3 \log(\mu D)\})$-competitive algorithm is developed (Azar et al., 2021), where $\mu$ is the multiplicative error for job size predictions and $D$ is the maximum density ratio. For parallel identical machine scheduling, an $O(\min\{\mu P, \mu \log \mu + \mu \log \frac{n}{m}\})$-competitive algorithm is developed to minimize the mean response time, along with an $O(\mu^2)$-competitive non-migrative algorithm for minimizing mean stretch (Azar et al., 2022). Our problem is a special case of minimizing maximum weighted response time, but prior works under the learning-augmented framework focus on min-sum objective, while our work focuses on min-max. Results on prediction error metrics (Im et al., 2021), multiple predictions (Dinitz et al., 2022), machine speed predictions (Balkanski et al., 2023), and stochastic scheduling (Merlis et al., 2023) are also available.

## 3 OFFLINE, ONLINE CLAIRVOYANT, ONLINE NON-CLAIRVOYANT SCHEDULING

We start with offline scheduling. We show an optimal algorithm with $O(n^2 \log n)$ run time. We review the lower bound and competitive algorithms for online clairvoyant scheduling. We consider online non-clairvoyant scheduling and show a $\Omega(n)$ competitive lower bound with an $O(n)$-competitive algorithm. Proofs for Section 3.1 and 3.3 are in Appendix A and B.

### 3.1 OFFLINE SCHEDULING

We have already known that an approximation exists for offline scheduling.

**Theorem 3.1** (Bender et al. (1998)). *There exists an algorithm that finds a preemptive offline schedule with max-stretch at most $1 + \epsilon$ times the optimum in time polynomial in $n$ and $\frac{1}{\epsilon}$, for any fixed $\epsilon > 0$.*

The $(1 + \epsilon)$-approximation algorithm is based on a bisection search over the value of the optimal max-stretch, using Earliest Deadline First (EDF) to check the feasibility of each candidate value. While the algorithm is polynomial in terms of the number of jobs $n$ and the precision parameter $\epsilon$, it is not purely polynomial in the input size $n$. This is because, traditionally, finding the exact minimum max-stretch was considered to involve an infinite search space, thus requiring a predefined precision

$\epsilon$ to discretize the search space. Bender et al. (2002) propose that the search for the maximum stretch can be reduced to a finite set of candidates, but they do not formalize this result. Recent work (Benoit et al., 2021) provides an exact polynomial-time greedy algorithm for the case where all jobs are released at time $0$. This algorithm follows the smallest job size first policy but does not apply to the general case with arbitrary release times. To close this gap, our first technical contribution is a formal proof of an exact, polynomial-time offline scheduling algorithm for arbitrary release times. The insight is identifying an $O(n^2)$-size list of candidates that contains the optimal max-stretch. We establish the following result, with the proof provided in Appendix A.

**Theorem 3.2** (Optimal Offline Scheduling). *There exists an algorithm that finds an offline schedule with optimal max-stretch in time polynomial in $n$. It admits an $O(n^2 \log n)$ implementation.*

## 3.2 ONLINE CLAIRVOYANT SCHEDULING

We consider online clairvoyant scheduling, where we know a job at its release time. We also know the exact job size $p_j^*$ when job $J_j$ is released. This problem has been studied by Bender et al. (1998; 2002). We summarize the key results, including the lower bound and competitive algorithms.

**Theorem 3.3** (Bender et al. (1998)). *No $\frac{P^{\frac{1}{3}}}{2}$-competitive deterministic clairvoyant algorithm exists.*

**Theorem 3.4** (Bender et al. (2002)). *There exist $O(\sqrt{P})$-competitive clairvoyant algorithms.*

## 3.3 ONLINE NON-CLAIRVOYANT SCHEDULING

We consider online non-clairvoyant scheduling, where we know a job at its release time but not the job size $p_j^*$ at job $J_j$'s release. The job size is revealed only after the job is completed. We give the lower bound on the competitive ratio for any deterministic algorithm. The key construction of the adversary is to force the algorithm to allocate no more than a $\frac{1}{n}$ share of processing at some point for the smallest-sized job, resulting in a stretch of $n$ for that job. Subsequently, the rest of the jobs are assigned geometrically increasing sizes, ensuring that the optimum max-stretch is approximately $1$.

**Theorem 3.5** (Non-clairvoyant Lower Bound). *No $c$-competitive deterministic algorithm exists with $c < n$ for online non-clairvoyant max-stretch scheduling, even if all jobs are released at time $0$.*

We show Round Robin (RR) scheduling where every job in the system shares an equal amount of processing power of the machine at any arbitrarily fine time, i.e., with $k$ jobs in the system, each gets $\frac{1}{k}$ units of the processing done every time unit until any job completes or any job arrives), is an $n$-competitive online non-clairvoyant scheduling algorithm.

**Theorem 3.6** ($n$-Competitive RR). *RR is $n$-competitive for online non-clairvoyant scheduling.*

We show that any non-lazy algorithm is $O(n \cdot P)$-competitive. Here, non-lazy means that the algorithm does not idle the machine unnecessarily when there are active jobs. This result will be used to establish the robustness of learning-augmented algorithms in the following sections.

**Theorem 3.7** (Worst-case Competitive Ratio). *Any non-lazy algorithm is $(n-1) \cdot P + 1$-competitive.*

## 4 ONLINE SCHEDULING WITH PREDICTIONS

We consider the integration of predictions. Rather than knowing the exact job size, we have a size prediction $p_j$ for job $J_j$. Initially, we assume that the extreme predictions are known beforehand; specifically, the algorithm knows the smallest prediction $p_{\min}$ (i.e., $\min_{1 \le j \le n} p_j$) and the largest prediction $p_{\max}$ ($\max_{1 \le j \le n} p_j$) at time $0$. We present the Relaxed-Greedy algorithm, which achieves an $O(\eta^3 \sqrt{P})$ competitive ratio, where $\eta$ is the prediction error for job sizes. Next, we show how to remove this assumption by updating the extreme predictions observed online and using predictions of the extreme job sizes. We introduce the Adaptive Relaxed-Greedy algorithm, which uses the prediction of the minimum job size and dynamically updates the maximum job size prediction, achieving an $O(\lambda^{0.5} \cdot \eta^{2.5} \cdot \sqrt{P})$ competitive ratio, where $\lambda$ is the prediction error for the minimum job size. Finally, we present the Predictive Relaxed-Greedy algorithm, which uses predictions of both extreme job sizes, achieving an $O(\lambda^{0.5} \cdot \varphi^{0.5} \cdot \eta \cdot \max\{\eta, \varphi\} \cdot \sqrt{P})$ competitive ratio, where $\varphi$ is the prediction error for the maximum job size. We begin by outlining the algorithmic ideas, with proofs for Section 4.2, 4.3, and 4.4 provided in Appendix C, E, and F.

## 4.1 ALGORITHM OVERVIEW

The key algorithmic idea is the reduction to the case with two job sizes. Recall that the stretch is defined as $\frac{C_j - r_j}{p_j^*}$. If there are only two job sizes — one short and the other long — the jobs of the same type should be executed in a First-In-First-Out (FIFO) manner. A short job should be executed before a long job if the long job is released at the same time as, or later than, the short job. It can be shown that greedily scheduling the jobs, i.e., always running the job with the minimal stretch, results in a constant competitive ratio when there are only two job sizes (as shown in Bender et al. (2002) and Lemma C.6). Clearly, the problem becomes easier when there are only two job sizes. Our strategy is to reduce the general problem to this two-job-size case via predictions.

We classify jobs into two categories: job $J_j$ is considered short if its job size prediction $p_j \le \frac{\sqrt{p_{\min} \cdot p_{\max}}}{\mu}$, where $\mu$ is a constant, and long otherwise. We show in Appendix H that the optimal choice for $\mu$ is $\sqrt{3}$, though for now, we can ignore this constant. We then approximate the stretch for short jobs by $\frac{C_j - r_j}{\sqrt{p_{\min} \cdot p_{\max}}}$ and for long jobs by $\frac{C_j - r_j}{p_{\max}}$; these are defined to be *relaxed stretch*. This construction follows the idea in Bender et al. (2002), but requires both $p_{\min}$ and $p_{\max}$ to account for that the minimal job size is not necessarily 1, which is assumed in Bender et al. (2002). This approach overestimates the size of a short job $J_j$ by $\sqrt{p_{\min} \cdot p_{\max}} \le \sqrt{\eta^2 \cdot p_{\min}^* \cdot p_{\max}^*} = \eta \cdot \sqrt{P} \cdot p_{\min}^* \le \eta \cdot \sqrt{P} \cdot p_j^*$, resulting in an overestimation by at most a factor of $\eta\sqrt{P}$. Similarly, it overestimates the size of a long job $J_j$ ($p_j > \sqrt{p_{\min} \cdot p_{\max}}$) by $p_{\max} \le \sqrt{p_{\min} \cdot p_{\max}} \cdot \sqrt{\frac{p_{\max}}{p_{\min}}} \le \eta \cdot \sqrt{P} \cdot \eta \cdot p_j^* = \eta^2 \cdot \sqrt{P} \cdot p_j^*$, meaning the overestimation is at most a factor of $\eta^2\sqrt{P}$. Overall, the relaxed stretch overestimates job sizes by at most $\eta^2\sqrt{P}$. With these approximations, the stretch of a job $J_j$ is bounded by $O(\eta^2\sqrt{P})$ times the relaxed stretch, resulting in a competitive ratio loss of $O(\eta^2\sqrt{P})$ (Lemma C.8). However, the advantage of this approximation is that it reduces the scheduling problem to the easier two-job-size case. In Appendix H we show that using the square root of $p_{\min} \cdot p_{\max}$ provides the optimal dependency on $P$ in the competitive ratio, though it is possible to adjust the power to $p_{\min}^x \cdot p_{\max}^{1-x}$ for any $0 < x \le \frac{1}{2}$, yielding different power dependencies on $P$ in the competitive ratio.

Using predictions to classify jobs as short or long, we greedily schedule the jobs by running the job with the minimal relaxed stretch at any time. This heuristic minimizes the relaxed stretch to within a constant factor ($\sqrt{3}$). Finally, we observe that the optimal relaxed stretch is bounded by $\eta$ times the optimal max-stretch (Lemma C.7). The max-stretch bound from Relaxed-Greedy is $O(\eta^2\sqrt{P})$ of the optimal relaxed stretch, which is itself bounded by $O(\eta^3\sqrt{P})$ of the optimal max-stretch.

## 4.2 RELAXED-GREEDY

One of the assumptions for previous algorithms is that the minimum job size is known to be 1 and that the maximum job size is also known upfront. We relax these assumptions by setting a threshold $\alpha$ as the geometric mean of the predicted minimum and maximum job sizes, which serves as the boundary for classifying jobs as small or large. Our analysis shows that such methods allow the performance to tie to the prediction errors for these quantities.

**Definition 4.1** ($\alpha$-$\beta$ Relaxed-Stretch). Define $\alpha$-$\beta$ relaxed-stretch $\widetilde{S_{j,\alpha,\beta}}(t)$ for job $J_j$ at time $t$ given $\alpha$ and $\beta$: $\widetilde{S_{j,\alpha,\beta}}(t) = \frac{t - r_j}{\alpha}$ if $J_j$ is active at time $t$ and $p_j \le \alpha$; $\widetilde{S_{j,\alpha,\beta}}(t) = \frac{t - r_j}{\beta}$ if $J_j$ is active at time $t$ and $p_j > \alpha$. We drop the prefix $\alpha$-$\beta$ when the context is clear.

**Definition 4.2** ($\alpha$-$\beta$ Short and Long Jobs). Given parameters $\alpha$ and $\beta$, a job $J_j$ is said to be short if $p_j \le \alpha$; a job $J_j$ is said to be long otherwise.

We present an $O(\eta^3\sqrt{P})$-competitive algorithm Relaxed-Greedy (RG). Through the execution of RG, parameters $\alpha$ and $\beta$ are set to be constants: $\alpha = \frac{\sqrt{p_{\min} \cdot p_{\max}}}{\sqrt{3}}$ and $\beta = p_{\max}$. It holds $\alpha = \frac{\sqrt{p_{\min} \cdot p_{\max}}}{\sqrt{3}} < p_{\max} = \beta$. Note that the constant $\sqrt{3}$ in $\alpha$ serves to yield an optimal constant in the competitive ratio. Any constant can replace it and still maintain RG's $O(\eta^3\sqrt{P})$ competitive ratio as long as $\alpha < \beta$ (see Appendix D for a discussion). Algorithm RG maintains the invariant of running the active job with the largest relaxed-stretch. This means that at any time $t$, RG looks at all

---

**Algorithm 1:** Relaxed-Greedy

**Data**  : job size prediction $p_j$ upon job release and the extreme predictions $p_{\min}$ and $p_{\max}$

**Result** : schedule with a max-stretch $O(\eta^3 \sqrt{P})$ times the optimum max-stretch

1   $\alpha \leftarrow \frac{\sqrt{p_{\min} \cdot p_{\max}}}{\sqrt{3}}, \beta \leftarrow p_{\max}$

2   **Function** `relaxedStretch`$(j, t)$ `// computes job` $J_j$`'s relaxed-stretch at time` $t$

3     **if** *$J_j$ is completed before $t$* **then**

4       $t \leftarrow C_j$ `// set variable` $t$ `to be the completion time of` $J_j$

5     **return** $\frac{t-r_j}{\alpha}$ **if** $p_j \leq \alpha$ **else** $\frac{t-r_j}{\beta}$

6   Relaxed-Greedy maintains the active job (released but not completed job) set $\mathbf{U}$; initially, set $\mathbf{U} \leftarrow \emptyset$.

7   **Event Function** `JobRelease()` `//` $J_j$ `is released`

8     insert $J_j$ into $\mathbf{U}$; run $J_j$ on the machine **if** $\mathbf{U} = \emptyset$.

9   **Event Function** `JobComplete()` `//` $J_j$ `has been processed for` $p_j^*$ `unit at time` $t$

10     remove $J_j$ from $\mathbf{U}$.

11     **if** *there is at least one job in* $\mathbf{U}$ **then**

12       let $J_s$ be the job in $\mathbf{U}$ with the largest relaxed-stretch: $J_s \leftarrow \arg\max_{J_x \in \mathbf{U}}$ relaxedStretch$(x, t)$, with tie breaking by the smaller job index.

13       run $J_s$ on the machine.

14     **else**

15       idle the machine.

16   **Event Function** `JobSwitch()` `// there exists` $J_x \in \mathbf{U}, x \neq s$ `such that` relaxedStretch$(x, t) >$ relaxedStretch$(c, t)$**,** `where` $J_c$ `is the currently running job`

17     let $J_s$ be the job in $\mathbf{U}$ with the largest relaxed-stretch: $J_s \leftarrow \arg\max_{J_x \in \mathbf{U}}$ relaxedStretch$(x, t)$, with tie breaking by the smaller job index.

18     preempt $J_c$.

19     run $J_s$ on the machine.

---

the active jobs and processes the one with the largest relaxed-stretch until the job is completed or until there is another active job later with a larger relaxed-stretch. Note that RG does not need the knowledge of the prediction error $\eta$. The pseudo-code of RG is given in Algorithm 1.

**Theorem 4.3** (The $O(\eta^3 \sqrt{P})$ Competitive Ratio). *Relaxed-Greedy is $\sqrt{3} \cdot \eta^3 \cdot \sqrt{P}$-competitive for online scheduling with predictions.*

We show that RG is $\Theta(n \cdot P)$-competitive with the worst-case predictions.

**Theorem 4.4** (Robustness of RG). *RG is $(n-1) \cdot P + 1$-competitive with arbitrarily bad predictions, and this bound is tight.*

*Remark* 4.5. Relaxed-Greedy is $\sqrt{3} \cdot \sqrt{P}$-consistent, $\sqrt{3} \cdot \eta^3 \sqrt{P}$-smooth, and $(n-1) \cdot P + 1$-robust. It represents a consistency of $O(\sqrt{P})$ matching (asymptotically) the best-known clairvoyant competitive ratio but comes with the worst robustness among non-lazy algorithms.

Algorithm RG can be implemented with constant run-time for online decision-making. Observe that any later-released short (long) job has a smaller relaxed-stretch than any early-released active short (long) job. Thus, the short (long) jobs are processed in a First-In-First-Out (FIFO) manner, allowing us to maintain two queues, one for short and one for long jobs. Determining the job with the largest relaxed-stretch requires simply comparing the relaxed-stretch of the jobs at the front of the queues. We give RG's performance for its run-time and space complexity and preemptions.

**Proposition 4.6** (Complexity of RG). *Relaxed-Greedy admits an implementation with $O(1)$ run-time to make any online decision and $O(n)$ space.*

**Proposition 4.7** (Preemptions of RG). *Relaxed-Greedy generates a schedule with at most $n - 1$ preemptions. Thus, the amortized number of preemptions per job is at most $1$.*

### 4.3 ADAPTIVE RELAXED-GREEDY

We remove the assumption of knowing the extreme predictions beforehand and extend RG to Adaptive Relaxed-Greedy (ARG). We show that the absence of extreme predictions does not

significantly alter the competitive ratio. We show that ARG achieves an $O(\lambda^{0.5} \cdot \eta^{2.5} \cdot \sqrt{P})$ competitive ratio, where $\lambda$ denotes the prediction error for the minimum job size. Instead of knowing the exact minimum job size prediction, Algorithm ARG can access a black-box ML model that predicts the minimum job size of all jobs, denoted by $p_{\min}^e$. This predicted minimum job size $p_{\min}^e$ comes with a prediction error $\lambda = \max\{\frac{p_{\min}^e}{p_{\min}^*}, \frac{p_{\min}^*}{p_{\min}^e}\}$. Meanwhile, Algorithm ARG learns $p_{\max}$ along job release.

A key technical challenge in ARG is classifying jobs when $\alpha$ is continuously updated over time. Dynamically changing the classification of jobs might work, but the proof for its competitive ratio is difficult to establish (we did not find proof). Our solution overcomes this issue by ensuring that once a job is classified at its arrival time, based on $\alpha_{r_j}$, it remains unchanged in that classification. For example, if job $J_j$ is classified as $r$-long, it remains a long job, regardless of future updates of $\alpha$. This enables the competitive analysis for ARG. One innovation of ARG is the use of heterogeneous predictions, i.e., predictions for different types of quantities, in learning-augmented algorithms. Our analysis reveals that these different predictions impact the algorithm's performance differently.

We introduce subscript $t$ to denote the value of a variable at time $t$. Thus, $p_{\max}$ at time $t$ (denoted by $p_{\max,t}$) represents the maximum job size prediction seen so far, i.e., $p_{\max,t} = \max_{r_j \leq t} p_j$. Parameters $\alpha$ and $\beta$ are functions in $p_{\min}$ and $p_{\max}$ and thus are time-varying. They are updated with updates of $p_{\max}$ in JobRelease. Formally, ARG sets $\alpha_t = \frac{\sqrt{p_{\min}^e \cdot p_{\max,t}}}{\sqrt{3}}$ and $\beta_t = \max\{p_{\max,t}, \alpha_t\}$, where $\alpha_t$ and $\beta_t$ denote the values of $\alpha$ and $\beta$ at time $t$. We record $\alpha_{r_j}$ and $\beta_{r_j}$ for every job $J_j$. These values are used to compute the *adaptive relaxed-stretch* of $J_j$, defined as follows.

**Definition 4.8** ($\alpha$-$\beta$ Adaptive Relaxed-Stretch). Define $\alpha$-$\beta$ adaptive relaxed-stretch $\widehat{S_{j,\alpha,\beta}}(t)$ for job $J_j$ at time $t$ given $\alpha$ and $\beta$: $\widehat{S_{j,\alpha,\beta}}(t) = \frac{t-r_j}{\alpha_{r_j}}$ if $J_j$ is active at time $t$ and $p_j \leq \alpha_{r_j}$; $\widehat{S_{j,\alpha,\beta}}(t) = \frac{t-r_j}{\beta_{r_j}}$ if $J_j$ is active at time $t$ and $p_j > \alpha_{r_j}$. We drop the prefix $\alpha$-$\beta$ when the context is clear.

**Definition 4.9** ($\alpha$-$\beta$ $r$-short and $r$-long Jobs). Given parameters $\alpha$ and $\beta$, a job $J_j$ is said to be $r$-short if $p_j \leq \alpha_{r_j}$; a job $J_j$ is said to be $r$-long otherwise.

The prefix $r$ for *r-short* or *r-long* means that the classification depends on the values of $\alpha$ and $\beta$ at $r_j$ for job $J_j$. It is thus possible that a $r$-short job released later has a larger job size prediction than a $r$-long job released earlier. Other than that, ARG maintains the (same) invariant of running the active job with the largest adaptive relaxed-stretch. The pseudo-code of ARG is given in Appendix E.

**Theorem 4.10** (The $O(\lambda^{0.5} \cdot \eta^{2.5} \cdot \sqrt{P})$ Competitive Ratio). *Adaptive Relaxed-Greedy is $\sqrt{3} \cdot \lambda^{0.5} \cdot \eta^{2.5} \cdot \sqrt{P}$-competitive for online scheduling with predictions.*

*Remark* 4.11. Adaptive Relaxed-Greedy is $\sqrt{3} \cdot \sqrt{P}$-consistent, $\sqrt{3} \cdot \lambda^{0.5} \cdot \eta^{2.5} \cdot \sqrt{P}$-smooth, and $(n-1) \cdot P + 1$-robust. This robustness result is tight.

The ARG's performance remains the same as RG with respect to run-time and space complexity and preemptions. Since the parameters $\alpha$ and $\beta$ are non-decreasing, it still holds that any later-released short (long) job has a smaller adaptive relaxed-stretch than any early-released active short (long) job, allowing ARG to have the same implementation of two FIFO job queues as RG does.

**Proposition 4.12** (Complexity of ARG). *Adaptive Relaxed-Greedy admits an implementation with $O(1)$ run-time to make any online decision and $O(n)$ space.*

**Proposition 4.13** (Preemptions of ARG). *Adaptive Relaxed-Greedy generates a schedule with at most $n-1$ preemptions. Thus, the amortized number of preemptions per job is at most $1$.*

The competitive result in this section does not hold if the algorithm is instead given the predicted maximum job size and has to estimate the minimum job size prediction online. There seems to be an asymmetric importance for the knowledge about extreme job sizes. Having some knowledge of the minimum job size at the start may be a requirement.

## 4.4 PREDICTIVE RELAXED-GREEDY

If the prediction error(s) for extreme job sizes are believed to be lower than $\eta$, the algorithm can use an additional prediction about the maximum job size. We extend RG to Predictive Relaxed-Greedy (PRG), which needs access to predictions of both the minimum job size, denoted by $p_{\min}^e$, and the maximum job size, denoted by $p_{\max}^e$, of all jobs, with prediction errors $\lambda = \max\{\frac{p_{\min}^e}{p_{\min}^*}, \frac{p_{\min}^*}{p_{\min}^e}\}$

and $\varphi = \max\{\frac{p_{\max}^e}{p_{\max}^*}, \frac{p_{\max}^*}{p_{\max}^e}\}$. PRG sets static $\alpha$ and $\beta$ independent of time: $\alpha = \frac{\sqrt{p_{\min}^e \cdot p_{\max}^e}}{\mu}$ and $\beta = p_{\max}^e$, where $\mu$ is a symbolic constant $\mu = \max\{\sqrt{3}, \sqrt{\frac{p_{\min}^e}{p_{\max}^e} + \epsilon}\}$ for any small fixed $\epsilon > 0$. This ensures $\alpha < \beta$. Algorithm PRG maintains the same invariant of running the active job with the largest relaxed-stretch determined by the above $\alpha$ and $\beta$.

**Theorem 4.14** (The $O(\lambda^{0.5} \cdot \varphi^{0.5} \cdot \eta \cdot \max\{\eta, \varphi\} \cdot \sqrt{P})$ Competitive Ratio). *Predictive Relaxed-Greedy is* $\max\{\frac{3}{\mu}, \mu\} \cdot \lambda^{0.5} \cdot \varphi^{0.5} \cdot \eta \cdot \max\{\eta, \varphi\} \cdot \sqrt{P}$-*competitive.*

With reasonable predictions where $p_{\min}^e \leq p_{\max}^e$ and $\varphi \leq \eta$, PRG is $\sqrt{3} \cdot \lambda^{0.5} \cdot \varphi^{0.5} \cdot \eta^2 \cdot \sqrt{P}$-competitive. With extremely bad predictions (i.e., $p_{\min}^e > 3 \cdot p_{\max}^e$), however, the competitive ratio depends on the prediction values and becomes $(\sqrt{\frac{p_{\min}^e}{p_{\max}^e}} + \epsilon) \cdot \lambda^{0.5} \cdot \varphi^{0.5} \cdot \eta \cdot \max\{\eta, \varphi\} \cdot \sqrt{P}$. This can be avoided by switching to ARG when $p_{\min}^e > 3 \cdot p_{\max}^e$. Combining of ARG and PRG guarantees the competitive ratio of $\sqrt{3} \cdot \lambda^{0.5} \cdot \varphi^{0.5} \cdot \eta \cdot \max\{\eta, \varphi\} \cdot \sqrt{P}$ or $\sqrt{3} \cdot \lambda^{0.5} \cdot \eta^{2.5} \cdot \sqrt{P}$.

*Remark* 4.15. Predictive Relaxed-Greedy is $\max\{\frac{3}{\mu}, \mu\} \cdot \sqrt{P}$-consistent, $\max\{\frac{3}{\mu}, \mu\} \cdot \lambda^{0.5} \cdot \varphi^{0.5} \cdot \eta \cdot \max\{\eta, \varphi\} \cdot \sqrt{P}$-smooth, and $(n-1) \cdot P + 1$-robust. This robustness result is tight. The algorithm admits an implementation with $O(1)$ run-time to make any online decision and $O(n)$ space and generates a schedule with at most $n-1$ preemptions.

## 5 CONSISTENCY-SMOOTHNESS TRADE-OFFS

We study the trade-off between consistency and smoothness for RG by altering the parameters ($\alpha$ and $\beta$) in the definition of relaxed-stretch. The idea directly applies to ARG and PRG, so this paper does not discuss them. We introduce a hyper-parameter $x$ ($0 \leq x \leq \frac{1}{2}$). Denote $\mathsf{RG}^x$ to be Relaxed Greedy parametrized by $x$, where $\mathsf{RG}^x$ sets $\alpha = \frac{p_{\min}^x p_{\max}^{1-x}}{\sqrt{3}}$ and $\beta = p_{\max}$ and runs the same process as RG. This constructs a family of algorithms and RG is one member $\mathsf{RG}^{\frac{1}{2}}$. We show that $\mathsf{RG}^x$ interpolates between the First Come First Serve (denoted by FIFO) algorithm and RG, where FIFO is $O(P)$-competitive and RG is $O(\eta^3 \cdot \sqrt{P})$-competitive. See Appendix G for proofs for this section and Appendix H for the construction process of $\mathsf{RG}^x$.

**Theorem 5.1** (The $O(\eta^{2+2x} \cdot P^{1-x})$ Competitive Ratio). *$\mathsf{RG}^x$ ($0 \leq x \leq \frac{1}{2}$) is $\sqrt{3} \cdot \eta^{2+2x} \cdot P^{1-x}$-competitive for online scheduling with predictions.*

If we set $x = 0$ with $\alpha = p_{\max}, \beta = p_{\max}$, we have a singular point where $\mathsf{RG}^0$ decays to FIFO, as every job would be classified as a short job. We drop the constant $\sqrt{3}$ here as this term in relaxed-stretch does not improve the performance of $\mathsf{RG}^0$. Algorithm $\mathsf{RG}^0$ with $\alpha = p_{\max}, \beta = p_{\max}$, or FIFO, achieves an $O(P)$ competitive ratio.

**Theorem 5.2** (The $O(P)$ Competitive Ratio). *$\mathsf{RG}^0$, or FIFO, is $P$-competitive.*

Setting $x = \frac{1}{2}$ gives the best consistency ($O(\sqrt{P})$) but the worst smoothness ($O(\eta^3)$). In contrast, setting $x = 0$ with $\alpha = p_{\max}, \beta = p_{\max}$ gives the worst consistency ($O(P)$) but the optimal smoothness as a function on $\eta$ ($O(1)$), for which the prediction quality does not affect the performance at all. If we exclude the singular point $\mathsf{RG}^0$, i.e., $0 < x \leq \frac{1}{2}$, increasing hyper-parameter $x$ decreases the power of $P$ from 1 to $\frac{1}{2}$ and increases that of $\eta$ from 2 to 3 (a visualization of the trade-offs is in Appendix I). Practitioners can choose the hyper-parameter $x$ based on the confidence of the prediction quality: high-quality predictions deserve a high $x$; low-quality predictions a low $x$.

## 6 BOUNDING ROBUSTNESS VIA RESOURCE AUGMENTATION

We introduce a general method for bounding robustness. The goal is to address the weak robustness ($O(n \cdot P)$) of the learning-augmented algorithms, i.e., they cannot deal with the worst predictions. We first show that the recent technique of *Preferential Round Robin (PRR)* (Purohit et al., 2018), which allocates partial processing power to multiple algorithms, fails to address this issue with the max-stretch metric. This shows the necessity of resource augmentation (Kalyanasundaram

and Pruhs, 2000). We produce a family of *RR-augmented algorithms* with asymptotically optimal robustness and near-optimal consistency. Algorithm $\mathsf{RG}^+$, one member of the family, is $(1+\epsilon)$-speed $O(\min\{\eta^3 \cdot \sqrt{P}, \frac{n}{\epsilon}\})$-competitive. Proofs are in Appendix J.

We first review the *PRR* method. Suppose we have two algorithms, $A$ and $B$, with competitive ratios $c_A$ and $c_B$. Combining the two algorithms by giving $x$ $(0 < x < 1)$ proportion of processing power to $A$ and $1 - x$ processing power to $B$ constructs a competitive ratio of $\min\{\frac{c_A}{x}, \frac{c_B}{1-x}\}$ for some metrics (e.g., total completion time) under static scheduling (all jobs are released at time 0). This method, however, requires a critical assumption that each algorithm, when given a processing power less than 1, has a competitive ratio scaled by the allocated speed. This assumption, unfortunately, fails to hold in general and, in particular, not for the max-stretch scheduling.

**Theorem 6.1** (Motivation for Resource Augmentation). *Any algorithm is at best $\frac{\epsilon}{1-\epsilon} \cdot n + 1$-competitive, if given a $(1 - \epsilon)$-speed machine, for any $0 < \epsilon < 1$.*

With a slower machine, any algorithm is $\Omega(n)$-competitive, matching the non-clairvoyant lower bound. This means a (slower) speed takes away the advantages of clairvoyance, echoing a similar phenomenon as previously observed by Kalyanasundaram and Pruhs (2000). As a result, the assumption for *PRR* does not hold. Theorem 6.1 also suggests that $\mathsf{RR}$ has an asymptotically optimal competitive ratio $(O(n))$ with a slower machine. This leads to the design of $\mathsf{RR}$-augmented algorithms with resource augmentation. The idea is to augment a base algorithm (e.g., $\mathsf{RG}$) with a higher-speed machine, using the additional speed to run $\mathsf{RR}$ concurrently at unit speed.

**Definition 6.2** (Monotonicity (Purohit et al., 2018)). A non-clairvoyant scheduling algorithm is called *monotonic* if given two instances with identical inputs and actual job sizes $(p_1^*, ..., p_n^*)$ and $(p_1^{*\prime}, ..., p_n^{*\prime})$ such that $p_j^* \leq p_j^{*\prime}$ for all $j$, the objective function value found by the algorithm for the first instance is no higher than that for the second.

**Lemma 6.3.** *All algorithms mentioned in this paper ($\mathsf{RG}$, $\mathsf{ARG}$, $\mathsf{PRG}$, $\mathsf{RG}^x$, and $\mathsf{RR}$) are monotonic.*

**Theorem 6.4** ($\mathsf{RR}$-augmented Algorithms). *Given a monotonic algorithm $A$ with competitive ratio $c_A$, one can obtain a $(1 + \epsilon)$-speed $\min\{c_A, \frac{n}{\epsilon}\}$-competitive ($\mathsf{RR}$-augmented) algorithm.*

*Remark* 6.5. $\mathsf{RR}$-augmented $\mathsf{RG}$ (denoted by $\mathsf{RG}^+$) is $(1 + \epsilon)$-speed $\min\{\sqrt{3} \cdot \eta^3 \cdot \sqrt{P}, \frac{n}{\epsilon}\}$-competitive. Similarly, we produce $\mathsf{ARG}^+$ ($(1+\epsilon)$-speed $\min\{\sqrt{3} \cdot \lambda^{0.5} \cdot \eta^{2.5} \cdot \sqrt{P}, \frac{n}{\epsilon}\}$-competitive), $\mathsf{PRG}^+$ ($(1+\epsilon)$-speed $\min\{\max\{\frac{3}{\mu}, \mu\} \cdot \lambda^{0.5} \cdot \varphi^{0.5} \cdot \eta \cdot \max\{\eta, \varphi\} \cdot \sqrt{P}, \frac{n}{\epsilon}\}$-competitive), and $\mathsf{PRG}^{x+}$ ($(1 + \epsilon)$-speed $\min\{\sqrt{3} \cdot \eta^{2+2x} \cdot P^{1-x}, \frac{n}{\epsilon}\}$-competitive). While preserving their consistency, these algorithms achieve asymptotically optimal robustness, or strictly optimal with 2-speed augmentation.

## 7 EXPERIMENTS

We evaluate Greedy-with-Rounding ($\mathsf{GWR}$) (Bender et al., 2002), $\mathsf{RR}$, $\mathsf{RG}$, $\mathsf{ARG}$, and $\mathsf{RG}^+$ on synthetic and real-world datasets ((Google, 2019), (Alibaba, 2023), and Azure (Cortez et al., 2017)) in minimizing max-stretch, variance of stretch, and mean response time. Stretch variance measures fairness by comparing jobs to each other; the mean response time measures total efficiency. This section presents the results for the max-stretch minimization. The rest of the results and discussions are in Appendix K, L, M, and N. Code is available on GitHub (Anonymous, 2024).

We measure the *performance ratio*, which is the ratio between the max-stretch obtained by an algorithm and the optimum. Algorithm 2 is used to compute the optimal max-stretch. We set $\mathsf{RG}^+$ to be $(1 + 0.3)$-speed $\mathsf{RR}$-augmented $\mathsf{RG}$, i.e., it runs $\mathsf{RG}$ at unit speed and $\mathsf{RR}$ at speed 0.3. The job sizes are generated following exponential distributions. With the maximum job size ratio $P$, we set $p_j^*$ to $\max\{1, -\log X \cdot \mathcal{A}\}$ where $X$ is a random variable with $X \sim \mathcal{U}(0, 1)$ and $\mathcal{A}$ a scaling factor to ensure $p_j^* \leq P$. Given $\eta$ and $p_j^*$, we set $p_j \leftarrow p_j^* \cdot \exp(Y)$ with $Y \sim \mathcal{U}(-\log \eta, \log \eta)$. We generate 50 independent instances for every problem set defined by $n$, $P$, $\eta$, and the range of release times.

Figures 1 to 3 are the results for synthetic datasets. Figures 1 and 2 present the performance ratio under different $n$ and $P$ with perfect predictions. $\mathsf{RG}$ and $\mathsf{ARG}$ outperform $\mathsf{GWR}$ and $\mathsf{RR}$ in most scenarios. $\mathsf{RG}^+$, with a slight 0.3 speed augmentation, consistently achieves the best performance. Figure 3 presents the results under an increasing $\eta$. While the performance ratio of $\mathsf{RG}$, $\mathsf{ARG}$, and $\mathsf{RG}^+$ increases with $\eta$, $\mathsf{RG}^+$ is robust to prediction error with a bounded performance ratio.

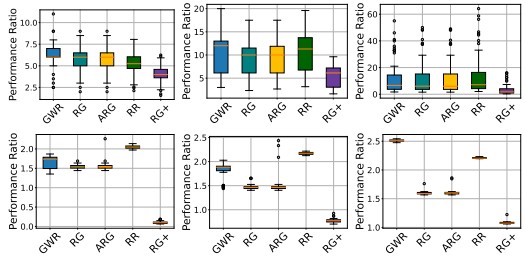

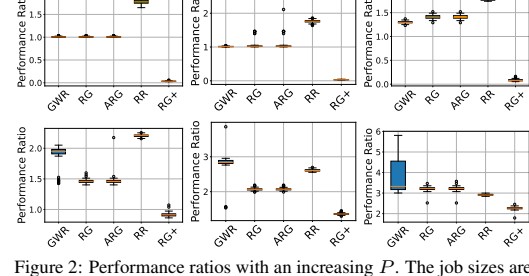

Figure 1: Performance ratios with increasing jobs. Each box represents the distribution of the performance ratios. The number of jobs is 1000 (R1C1), 2000 (R1C2), 3000 (R1C3), 4500 (R2C1), 6000 (R2C2), and 10000 (R2C3), released uniformly in $[0, 5000]$ with sizes in $[1, 10]$. Prediction error is 1 (i.e., $p_j = p_j^*$).

Figure 2: Performance ratios with an increasing $P$. The job sizes are drawn from interval $[1, P]$, where $P$ is set to 1 (R1C1), 2 (R1C2), 5 (R1C3), 10 (R2C1), 20 (R2C2), and 40 (R2C3). The number of jobs is fixed at 4500. Jobs are released in time $[0, 5000]$ uniformly at random. The prediction error is set to 1.

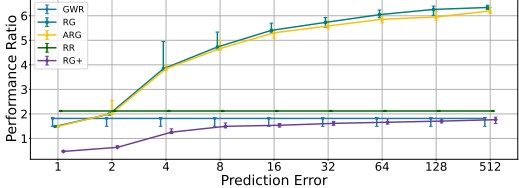

Figure 3: Performance ratios with an increasing prediction error $\eta$. The prediction error increases from 1 to 512. The number of jobs is fixed at 4500. Jobs are released in time $[0, 5000]$ uniformly at random. The job sizes are drawn from $[1, 10]$.

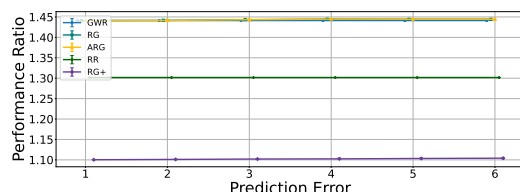

Figure 4: Performance ratios with an increasing prediction error (1 to 6) on real-world trace-log data from Google Cloud.

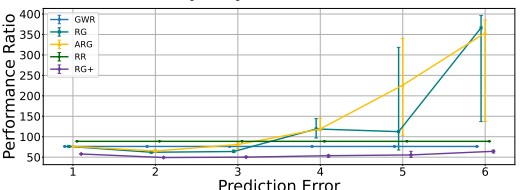

Figure 5: Performance ratios with an increasing prediction error (1 to 6) on real-world trace-log data from Azure Cloud.

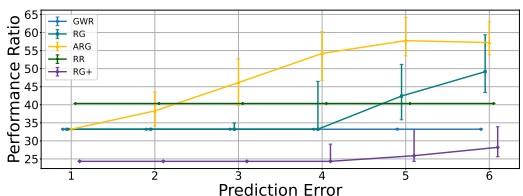

Figure 6: Performance ratios with an increasing prediction error (1 to 6) on real-world trace-log data from Alibaba Cloud.

Figure 4 to 6 presents the results on real-world trace-log data under different $\eta$. RG and ARG have strong performance under perfect predictions. With limited job size information, the performance ratio of ARG increases the fastest with increasing $\eta$. RG's performance ratio falls between ARG and GWR under imperfect predictions. Overall, RG$^+$ achieves the best performance at all times.

## 8 CONCLUSION

Under classic scheduling settings, we propose an exact polynomial-time offline algorithm, establish a non-clairvoyant competitive lower bound of $\Omega(n)$, and present a matching $O(n)$-competitive algorithm. We develop a family of algorithms for online scheduling to minimize max-stretch with different types of predictions. These algorithms offer various guarantees and leverage predictions from multiple sources. They achieve a consistency of $O(\sqrt{P})$ matching the best-known clairvoyant result but exhibit relatively weak robustness of $O(n \cdot P)$. We demonstrate how a consistency-smoothness trade-off can be achieved with a parameterized algorithm. Furthermore, we show how resource augmentation can overcome the poor robustness guarantee, resulting in asymptotically optimal robustness. Through experiments, we demonstrate the practicality of these algorithms. We anticipate that many of the techniques we employ are applicable to other problems.

Many questions remain open for future work. We have not established whether the cubic dependency on $\eta$ is optimal for learning-augmented algorithms. Therefore, analyzing the lower bound on the dependency on $\eta$ remains a subject for future study. While we have shown that our construction of RG$^x$ achieves the best possible trade-off for consistency and smoothness under this method, we have not demonstrated the Pareto-optimality of the trade-off. Establishing this remains an interesting direction for future work. Additionally, extending the approach to parallel identical machines seems to be a promising direction.

ACKNOWLEDGMENTS

We sincerely thank the anonymous reviewers for their valuable feedback and constructive suggestions, which have significantly improved this work. Professor Albert Zomaya and Dr Wei Li would like to acknowledge the support of the Australian Research Council Discovery Grant (DP200103494). Dr Wei Li acknowledges the support of the Australian Research Council (ARC) through the Discovery Early Career Researcher Award (DE210100263). Dr Tianming Zhao expresses his deepest gratitude to his family. In particular, Tianming is grateful to his wife, Min Li, and his newborn son, Michael Renqian Zhao, for their love, encouragement, and inspiration throughout this journey. Their support has been an invaluable source of motivation.

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

## TABLE OF NOTATIONS

See Table 1 for the table of notations used in the analysis.

| S | Meaning |
|---|---|
| $n$ | the number of jobs |
| $J_j$ | job with index $j$ |
| $r_j$ | release time of $J_j$ |
| $p_j^*$ | job size of $J_j$ |
| $p_j$ | job size prediction for $J_j$ |
| $C_j$ | completion time of $J_j$ |
| $t$ | a time instant |
| $C_j^{\mathsf{A}}$ | completion time of $J_j$ in the schedule constructed by algorithm $\mathsf{A}$ |
| $I$ | a problem instance (in proofs for Section 6 this $I$ refers to problem input) |
| $\sigma$ | a schedule constructed for some problem instance $I$ |
| $\eta_j$ | prediction error for $J_j$: $\eta_j = \max\{\frac{p_j^*}{p_j}, \frac{p_j}{p_j^*}\}$ |
| $\eta$ | total prediction error: $\eta = \max_{1 \le j \le n} \eta_j = \max_{1 \le j \le n} \max\{\frac{p_j^*}{p_j}, \frac{p_j}{p_j^*}\}$ |
| $p_{\min}^*, p_{\max}^*$ | minimum/maximum job size: $p_{\min}^* = \min_{1 \le j \le n} p_j^*$ and $p_{\max}^* = \max_{1 \le j \le n} p_j^*$ |
| $P$ | maximum ratio of any two job sizes: $P = \frac{p_{\max}^*}{p_{\min}^*}$ |
| $S_j$ | stretch of job $J_j$: $S_j = \frac{C_j - r_j}{p_j^*}$ |
| $S_j^{\mathsf{A}}$ | stretch of job $J_j$ in the schedule constructed by algorithm $\mathsf{A}$: $S_j^{\mathsf{A}} = \frac{C_j^{\mathsf{A}} - r_j}{p_j^*}$ |
| $S^*$ | optimal max-stretch for some problem instance $I$ |
| $S^{\mathsf{A}}$ | max-stretch achieved by algorithm $\mathsf{A}$ for some problem instance $I$ |
| $\pi$ | an ordering of jobs |
| $p_{\min}, p_{\max}$ | minimum/maximum job size prediction: $p_{\min} = \min_{1 \le j \le n} p_j$ and $p_{\max} = \max_{1 \le j \le n} p_j$ |
| $p_{\min}^e, p_{\max}^e$ | predicted minimum/maximum job size of all jobs |
| $\lambda$ | prediction error associated with $p_{\min}^e$: $\lambda = \max\{\frac{p_{\min}^e}{p_{\min}^*}, \frac{p_{\min}^*}{p_{\min}^e}\}$ |
| $\varphi$ | prediction error associated with $p_{\max}^e$: $\varphi = \max\{\frac{p_{\max}^*}{p_{\max}^e}, \frac{p_{\max}^e}{p_{\max}^*}\}$ |
| $\widetilde{S_{j,\alpha,\beta}}(t)$ | $\alpha$-$\beta$ relaxed-stretch of job $J_j$ at time $t$, abbreviated as $\widetilde{S_j}(t)$ |
| $\widetilde{S_{j,\alpha,\beta}}$ | $\alpha$-$\beta$ relaxed-stretch of job $J_j$ at its completion, abbreviated as $\widetilde{S_j}$ when $\alpha$ and $\beta$ are clear |
| $\widetilde{S_{\alpha,\beta}}$ | maximum $\alpha$-$\beta$ relaxed stretch of all jobs: $\widetilde{S_{\alpha,\beta}} = \max_{1 \le j \le n} \widetilde{S_{j,\alpha,\beta}}$, abbreviated as $\widetilde{S}$ |
| $\widetilde{S^*}$ | optimal max-relaxed-stretch for some problem instance $I$ |
| $\widetilde{S^{\mathsf{A}}}$ | max-relaxed-stretch achieved by algorithm $\mathsf{A}$ for some problem instance $I$ |
| $\widetilde{d_j}$ | deadline for job $J_j$ to achieve the optimal max-relaxed-stretch |
| $\widetilde{P_j}(t)$ | predecessors of $J_j$ (in terms of $\widetilde{d_j}$) at time $t$: $\widetilde{P_j}(t) = \{J_x | \widetilde{d_x} \le \widetilde{d_j}, J_x \text{ active at time } t\}$ |
| $\widetilde{F_j}(t)$ | followers of $J_j$ (in terms of $\widetilde{d_j}$) at time $t$: $\widetilde{F_j}(t) = \{J_x | \widetilde{d_x} > \widetilde{d_j}, J_x \text{ active at time } t\}$ |
| $\widetilde{C_j^{\mathsf{EDF}}}$ | completion time of $J_j$ in the schedule constructed by $\mathsf{EDF}$ with deadlines $\widetilde{d_j}$ |
| $\widetilde{S^{OPT}}$ | max-relaxed-stretch of the optimal schedule for the max-stretch problem |
| $\tau_j$ | the earliest time after which no job delays/$\oslash$-delays $J_j$ |
| $p_{\max,t}$ | maximum job size prediction seen by time $t$: $p_{\max,t} = \max_{r_j \le t} p_j$ |
| $\widehat{S_{j,\alpha,\beta}}(t)$ | $\alpha$-$\beta$ adaptive relaxed-stretch of job $J_j$ at time $t$, abbreviated as $\widehat{S_j}(t)$ |
| $\widehat{S_{j,\alpha,\beta}}$ | $\alpha$-$\beta$ adaptive relaxed-stretch of job $J_j$ at its completion, abbreviated as $\widehat{S_j}$ |
| $\widehat{S_{\alpha,\beta}}$ | maximum $\alpha$-$\beta$ adaptive relaxed stretch: $\widehat{S_{\alpha,\beta}} = \max_{1 \le j \le n} \widehat{S_{j,\alpha,\beta}}$, abbreviated as $\widehat{S}$ |
| $\widehat{S^*}$ | optimal max-adaptive-relaxed-stretch for some problem instance $I$ |
| $\widehat{S^{\mathsf{A}}}$ | max-adaptive-relaxed-stretch achieved by algorithm $\mathsf{A}$ for some problem instance $I$ |
| $\widehat{d_j}$ | deadline for job $J_j$ to achieve the optimal max-adaptive-relaxed-stretch |
| $\widehat{P_j}(t)$ | predecessors of $J_j$ (in terms of $\widehat{d_j}$) at time $t$: $\widehat{P_j}(t) = \{J_x | \widehat{d_x} \le \widehat{d_j}, J_x \text{ active at time } t\}$ |
| $\widehat{F_j}(t)$ | followers of $J_j$ (in terms of $\widehat{d_j}$) at time $t$: $\widehat{F_j}(t) = \{J_x | \widehat{d_x} > \widehat{d_j}, J_x \text{ active at time } t\}$ |
| $\widehat{C_j^{\mathsf{EDF}}}$ | completion time of $J_j$ in the schedule constructed by $\mathsf{EDF}$ with deadlines $\widehat{d_j}$ |
| $\widehat{S^{OPT}}$ | max-adaptive-relaxed-stretch of the optimal schedule for the max-stretch problem |

Table 1: A table of notations.

---

**Algorithm 2:** Offline Scheduling

---

**Data** : every job $J_j$ with release time $r_j$ and job size $p_j^*$

**Result** : schedule with optimal max-stretch

1 $\mathbf{S} \leftarrow \{\frac{r_i - r_j}{p_j^* - p_i^*} | 1 \leq i, j \leq n, p_i^* \neq p_j^*, \frac{r_i - r_j}{p_j^* - p_i^*} \geq 0\} \cup \{0\}$

2 view set $\mathbf{S}$ as a sorted index-based list.

3 $l, r, \sigma_o, ms_o \leftarrow 0, |\mathbf{S}|, \emptyset, \infty$

4 **while** $l < r$ **do**

5 $\quad k \leftarrow \lfloor \frac{l+r}{2} \rfloor$

6 $\quad$ assign $d_j \leftarrow r_j + \mathbf{S}_k \cdot p_j^*$ for all $1 \leq j \leq n$.

7 $\quad$ order jobs by increasing $d_j$; tie breaks by smaller $p_j^*$ first and tie breaks arbitrarily for the rest.

8 $\quad$ schedule jobs with respect to this order by always running the active arrived job with the highest priority (the most front in the order); let the resulting schedule be $\sigma$.

9 $\quad ms \leftarrow$ max-stretch of $\sigma$

10 $\quad$ **if** $ms < ms_o$ **then**

11 $\quad\quad \sigma_o, ms_o \leftarrow \sigma, ms$

12 $\quad$ **if** *every job* $J_j$ *completes no later than* $d_j$ *in* $\sigma$ **then**

13 $\quad\quad r \leftarrow k$

14 $\quad$ **else**

15 $\quad\quad l \leftarrow k + 1$

16 **return** $\sigma_o$

---

## A   MISSING PROOFS FOR SECTION 3.1

An exact minimum (optimal) max-stretch exists. This is a product of the following general result.

**Theorem A.1** (Min Max-user-defined-stretch). *Fix any positive vector* $\Gamma = (\gamma_1, \gamma_2, ..., \gamma_n)$. *There exists a schedule, for every problem instance, which achieves a minimum max-user-defined-stretch, defined as* $\widetilde{S_\Gamma} = \max_{1 \leq j \leq n} \widetilde{S_{\Gamma j}}$, *where* $\widetilde{S_{\Gamma j}} = \frac{C_j - r_j}{\gamma_j}$.

In the analysis of this section, we use the optimality of EDF: EDF minimizes the maximum lateness of the jobs and always finds a feasible schedule if one exists.

The max-stretch is a special case of the max-user-defined-stretch with $\Gamma = (p_1^*, p_2^*, ..., p_n^*)$. We show a polynomial-time exact algorithm exists. Observe that we can construct an optimal schedule using the Earliest Deadline First (EDF) algorithm if we know the optimal max-stretch. If $S^*$ is the optimal max-stretch, we can assign a deadline $d_j \leftarrow r_j + S^* \cdot p_j^*$ to every job $J_j$ and schedule jobs by EDF. This schedule has max-stretch $S^*$ by the optimality of EDF. Essentially, EDF uses the deadlines to determine a job ordering and schedules jobs according to the job priority indicated by the ordering. We can construct the optimal schedule if we know the optimal job ordering, which a bisection search can find on the following set. We let $\mathbf{S} = \{\frac{r_i - r_j}{p_j^* - p_i^*} | 1 \leq i, j \leq n, p_i^* \neq p_j^*, \frac{r_i - r_j}{p_j^* - p_i^*} \geq 0\} \cup \{0\}$ be the set of all intersections ($x$-axis) of lines $y = r_j + p_j^* \cdot x$ and a zero element. Every element $x \in \mathbf{S}$ corresponds to an order of jobs via assigning $d_j \leftarrow r_j + x \cdot p_j^*$ and ordering jobs by increasing $d_j$. One of these orders is the optimal job order. A bisection search on sorted $\mathbf{S}$ can efficiently find it. The pseudo-code of this algorithm is given in Algorithm 2.

The proof for Theorem A.1 uses the following observation.

*Observation* A.2. If a schedule exists with max-user-defined-stretch $x$, there exists a schedule with max-user-defined-stretch $y$ for any $y \geq x$. If no schedule achieves max-user-defined-stretch $x$, no schedule achieves max-user-defined-stretch $y$ for any $y \leq x$.

*Proof.* Suppose a schedule $\sigma$ achieves a max-user-defined-stretch $x$. Construct another schedule $\sigma'$, which is identical to $\sigma$ except for the job $J_k$ that is completed as the last job in $\sigma$. Preempt $J_k$ in $\sigma'$ before its completion and resume $J_k$ later so that the user-defined-stretch for job $J_k$ reaches exactly $y$. Thus, $\sigma'$ achieves a max-user-defined-stretch $y$.

Suppose no schedule achieves max-user-defined-stretch $x$. Suppose, for proof by contradiction, that there exists a schedule $\sigma$ that achieves max-user-defined-stretch $y$ with $y \leq x$. Construct a schedule

$\sigma'$ following the same procedure as above to produce $\sigma'$ with max-user-defined-stretch $x$. This contradicts the idea that no schedule achieves max-user-defined-stretch $x$. $\qquad\square$

We introduce some definitions and notations. Given an order of jobs $\pi = (J_{j_1}, J_{j_2}, ..., J_{j_n})$ indicating job priority (the more front in the order, the higher the priority is), define the *priority schedule* with respect to $\pi$ to be the schedule constructed from always running the active arrived job with the highest priority and preempting the running low priority job if necessary; the priority schedule is denoted by $PS(\pi)$.

For any $x \in \mathbf{S}$, define $\pi[x]$ to be the job order constructed by ordering jobs in increasing deadlines, where $d_j = r_j + x \cdot \gamma_j$, with tie-breaking by smaller $\gamma_j$ first and further tie-breaking by the smaller job index first (or any arbitrary order). View set $\mathbf{S}$ as a sorted index-based list $\mathbf{S} = (x_0, x_1, ..., x_\kappa)$. Note that $x_0 = 0$ by definition. We have $[x_0, x_1), [x_1, x_2), ..., [x_{\kappa-1}, x_\kappa), [x_\kappa, \infty)$ forms a partition of non-negative numbers.

Define a mapping $\Psi$ that maps positive numbers to orders of jobs. Order $\Psi(x)$ is defined as follows. Set deadline $d_j = r_j + x \cdot \gamma_j$ for all $1 \le j \le n$. Order the jobs by increasing deadlines. If two jobs have the same deadline, the one with a smaller weight ($\gamma_j$) comes first. If two jobs have the same deadline and size, i.e., the same release times and job sizes, the one with a smaller job index comes first (this tie-breaking criterion can be arbitrary). The resulting order is $\Psi(x)$.

*Proof for Theorem A.1.* We make two observations: (1) $PS(\pi[0])$ does not achieve a max-user-defined-stretch 0; (2) $PS(\pi[x])$ achieves a max-user-defined-stretch at most $x$, for a large $x$. To see observation 2, let $x' = \frac{\max_{1 \le j \le n} r_j + \sum_{j=1}^n p_j^*}{\min_{1 \le j \le n} \gamma_j}$ and $PS(\pi[x'])$ achieves a max-user-defined-stretch at most $x'$. By Observation A.2, there exists the largest index $p$ such that $PS(\pi[x_p])$ does not achieve a max-user-defined-stretch $x_p$. We show that the optimal value for max-user-defined-stretch exists in interval $[x_p, x_{p+1}]$, where $x_{p+1}$ denotes $\frac{\max_{1 \le j \le n} r_j + \sum_{j=1}^n p_j^*}{\min_{1 \le j \le n} \gamma_j}$ if $p = \kappa$. By definition of $p$, $PS(\pi[x_{p+1}])$ achieves a max-user-defined-stretch at most $x_{p+1}$. Let $X^*$ be the max-user-defined-stretch achieved by schedule $PS(\pi[x_p])$. Clearly, $X^* > x_p$.

We first show that $PS(\pi[x])$ is identical to each other for all $x \in [x_p, x_{p+1})$, by showing that $\Psi(x_p) = \Psi(x)$ for any $x \in [x_p, x_{p+1})$. Suppose, for proof by contradiction, there exists $y \in (x_p, x_{p+1})$ such that $\Psi(x_p) \ne \Psi(y)$. Let $k$ be the smallest index such that $\Psi(x_p)_k \ne \Psi(y)_k$ (the $k$-th elements of $\Psi(x_p)$ and $\Psi(S^*)$ are non-equal). For reference, we let $\Psi(x_p)_k = J_q$ and $\Psi(y)_k = J_l$. Clearly, $J_q$ is in front of $J_l$ in $\Psi(x_p)$, but $J_l$ is in front of $J_q$ in $\Psi(y)$, by the choice of $k$. This means that $J_q$ has a smaller deadline than $J_l$ in $\Psi(x_p)$, but a larger deadline in $\Psi(y)$, by the definition of $\Psi$. We have:
$$r_q + x_p \cdot \gamma_q < r_l + x_p \cdot \gamma_l \text{ and } r_q + y \cdot \gamma_q > r_l + y \cdot \gamma_l$$
This means that $\gamma_q > \gamma_l$ and $r_q < r_l$. However, we have:
$$x_p < \frac{r_l - r_q}{\gamma_q - \gamma_l} < y < x_{p+1}$$

This contradicts with $x_{p+1}$ being the smallest element (in $\mathbf{S}$) greater than $x_p$ or a new element should be in $\mathbf{S}$, by the definition of $x_{p+1}$. Thus, we must have $\Psi(x_p) = \Psi(y)$ for all $y \in (x_p, x_{p+1})$, and $PS(\pi[x])$ is identical to each other for all $x \in [x_p, x_{p+1})$.

If $X^* < x_{p+1}$, no schedule achieves a max-user-defined-stretch at most $x$ for any $x < X^*$, as $PS(\pi[x])$ achieves the same max-user-defined-stretch $X^*$ for all $x_p \le x < X^*$. In this case, $X^*$ is the minimum (optimal) max-user-defined-stretch. If $X^* \ge x_{p+1}$, no schedule achieves a max-user-defined-stretch at most $x$ for any $x < x_{p+1}$, as $PS(\pi[x])$ achieves the same max-user-defined-stretch $X^*$ for all $x \in [x_p, x_{p+1})$. In this case, $PS(\pi[x_{p+1}])$ achieves the minimum (optimal) max-user-defined-stretch $x_{p+1}$. $\qquad\square$

For the rest of this section, we restrict our discussion to the max-stretch, i.e., let $\Gamma = (p_1^*, p_2^*, ..., p_n^*)$ in the max-user-defined-stretch, and prove Theorem 3.2. We can find the optimal job order with set $\mathbf{S}$ ($\mathbf{S} = \{\frac{r_i - r_j}{p_j^* - p_i^*} | 1 \le i < j \le n, p_i^* \ne p_j^*, \frac{r_i - r_j}{p_j^* - p_i^*} \ge 0\} \cup \{0\}$). View set $\mathbf{S}$ as a sorted index-based list $\mathbf{S} = (x_0, x_1, ..., x_\kappa)$. We have $[x_0, x_1), ..., [x_{\kappa-1}, x_\kappa), [x_\kappa, \infty)$ forms a partition of non-negative

numbers. The optimal stretch $S^*$ must be within one of the intervals. Suppose $S^* \in [x_p, x_{p+1})$, where we use $x_{\kappa+1}$ to represent $\infty$ for convention. Then, $PS(\pi[x_p])$ achieves the optimal max-stretch.

**Lemma A.3.** *There exists $x \in \mathbf{S}$ such that $PS(\pi[x])$ achieves the optimal max-stretch.*

*Proof.* This is a product of Theorem A.1. $\square$

Let $x^* \in \mathbf{S}$ be the critical element with $PS(\pi[x^*])$ achieving the optimal max-stretch. Observe the following property enabling the bisection search.

**Lemma A.4.** *At lease one job $J_j$ in $PS(\pi[x])$ completes later than $r_j + x \cdot p_j^*$, for $x \in \mathbf{S}, x < x^*$. Every job $J_j$ in $PS(\pi[x])$ completes no later than $r_j + x \cdot p_j^*$, for $x \in \mathbf{S}, x > x^*$.*

*Proof.* We first show that $PS(\pi[x])$ contains at least one job $J_j$ that completes later than $r_j + x \cdot p_j^*$ for $x \in \mathbf{S}, x < x_p$. Suppose otherwise for proof by contradiction. There exists $\hat{x} \in \mathbf{S}, \hat{x} < x_p$ such that every job $J_j$ completes no later than $r_j + \hat{x} \cdot p_j^*$ in $PS(\pi[\hat{x}])$. Then, we have the max-stretch of $PS(\pi[\hat{x}])$ to be at most $\hat{x} < x_p$, which contradicts the fact that the optimal max-stretch is at least $x_p$. We then show that every job $J_j$ in $PS(\pi[x])$ completes no later than $r_j + x \cdot p_j^*$ for $x \in \mathbf{S}, x > x_p$. If $x \in \mathbf{S}, x > x_p$, then it holds that $x \geq x_{p+1} > S^*$. Since there exists a schedule with $S^*$ max-stretch, where every job $J_j$ completes no later than $r_j + S^* \cdot p_j^*$, a schedule where every job $J_j$ completes no later than $r_j + x \cdot p_j^*$ exists for $x > S^*$. By optimality of EDF, $PS(\pi[x])$ $(x > x_p)$ must ensure that every job $J_j$ completes no later than its deadline, $r_j + x \cdot p_j^*$. $\square$

**Lemma A.5.** *Algorithm 2 admits an $O(n^2 \log n)$ implementation.*

*Proof.* Computing $\mathbf{S}$ takes $O(n^2)$ time, as there are $O(n^2)$ pairs of jobs. For the same reason, $\mathbf{S}$ has $O(n^2)$ elements. Sorting $\mathbf{S}$ takes $O(n^2 \log n^2) = O(n^2 \log n)$ time. We sort all jobs by their release time before entering the bisection search, which takes $O(n \log n)$ time. The bisection search loop experiences $O(\log n^2) = O(\log n)$ iterations. Consider any iteration. Computing the job priority takes $O(n) + O(n \log n) = O(n \log n)$ time. Computing the priority schedule with the static job priority takes $O(n \log n)$. This is achieved by maintaining a priority queue of active arrived jobs and processing events, including job release and job completion. There are $O(n)$ events for job release and job completion. Every event takes $O(\log n)$ to process with the priority queue. Apart from computing the priority schedule, the rest operations in the iteration take $O(n)$ time. Therefore, an iteration of the bisection search takes $O(n \log n) + O(n \log n) + O(n) = O(n \log n)$ time. The overall complexity is:

$$O(n^2 \log n) + O(\log n) \cdot O(n \log n) = O(n^2 \log n) \qquad \square$$

**Theorem A.6** (Theorem 3.2 Restated). *Algorithm 2 finds an offline schedule with optimal max-stretch in time polynomial in $n$. It admits an $O(n^2 \log n)$ implementation.*

*Proof for Theorem 3.2.* Let the optimal stretch be $S^*$. Let $x_p \in \mathbf{S}$ such that $PS(\pi[x_p])$ achieves max-stretch $S^*$ by Lemma A.3. We show that one of the bisection search iterations visits the $x_p$ and finds the optimal job order $\pi[x_p]$. First observe that the bisection search procedure terminates within a finite number of iterations. Let $l_i, r_i$ denote the left and right ends of the searching interval at the start of the $i$-th iteration of the search (initially, $l_0 = 0$ and $r_0 = |\mathbf{S}|$). We show the invariant that, if $l_i \leq p < r_i$, either the $i$-th iteration finds $x_p$ or $l_{i+1} \leq p < r_{i+1}$. The $i$-th iteration finds $x_p$ if $\lfloor \frac{l_i + r_i}{2} \rfloor = p$. We consider the other two cases: $\lfloor \frac{l_i + r_i}{2} \rfloor < p$ and $\lfloor \frac{l_i + r_i}{2} \rfloor > p$. Suppose $k = \lfloor \frac{l_i + r_i}{2} \rfloor < p$. With $k < p$, at least one job $J_j$ in $PS(\pi[x_k])$ completes later than $r_j + x_k \cdot p_j^*$ in this iteration by Lemma A.4. Then, we have $l_{i+1} = k + 1 \leq p$ (by $k, p$ both integers) and $p < r_i = r_{i+1}$. Suppose $k = \lfloor \frac{l_i + r_i}{2} \rfloor > p$. With $k > p$, every job $J_j$ completes no later than $r_j + x_k \cdot p_j^*$ in this iteration by Lemma A.4. Then, we have $l_{i+1} = l_i \leq p$ and $p < k = r_{i+1}$. Observe that we have $l_e \geq r_e$, where $e$ is the total number of iterations, at the termination of the search. This means that one of the iterations must visit the $x_p$ and finds the optimal job order $\pi[x_p]$, which shows the correctness of the algorithm. Further, Lemma A.5 shows that the algorithm admits a polynomial-time $O(n^2 \log n)$ implementation. $\square$

## B  MISSING PROOFS FOR SECTION 3.3

**Theorem B.1** (Theorem 3.5 Restated). *No c-competitive deterministic algorithm exists with $c < n$ for online non-clairvoyant max-stretch scheduling, even if all jobs are released at time $0$.*

*Proof for Theorem 3.5.* We use proof by contradiction. Suppose there exists a $c$-competitive deterministic algorithm $A$ with $c < n$. Consider $n$ identical jobs all with job size 1 to be released at time 0. Run $A$ against this set of jobs. Without loss of generality, let $J_1$ be the job with the least amount of the processed size $l_1$. We have $l_1 \leq \frac{1}{n}$ by the pigeonhole principle.

Construct the problem instance consisting $n$ ($n \geq 2$) jobs to be released at time 0, with job sizes $p_1^* = \frac{1}{n} + \epsilon$, $p_2^* = \psi - (\frac{1}{n} + \epsilon)$, $p_3^* = \psi^2 - \psi$, $p_4^* = \psi^3 - \psi^2$, ..., $p_n^* = \psi^{n-1} - \psi^{n-2}$. Here, $p_j^* = \psi^{j-1} - \psi^{j-2}$ for $j \geq 3$, and $\psi$ is a large constant $\psi = \max\{2, \frac{n}{n-c}\}$ and $\epsilon$ is a small constant $\epsilon = \min\{\frac{1}{n}, \frac{1}{c} - \frac{1}{c \cdot \psi} - \frac{1}{n}\}$. Observe that $\frac{1}{c} - \frac{1}{c \cdot \psi} - \frac{1}{n} > 0$ and thus $\epsilon > 0$ by the definition of $\psi$. Run $A$ against this job set. Since the algorithm is deterministic, job $J_1$ will be completed after time 1. The max-stretch achieved by $A$ on this instance is more than $\frac{1}{p_1^*} = 1/(\frac{1}{n} + \epsilon) = \frac{n}{1+n\cdot\epsilon}$. Consider the schedule that runs $J_1, J_2, ..., J_n$ sequentially in the order of job index. The stretch for job $J_1$ is $S_1 = 1$. The stretch for job $J_2$ is $S_2 = \psi/(\psi - \frac{1}{n} - \epsilon)$. Observe that job $J_j$ completes at time $\psi^{j-1}$ for $j \geq 3$. The stretch for job $J_j$ is:

$$\frac{\psi^{j-1}}{\psi^{j-1} - \psi^{j-2}} = \frac{\psi}{\psi - 1}$$

for all $j \geq 3$. Observe that $\frac{\psi}{\psi-1} \geq 1$ and $\frac{\psi}{\psi-1} \geq \psi/(\psi - \frac{1}{n} - \epsilon)$ by the definition of $\epsilon$. Thus, the max-stretch achieved by this schedule is $\frac{\psi}{\psi-1}$. Therefore, the minimum max-stretch for this job set is at most $\frac{\psi}{\psi-1}$. However, we have the ratio between the max-stretch achieved by $A$ and the optimal schedule more than:

$$\frac{\frac{n}{1+n\cdot\epsilon}}{\frac{\psi}{\psi-1}} = n \cdot \frac{1 - \frac{1}{\psi}}{1 + n \cdot \epsilon} \geq n \cdot \frac{1 - \frac{1}{\psi}}{1 + n \cdot (\frac{1}{c} - \frac{1}{c \cdot \psi} - \frac{1}{n})} = c$$

which contradicts the competitive ratio of $A$. $\qquad\square$

**Theorem B.2** (Theorem 3.6 Restated). *RR is $n$-competitive for online non-clairvoyant scheduling.*

*Proof for Theorem 3.6.* With RR, job $J_j$ is processed by at least $p_j^*$ amount during time $r_j$ (when the job arrives) and $r_j + n \cdot p_j^*$. Therefore, $C_j \leq r_j + n \cdot p_j^*$ and $S_j \leq n$ for every $1 \leq j \leq n$. We have max-stretch $= \max_{1 \leq j \leq n} S_j \leq n = n \cdot 1$. Observe that the optimal schedule has a max-stretch of at least 1. Therefore, RR is $n$-competitive. $\qquad\square$

We observe instantaneous fairness achieves global fairness under the non-clairvoyant setting. This processor-sharing policy is necessary for non-clairvoyant scheduling, i.e., any algorithm must adapt a certain degree of processor sharing to be competitive. We define *processor-sharing* as follows.

**Definition B.3** ($k$ processor-sharing). An algorithm is said to be $k$ processor-sharing for a problem instance $I$, if it holds either (1) $J_j$ is completed before $t$, or (2) $J_j$ has been processed for at least $\frac{t-r_j}{k}$ unit, for any job $J_j$ and any time instant $t \in [r_j, \infty)$, in the execution of $A$ on $I$.

Intuitively, a $k$ processor-sharing algorithm allocates at least $\frac{1}{k}$ unit of processing power to all active jobs. By definition, RR is $n$ processor-sharing for every problem instance. We show the criticality of processor-sharing in obtaining competitiveness in online non-clairvoyant scheduling.

**Theorem B.4** (Equivalence of Competitiveness and Processor-sharing). *In online non-clairvoyant scheduling, a deterministic algorithm is $c$-competitive if it is $c \cdot S^*$ processor-sharing for any problem instance $I$ with the minimum max-stretch $S^*$. Any $c$-competitive algorithm must be $c \cdot n$ processor-sharing for every problem instance $I$.*

*Remark* B.5. Any $k_1$ processor-sharing algorithm (for every problem instance) is $k_2$ processor-sharing for any $k_1 \leq k_2$. As a corollary of Theorem B.4, any $k$ processor-sharing algorithm (for every problem instance) is $k$-competitive.

The proof for Theorem B.4 uses the following bound for the minimum max-stretch.

*Observation* B.6. The minimum max-stretch is at most $n$ for any problem instance with $n$ jobs.

*Proof.* The max-stretch obtained by RR on any problem instance is at most $n$. Therefore, the minimum max-stretch for problem instance $I$ is at most $n$. □

*Proof for Theorem B.4.* Fix a problem instance. If an algorithm is $c \cdot S^*$ processor-sharing for that instance, every job $J_j$ completes before $r_j + c \cdot S^* \cdot p_j^*$ with $S_j \leq c = c \cdot S^*$. The max-stretch is at most $c$ times the optimum, so the algorithm is $c$-competitive. Below we consider the reverse.

Let $A$ be any $c$-competitive algorithm. Suppose, for proof by contradiction, that there exists a problem instance $I$ with minimum max-stretch $S^*$, such that there exists a job $J_j$ and a time instant $t' \in [r_j, \infty)$, where $J_j$ has been processed for less than $q = \frac{t'-r_j}{c \cdot n}$ at time $t'$. Construct another instance $I'$, which contains the same job set as $I$ except for the job $J_j$ with $p_j^* = q$. Run $A$ on $I'$. The schedule is the same as the one generated for $I$ before and at time $t'$. Therefore, $J_j$ is completed after $t'$ for $I'$. The max-stretch achieved on $I'$ is more than $\frac{t'-r_j}{q} = c \cdot n$. The minimum max-stretch for $I'$ is at most $n$ by Observation B.6. Thus, algorithm $A$ achieves a max-stretch on $I'$ more than $c$ times the optimum, which contradicts the $c$ competitive ratio. □

**Theorem B.7** (Theorem 3.7 Restated). *Any non-lazy algorithm is $(n-1) \cdot P + 1$-competitive.*

*Proof for Theorem 3.7.* Pick any job $J_j$. The algorithm is processing active jobs in time $[r_j, C_j]$. We have $S_j = \frac{C_j - r_j}{p_j^*} \leq \frac{(n-1) \cdot p_{\max}^* + p_{\min}^*}{p_{\min}^*} = (n-1) \cdot P + 1$ and max-stretch $= \max_{1 \leq j \leq n} S_j \leq (n-1) \cdot P + 1$. □

## C    MISSING PROOFS FOR SECTION 4.2

Define a heuristic *relaxed-stretch* based on two parameters $\alpha$ and $\beta$ ($\alpha < \beta$).

**Definition C.1** ($\alpha$-$\beta$ Relaxed-Stretch). Define $\alpha$-$\beta$ relaxed-stretch $\widetilde{S_{j,\alpha,\beta}}(t)$ for job $J_j$ at time $t$ given $\alpha$ and $\beta$: $\widetilde{S_{j,\alpha,\beta}}(t) = \frac{t-r_j}{\alpha}$ if $J_j$ is active at time $t$ and $p_j \leq \alpha$; $\widetilde{S_{j,\alpha,\beta}}(t) = \frac{t-r_j}{\beta}$ if $J_j$ is active at time $t$ and $p_j > \alpha$. We use $\widetilde{S_{j,\alpha,\beta}}$ to denote the $\alpha$-$\beta$ relaxed-stretch at the completion of job $J_j$: $\widetilde{S_{j,\alpha,\beta}} = \frac{C_j - r_j}{\alpha}$ if $p_j \leq \alpha$; $\widetilde{S_{j,\alpha,\beta}} = \frac{C_j - r_j}{\beta}$ if $p_j > \alpha$. Fixing a problem instance of online scheduling with predictions and a schedule, we use *max-relaxed-stretch*, denoted by $\widetilde{S_{\alpha,\beta}}$, to represent the maximum $\alpha$-$\beta$ relaxed stretch of all jobs: $\widetilde{S_{\alpha,\beta}} = \max_{1 \leq j \leq n} \widetilde{S_{j,\alpha,\beta}}$. We drop the subscripts $\alpha$ and $\beta$ from the notations when the context is clear for simplicity. We also drop the prefix $\alpha$-$\beta$ when the context is clear.

Classify jobs into *short* and *long* jobs according to their job size predictions, with a given $\alpha$ and $\beta$.

**Definition C.2** ($\alpha$-$\beta$ Short and Long Jobs). Given parameters $\alpha$ and $\beta$, a job $J_j$ is said to be short if $p_j \leq \alpha$; a job $J_j$ is said to be long otherwise.

Fix a problem instance $I$. Run RG. Let $S^{\mathsf{RG}}$ and $S^*$ denote the max-stretch achieved by RG and the optimal max-stretch of $I$. Let $\widetilde{S^{\mathsf{RG}}}$ and $\widetilde{S^*}$ denote the max-relaxed-stretch achieved by RG and the optimal max-relaxed-stretch of $I$. The quantity $\widetilde{S^*}$ exists due to Theorem A.1, regardless of the values of $\alpha$ and $\beta$. We study the relations between $S^{\mathsf{RG}}$, $\widetilde{S^{\mathsf{RG}}}$, $\widetilde{S^*}$, and $S^*$. We begin with showing $\widetilde{S^{\mathsf{RG}}} \leq 3 \cdot \widetilde{S^*}$. We consider how much RG differs from an optimal execution by examining the maximum amount of processing units a job can be delayed compared with the optimal schedule. Define $\widetilde{d_j}$ to be the deadline of completing job $J_j$ to achieve the max-relaxed-stretch, i.e., $\widetilde{d_j} = r_j + \widetilde{S^*} \cdot \alpha$ for short job $J_j$ and $\widetilde{d_j} = r_j + \widetilde{S^*} \cdot \beta$ for long job $J_j$. A schedule achieves the optimal max-relaxed-stretch if every job $J_j$ completes no later than $\widetilde{d_j}$. We study how much any job $J_j$ can be delayed after $\widetilde{d_j}$ in RG.

Observe that EDF finds a feasible schedule meeting all deadlines if one exists. The schedule obeying EDF with deadlines $\widetilde{d}_j$ achieves the optimal max-relaxed-stretch $\widetilde{S^*}$. Define set $\widetilde{P}_j(t) = \{J_x | \widetilde{d}_x \leq \widetilde{d}_j, J_x \text{ active at time } t\}$ and set $\widetilde{F}_j(t) = \{J_x | \widetilde{d}_x > \widetilde{d}_j, J_x \text{ active at time } t\}$ to be the predecessors and followers (in terms of the deadlines) of $J_j$ at time $t$.

**Definition C.3** (Definition of Delay). Job $J_j$ is said to be *delayed* by another job $J_k$ at time $t$, if the following four conditions are simultaneously met: (1) $J_j$ is not completed or not released at time $t$, (2) $J_k \in \widetilde{F}_j(t)$, (3) $\widetilde{P}_j(t) \neq \emptyset$, and (4) RG is processing $J_k$ at time $t$ with $\max_{J_x \in \widetilde{P}_j(t)} \widetilde{S}_x(t) < \widetilde{S}_k(t)$.

If $\widetilde{C_j^{\mathsf{EDF}}}$ and $C_j^{\mathsf{RG}}$ denote the completion time for $J_j$ in the (optimal) schedule constructed by EDF with $\widetilde{S^*}$ and the schedule by RG, it follows $C_j^{\mathsf{RG}} \leq \widetilde{C_j^{\mathsf{EDF}}} + \int_{\text{any job delays } J_j \text{ at time } t} dt$. The following two lemmas hold with any given $\alpha$ and $\beta$ $(\alpha < \beta)$.

**Lemma C.4** (No Job Delays a Long Job). *A long job is never delayed, for any $\alpha$ and $\beta$ $(\alpha < \beta)$.*

*Proof.* Fix any long job $J_j$ and a time $t$ before the completion of $J_j$. Suppose, for proof by contradiction, that another job $J_k \in \widetilde{F}_j(t)$ delays $J_j$ at time $t$. Job $J_k$ must be a short job, since otherwise, we would have $r_j < r_k$ and $\widetilde{S}_j(t) > \widetilde{S}_k(t)$. We also have $\widetilde{d}_k > \widetilde{d}_j \Rightarrow r_k + \widetilde{S^*} \cdot \alpha > r_j + \widetilde{S^*} \cdot \beta \Rightarrow r_k > r_j$. Thus, $J_j$ is active at time $t$. We have:

$$\widetilde{S}_j(t) < \widetilde{S}_k(t) \Rightarrow \frac{t - r_j}{\beta} < \frac{t - r_k}{\alpha} \Rightarrow t > \frac{r_k \cdot \beta - r_j \cdot \alpha}{\beta - \alpha}$$

Observe, by the definition of delay, that $J_k \in \widetilde{F}_j(t)$ implies $r_k + \widetilde{S^*} \cdot \alpha > r_j + \widetilde{S^*} \cdot \beta$. It holds that:

$$t > \frac{(r_j + \widetilde{S^*} \cdot \beta - \widetilde{S^*} \cdot \alpha) \cdot \beta - r_j \cdot \alpha}{\beta - \alpha} = r_j + \widetilde{S^*} \cdot \beta = \widetilde{d}_j$$

Thus, no job delays $J_j$ before time $\widetilde{d}_j$. In such a case, however, $J_j$ completes no later than $\widetilde{d}_j$, which is earlier than $t$, contradicting the definition of delay. $\square$

**Lemma C.5** (Bound on the Delay for a Short Job). *A short job $J_j$ completes no later than $r_j + 3 \cdot \widetilde{S^*} \cdot \alpha$, for any $\alpha$ and $\beta$ $(\alpha < \beta)$.*

*Proof.* We claim that any short job $J_j$ can only be delayed before time $\widetilde{d}_j$, i.e., some job $J_k \in \widetilde{F}_j(t)$ delays $J_j$ at some time $t$ implies $t < \widetilde{d}_j$. We show this via proof by contradiction. Fix a short job $J_j$. Suppose another job $J_k \in \widetilde{F}_j(t)$ delays $J_j$ at some time $t \geq \widetilde{d}_j$. Job $J_j$ must be active at time $t$. Job $J_k$ must be a long job, since otherwise, we would have $J_k \in \widetilde{F}_j(t)$ implying $\widetilde{d}_k > \widetilde{d}_j$, which means $r_k > r_j$ and $\widetilde{S}_j(t) > \widetilde{S}_k(t)$. We have:

$$\widetilde{S}_j(t) < \widetilde{S}_k(t) \Rightarrow \frac{t - r_j}{\alpha} < \frac{t - r_k}{\beta} \Rightarrow t < \frac{r_j \cdot \beta - r_k \cdot \alpha}{\beta - \alpha}$$

With $J_k \in \widetilde{F}_j(t)$, it holds $r_k + \widetilde{S^*} \cdot \beta > r_j + \widetilde{S^*} \cdot \alpha$. We have:

$$t < \frac{r_j \cdot \beta - (r_j + \widetilde{S^*} \cdot \alpha - \widetilde{S^*} \cdot \beta) \cdot \alpha}{\beta - \alpha} = r_j + \widetilde{S^*} \cdot \alpha = \widetilde{d}_j$$

which contradicts $J_k$ delays job $J_j$ at time $t \geq \widetilde{d}_j$.

Next, consider any short job $J_j$. Define $\tau_j$ to be the earliest time after which no job delays $J_j$, i.e., for any time $t$ after $\tau_j$ and before the completion time of $J_j$, the processor always processes a job from $\widetilde{P}_j(t)$ when $\widetilde{P}_j(t) \neq \emptyset$, and $\tau_j$ is the earliest time instant such that the condition holds ($\tau_j$ is the last time $J_j$ is delayed). Consider set $\widetilde{P}_j(\tau_j)$. Observe that every job $J_x \in \widetilde{P}_j(\tau_j)$ is delayed by the job (call it $J_q$) running at time instant $\tau_j$, since $\widetilde{d}_x \leq \widetilde{d}_j$ and $\widetilde{d}_j < \widetilde{d}_q$. It follows $\widetilde{P}_j(\tau_j)$ contains only short jobs, by Lemma C.4. Observe that, for any job $J_x \in \widetilde{P}_j(\tau_j)$, we have $\tau_j \leq \widetilde{d}_x$, or

equivalently $r_x \geq \tau_j - \widetilde{S^*} \cdot \alpha$, as job $J_x$ is delayed at $\tau_j$. We bound the total amount of delay on $J_j$ by $\sum_{J_x \in \widetilde{P_j}(\tau_j)} p_x^*$. Consider the optimal schedule constructed by EDF with $\widetilde{S^*}$. We have two cases.

**Case 1** ($\tau_j < r_j$)**.** In the optimal schedule, all jobs in $\widetilde{P_j}(\tau_j)$ are released no earlier than $\tau_j - \widetilde{S^*} \cdot \alpha$ and no later than $\tau_j$ and completed no later than $\tau_j + \widetilde{S^*} \cdot \alpha$. We have:

$$\sum_{J_x \in \widetilde{P_j}(\tau_j)} p_x^* \leq \tau_j + \widetilde{S^*} \cdot \alpha - (\tau_j - \widetilde{S^*} \cdot \alpha) = 2 \cdot \widetilde{S^*} \cdot \alpha$$

**Case 2** ($\tau_j \geq r_j$)**.** In the optimal schedule, all jobs in $\widetilde{P_j}(\tau_j)$ (including job $J_j$) are released no earlier than $\tau_j - \widetilde{S^*} \cdot \alpha$ and completed no later than $\widetilde{d}_j = r_j + \widetilde{S^*} \cdot \alpha$. We have:

$$\sum_{J_x \in \widetilde{P_j}(\tau_j)} p_x^* \leq \widetilde{d}_j - (\tau_j - \widetilde{S^*} \cdot \alpha) = r_j + 2 \cdot \widetilde{S^*} \cdot \alpha - \tau_j \leq 2 \cdot \widetilde{S^*} \cdot \alpha$$

Thus, the completion time for a short job $J_j$ in RG is at most $\widetilde{d}_j + \sum_{J_x \in \widetilde{P_j}(\tau_j)} p_x^* \leq r_j + 3 \cdot \widetilde{S^*} \cdot \alpha$  □

Let $\widetilde{S_j^*}$ and $\widetilde{S_j^{RG}}$ be the relaxed-stretch of $J_j$ in the schedule constructed by EDF with $S^*$ and RG.

**Lemma C.6.** *For any given $\alpha$ and $\beta$ ($\alpha < \beta$), it holds $\widetilde{S_j^{RG}} \leq \widetilde{S^*}$ for any long job $J_j$ and $\widetilde{S_j^{RG}} \leq 3 \cdot \widetilde{S^*}$ for any short job $J_j$.*

*Proof.* Pick any job $J_j$. If $J_j$ is a long job, it follows by Lemma C.4 that:

$$C_j^{RG} \leq \widetilde{C_j^{EDF}} + \int_{\text{any job delays } J_j \text{ at time } t} dt = \widetilde{C_j^{EDF}} \Rightarrow \widetilde{S_j^{RG}} \leq \widetilde{S_j^*} \leq \widetilde{S^*}$$

If $J_j$ is a short job, it follows by Lemma C.5 that:

$$\widetilde{S_j^{RG}} = \frac{C_j^{RG} - r_j}{\alpha} \leq \frac{(r_j + 3 \cdot \widetilde{S^*} \cdot \alpha) - r_j}{\alpha} = 3 \cdot \widetilde{S^*}$$  □

Denote $S_j^{RG}$ and $\widetilde{S_j^{RG}}$ to be the stretch and the relaxed-stretch of job $J_j$ in the execution of RG. The following lemmas show the relation between $\widetilde{S^*}$ with $S^*$ and $S_j^{RG}$ with $\widetilde{S_j^{RG}}$ by using $\alpha = \frac{\sqrt{p_{\min} \cdot p_{\max}}}{\sqrt{3}}$ and $\beta = p_{\max}$. Lemma C.7 requires $\alpha < \beta$ and $p_j \leq \beta$ for all $1 \leq j \leq n$ to hold; Lemma C.8 requires the specific values for $\alpha$ and $\beta$ to hold. They do not hold under any arbitrary $\alpha$ and $\beta$.

**Lemma C.7.** $\widetilde{S^*} \leq \eta \cdot S^*$, *for any given $\alpha < \beta$ and $p_j \leq \beta$ for all $1 \leq j \leq n$.*

*Proof.* Denote $\widetilde{S^{OPT}}$ to be the max-relaxed-stretch of the optimal schedule for the max-stretch problem (the schedule with max-stretch $S^*$). It follows $\widetilde{S^*} \leq \widetilde{S^{OPT}}$. Consider the optimal schedule with max-stretch $S^*$. Let $C_j^*$ and $\widetilde{S_j^{OPT}}$ denote the completion time of job $J_j$ and the relaxed-stretch of $J_j$ in this schedule. Pick any job $J_j$. It follows $p_j \leq \alpha$ and $\widetilde{S_j^{OPT}} = \frac{C_j^* - r_j}{\alpha} \leq \frac{C_j^* - r_j}{p_j} \leq \eta \cdot \frac{C_j^* - r_j}{p_j^*} \leq \eta \cdot S^*$, if $J_j$ is a short job. It follows $p_j \leq \beta$ and $\widetilde{S_j^{OPT}} = \frac{C_j^* - r_j}{\beta} \leq \frac{C_j^* - r_j}{p_j} \leq \eta \cdot \frac{C_j^* - r_j}{p_j^*} \leq \eta \cdot S^*$, if $J_j$ is a long job. Combining these two cases gives us $\widetilde{S^*} \leq \widetilde{S^{OPT}} \leq \eta \cdot S^*$.  □

**Lemma C.8.** *It holds $S_j^{RG} \leq \sqrt{3} \cdot \eta^2 \cdot \sqrt{P} \cdot \widetilde{S_j^{RG}}$ for any long job $J_j$ and $S_j^{RG} \leq \frac{\eta}{\sqrt{3}} \cdot \sqrt{P} \cdot \widetilde{S_j^{RG}}$ for any short job $J_j$, with $\alpha = \frac{\sqrt{p_{\min} \cdot p_{\max}}}{\sqrt{3}}$ and $\beta = p_{\max}$.*

*Proof.* Pick any job $J_j$. If job $J_j$ is a long job, it holds that $p_j > \alpha = \frac{\sqrt{p_{\min} \cdot p_{\max}}}{\sqrt{3}}$. We have:

$$S_j^{\mathsf{RG}} = \frac{C_j^{\mathsf{RG}} - r_j}{p_j^*} = \frac{C_j^{\mathsf{RG}} - r_j}{\beta} \cdot \frac{\beta}{p_j^*} \leq \widetilde{S_j^{\mathsf{RG}}} \cdot \eta \cdot \frac{\beta}{p_j} < \widetilde{S_j^{\mathsf{RG}}} \cdot \eta \cdot \sqrt{3} \cdot \frac{p_{\max}}{\sqrt{p_{\min} \cdot p_{\max}}}$$

$$\leq \widetilde{S_j^{\mathsf{RG}}} \cdot \eta \cdot \sqrt{3} \cdot \sqrt{\frac{p_{\max}}{p_{\min}}} \leq \widetilde{S_j^{\mathsf{RG}}} \cdot \eta \cdot \sqrt{3} \cdot \sqrt{\frac{\eta \cdot p_{\max}^*}{p_{\min}^*/\eta}} = \sqrt{3} \cdot \eta^2 \cdot \sqrt{P} \cdot \widetilde{S_j^{\mathsf{RG}}}$$

If job $J_j$ is a short job, we have:

$$S_j^{\mathsf{RG}} = \frac{C_j^{\mathsf{RG}} - r_j}{p_j^*} \leq \frac{C_j^{\mathsf{RG}} - r_j}{p_{\min}^*} = \frac{C_j^{\mathsf{RG}} - r_j}{\alpha} \cdot \frac{\alpha}{p_{\min}^*} \leq \widetilde{S_j^{\mathsf{RG}}} \cdot \frac{\sqrt{\eta \cdot p_{\min}^* \cdot \eta \cdot p_{\max}^*}}{\sqrt{3} \cdot p_{\min}^*} \leq \frac{\eta}{\sqrt{3}} \cdot \sqrt{P} \cdot \widetilde{S_j^{\mathsf{RG}}}$$

The derivation uses the observation: $p_{\max} \leq \eta \cdot p_{\max}^*$, $p_{\min} \geq \frac{p_{\min}^*}{\eta}$, and $p_{\min} \leq \eta \cdot p_{\min}^*$. $\quad\square$

Combining the above results completes the proof for Theorem 4.3.

**Theorem C.9** (Theorem 4.3 Restated). *Relaxed-Greedy is $\sqrt{3} \cdot \eta^3 \cdot \sqrt{P}$-competitive for online scheduling with predictions.*

*Proof for Theorem 4.3.* Consider the execution of RG. Pick any job $J_j$. If job $J_j$ is a long job, we have, by Lemmas C.8, C.6, and C.7, that:

$$S_j^{\mathsf{RG}} \leq \sqrt{3} \cdot \eta^2 \cdot \sqrt{P} \cdot \widetilde{S_j^{\mathsf{RG}}} \leq \sqrt{3} \cdot \eta^2 \cdot \sqrt{P} \cdot \widetilde{S^*} \leq \sqrt{3} \cdot \eta^3 \cdot \sqrt{P} \cdot S^*$$

If job $J_j$ is a short job, we have, by Lemmas C.8, C.6, and C.7, that:

$$S_j^{\mathsf{RG}} \leq \frac{\eta}{\sqrt{3}} \cdot \sqrt{P} \cdot \widetilde{S_j^{\mathsf{RG}}} \leq \frac{\eta}{\sqrt{3}} \cdot \sqrt{P} \cdot 3 \cdot \widetilde{S^*} \leq \sqrt{3} \cdot \eta^2 \cdot \sqrt{P} \cdot S^*$$

Therefore, we have $S^{\mathsf{RG}} = \max_{1 \leq j \leq n} S_j^{\mathsf{RG}} \leq \sqrt{3} \cdot \eta^3 \cdot \sqrt{P} \cdot S^*$. RG is $\sqrt{3} \cdot \eta^3 \cdot \sqrt{P}$-competitive. $\quad\square$

We show that RG is $\Theta(n \cdot P)$-competitive with the worst-case predictions.

**Theorem C.10** (Theorem 4.4 Restated). *RG is $(n-1) \cdot P + 1$-competitive with arbitrarily bad predictions, and this bound is tight.*

*Proof for Theorem 4.4.* The non-lazy RG is $(n-1) \cdot P + 1$-competitive by Theorem 3.7. To see the bound is tight, we show that, for any small $\epsilon > 0$, there exists a set of $n$ jobs with the associated predictions such that the max-stretch obtained by RG is at least $(n-1) \cdot P + 1 - \epsilon$ times the optimum. Fix any $0 < \epsilon < 1$. Let $\omega = \frac{\epsilon \cdot P}{(n-1) \cdot P + (1-\epsilon)}$. Construct $n$ jobs: $p_1^* = 1, r_1 = 0, p_1 = \frac{n^4 \cdot P^2}{\omega^2} + 3$ and $p_j^* = P, r_j = (1 - \omega) + (j - 2) \cdot P, p_j = 1$ for all $j \geq 2$. For this instance, we have $p_{\min} = 1$ and $p_{\max} = p_1$. The optimal schedule processes job $J_1, J_2, ..., J_n$ in order and has max-stretch $1 + \frac{\omega}{P}$. Algorithm RG will classify job $J_1$ as a long job and the rest as short jobs. In the execution of RG, job $J_1$ is not processed in the intervals $(r_j + \frac{r_j}{\sqrt{p_1} - 1}, C_j^{\mathsf{RG}}]$ as $\widetilde{S_1}(t) < \widetilde{S_j}(t)$ for $t \in (r_j + \frac{r_j}{\sqrt{p_1} - 1}, C_j^{\mathsf{RG}}]$ for all $2 \leq j \leq n$. Thus, before $C_n^{\mathsf{RG}}$, job $J_1$ has been processed for at most:

$$(1 - \omega) + \sum_{j=2}^{n} \frac{r_j}{\sqrt{p_1} - 1} = (1 - \omega) + \sum_{j=2}^{n} \frac{(1 - \omega) + (j - 2) \cdot P}{\sqrt{\frac{n^4 \cdot P^2}{\omega^2} + 3} - 1} < 1$$

At $C_n^{\mathsf{RG}}$, jobs $J_2, ..., J_n$ are completed, leaving $J_1$ to be completed at time $(n - 1) \cdot P + 1$. The max-stretch obtained by RG is $(n - 1) \cdot P + 1$, which is $(n - 1) \cdot P + 1 - \epsilon$ times $1 + \frac{\omega}{P}$, the optimum max-stretch. $\quad\square$

We show RG's performance for its run-time and space complexity and preemptions.

**Lemma C.11.** *A short job $J_j$ remains to have the largest relaxed-stretch among active jobs for all time $t \geq t'$ before its completion if it had the largest relaxed-stretch among active jobs at some time $t'$.*

*Proof.* Suppose job $J_j$ has the largest relaxed-stretch among active jobs at time $t'$. For any short job $J_k$ active at time $t'$, we have $\widetilde{S_j}(t') \geq \widetilde{S_k}(t') \Rightarrow r_j \leq r_k$. Fix any time $t \geq t'$. Any short job $J_k$ active at time $t$ (including those released after $t'$) has $r_j \leq r_k$ and, thus, $\widetilde{S_j}(t) \geq \widetilde{S_k}(t)$. For any long job $J_k$ released after $t'$, we have $\frac{t-r_j}{\alpha} \geq \frac{t-r_k}{\beta} \Rightarrow \widetilde{S_j}(t) \geq \widetilde{S_k}(t)$, as $r_j \leq r_k$ and $\alpha < \beta$. For any long job $J_k$ active at time $t'$, we have $\frac{t'-r_j}{\alpha} \geq \frac{t'-r_k}{\beta} \Rightarrow t' \geq \frac{\beta \cdot r_j - \alpha \cdot r_k}{\beta - \alpha} \Rightarrow t \geq \frac{\beta \cdot r_j - \alpha \cdot r_k}{\beta - \alpha} \Rightarrow \widetilde{S_j}(t) \geq \widetilde{S_k}(t)$. $\square$

**Theorem C.12** (Theorem 4.6 Restated). *Relaxed-Greedy admits an implementation with $O(1)$ run-time to make any online decision and $O(n)$ space.*

*Proof for Theorem 4.6.* Main two First-In-First-Out (FIFO) queues: *short queue* and *long queue*. The short/long queue contains short/long jobs in order of release time. The algorithm always processes the job with the largest relaxed-stretch at the front of the two queues and removes a job from the (front of the) queue when the job is completed. We claim that one of the jobs at the front of the queues has the overall largest relaxed-stretch at any time $t$. To see this, it holds that $r_j \leq r_k \Rightarrow \frac{t-r_j}{\alpha} \geq \frac{t-r_k}{\alpha} \Rightarrow \widetilde{S_j}(t) \geq \widetilde{S_k}(t)$ for any short jobs $J_j, J_k$, and that $r_j \leq r_k \Rightarrow \frac{t-r_j}{\beta} \geq \frac{t-r_k}{\beta} \Rightarrow \widetilde{S_j}(t) \geq \widetilde{S_k}(t)$ for any long jobs $J_j, J_k$. Therefore, Lines 17 and 22 in Algorithm 1 each require $O(1)$ time to run. The rest operations take $O(1)$ time to run. Finally, since the algorithm maintains at most all jobs in memory and a constant number of support variables, the space required is $O(n)$. $\square$

Instead of running JobSwitch at every time $t$, the algorithm may compute a critical time $t'$ such that $\frac{t'-r_j}{\alpha} = \frac{t'-r_k}{\beta}$, if a long job $J_k$ is currently being processed, where $J_j$ and $J_k$ are the short (if there are any) and long jobs at the front of the queues. Update or re-compute this $t'$ if $J_k$ is completed, or a new short job is released when there is no short job prior to the job's release. The algorithm calls JobSwitch only at time $t'$, which minimizes the number of function calls. This is supported by the observation that JobSwitch is executed only when a long job is being processed by Lemma C.11.

**Theorem C.13** (Theorem 4.7 Restated). *Relaxed-Greedy generates a schedule with at most $n-1$ preemptions. Thus, the amortized number of preemptions per job is at most $1$.*

*Proof for Theorem 4.7.* Preemptions are triggered by JobSwitch, which is not triggered while a short job is being processed by Lemma C.11. A long job may be preempted by short jobs; these short jobs, however, will not be preempted by Lemma C.11. Thus, the number of preemptions is bounded by the total number of short jobs, which is at most $n-1$. No preemptions occur if all jobs have the same job size prediction. Therefore, the amortized number of preemptions (per job) is at most $\frac{n-1}{n} \leq 1$. $\square$

# D  DISCUSSION OF THE CONSTANT IN RELAXED-STRETCH FOR SECTION 4.2

We show that replacing the constant $\sqrt{3}$ in $\alpha$ with any other constant maintains RG's competitive ratio, as long as $\alpha < \beta$. The idea applies to the family of RG-based algorithms, including ARG, PRG, $RG^x$, and their RR-augmentations. For simplicity, we focus our discussion on RG.

Fix any constant $\mu$ ($\mu > \sqrt{\frac{p_{\min}}{p_{\max}}}$). Set $\alpha = \frac{\sqrt{p_{\min} \cdot p_{\max}}}{\mu}$ and $\beta = p_{\max}$ in RG. It holds $\alpha < \beta$. With parameters $\alpha$ and $\beta$, we (re-)use the definitions, lemmas, and notations from Section 4.2. We show the following theorem.

**Theorem D.1.** *Relaxed-Greedy is $\max\{\mu, \frac{3}{\mu}\} \cdot \eta^3 \cdot \sqrt{P}$-competitive with $\alpha = \frac{\sqrt{p_{\min} \cdot p_{\max}}}{\mu}$, $\beta = p_{\max}$, and $\mu > \sqrt{\frac{p_{\min}}{p_{\max}}}$, for online scheduling with predictions.*

---

**Algorithm 3:** Adaptive Relaxed-Greedy

---

**Data** : job size prediction $p_j$ upon job release and a prediction for minimum job size $p_{\min}^e$
**Result** : schedule with a max-stretch $O(\lambda^{0.5} \cdot \eta^{2.5} \cdot \sqrt{P})$ times the optimum max-stretch

1  $p_{\min} \leftarrow p_{\min}^e, p_{\max} \leftarrow -1, \alpha \leftarrow -1, \beta \leftarrow -1$
2  **Function** `relaxedStretch(j,t)` `// computes adaptive relaxed-stretch for job`
    $J_j$ `at` $t$
3      **if** $J_j$ *is completed before* $t$ **then**
4         $t \leftarrow C_j$
5      **if** $p_j \leq \alpha_{r_j}$ **then**
6         **return** $\frac{t-r_j}{\alpha_{r_j}}$
7      **else**
8         **return** $\frac{t-r_j}{\beta_{r_j}}$

9  **Event Function** `JobRelease()` `// job` $J_j$ `is released at time` $r_j$
10     **if** $\mathbf{U} = \emptyset$ **then**
11        run $J_j$ on the machine.
12     insert $J_j$ into $\mathbf{U}$.
13     $p_{\max} \leftarrow \max(p_{\max}, p_j), \alpha \leftarrow \frac{\sqrt{p_{\min} \cdot p_{\max}}}{\sqrt{3}}, \beta \leftarrow \max(p_{\max}, \alpha), \alpha_{r_j} \leftarrow \alpha, \beta_{r_j} \leftarrow \beta$
14 The other functions (JobComplete, JobSwitch) remain the same as in Algorithm 1.

---

*Proof.* Consider the execution of RG. Pick any job $J_j$. If job $J_j$ is a long job ($p_j > \alpha = \frac{\sqrt{p_{\min} \cdot p_{\max}}}{\mu}$), we have:

$$S_j^{\mathsf{RG}} = \frac{C_j^{\mathsf{RG}} - r_j}{p_j^*} = \frac{C_j^{\mathsf{RG}} - r_j}{\beta} \cdot \frac{\beta}{p_j^*} \leq \widetilde{S_j^{\mathsf{RG}}} \cdot \eta \cdot \frac{\beta}{p_j} < \widetilde{S_j^{\mathsf{RG}}} \cdot \eta \cdot \mu \cdot \frac{p_{\max}}{\sqrt{p_{\min} \cdot p_{\max}}}$$

$$\leq \widetilde{S_j^{\mathsf{RG}}} \cdot \eta \cdot \mu \cdot \sqrt{\frac{\eta \cdot p_{\max}^*}{p_{\min}^*/\eta}} = \mu \cdot \eta^2 \cdot \sqrt{P} \cdot \widetilde{S_j^{\mathsf{RG}}} \leq \mu \cdot \eta^2 \cdot \sqrt{P} \cdot \widetilde{S^*} \leq \mu \cdot \eta^3 \cdot \sqrt{P} \cdot S^*$$

where the second last inequality is by Lemma C.6, and the last inequality is by Lemma C.7. If job $J_j$ is a short job, we have:

$$S_j^{\mathsf{RG}} = \frac{C_j^{\mathsf{RG}} - r_j}{p_j^*} \leq \frac{C_j^{\mathsf{RG}} - r_j}{p_{\min}^*} = \frac{C_j^{\mathsf{RG}} - r_j}{\alpha} \cdot \frac{\alpha}{p_{\min}^*} = \widetilde{S_j^{\mathsf{RG}}} \cdot \frac{\sqrt{p_{\min} \cdot p_{\max}}}{\mu \cdot p_{\min}^*}$$

$$\leq \widetilde{S_j^{\mathsf{RG}}} \cdot \frac{\sqrt{\eta \cdot p_{\min}^* \cdot \eta \cdot p_{\max}^*}}{\mu \cdot p_{\min}^*} = \frac{\eta}{\mu} \cdot \sqrt{P} \cdot \widetilde{S_j^{\mathsf{RG}}} \leq \frac{3}{\mu} \cdot \eta \cdot \sqrt{P} \cdot \widetilde{S^*} \leq \frac{3}{\mu} \cdot \eta^2 \cdot \sqrt{P} \cdot S^*$$

where the second last inequality is by Lemma C.6, and the last inequality is by Lemma C.7. Therefore, we have $S^{\mathsf{RG}} = \max_{1 \leq j \leq n} S_j^{\mathsf{RG}} \leq \max\{\frac{3}{\mu}, \mu\} \cdot \eta^3 \cdot \sqrt{P} \cdot S^*$. $\qquad \square$

The implications of Theorem D.1 are as follows. Altering the constant $\sqrt{3}$ in $\alpha$ preserves RG's $O(\eta^3 \cdot \sqrt{P})$ competitive ratio, i.e., the choice does not affect the asymptotic performance guarantee. This opens the door to searching for the appropriate constant for specific applications via algorithm engineering. In addition, choosing $\mu = \sqrt{3}$ yields the optimal constant in the competitive ratio, as $\max\{\frac{3}{\mu}, \mu\} \geq \sqrt{3}$ for all $\mu > 0$ and the minimum is obtained at $\mu = \sqrt{3}$. This justifies our choice of $\sqrt{3}$ in the definition of $\alpha$.

## E   MISSING PROOFS FOR SECTION 4.3

We introduce subscript $t$ to denote the value of a variable at time $t$. Thus, $p_{\max}$ at time $t$ (denoted by $p_{\max,t}$) represents the maximum job size prediction seen so far, i.e., $p_{\max,t} = \max_{r_j \leq t} p_j$. Parameters $\alpha$ and $\beta$ are functions in $p_{\min}$ and $p_{\max}$ and thus are time-varying. They are updated with updates of $p_{\max}$ in JobRelease. Formally, ARG sets $\alpha_t = \frac{\sqrt{p_{\min}^e \cdot p_{\max,t}}}{\sqrt{3}}$ and $\beta_t = \max\{p_{\max,t}, \alpha_t\}$,

where $\alpha_t$ and $\beta_t$ denote the values of $\alpha$ and $\beta$ at time $t$. We record $\alpha_{r_j}$ and $\beta_{r_j}$ for every job $J_j$. These values are used to compute the *adaptive relaxed-stretch* of $J_j$, defined as follows.

**Definition E.1** ($\alpha$-$\beta$ Adaptive Relaxed-Stretch). Define $\alpha$-$\beta$ adaptive relaxed-stretch $\widehat{S_{j,\alpha,\beta}}(t)$ for job $J_j$ at time $t$ given $\alpha$ and $\beta$: $\widehat{S_{j,\alpha,\beta}}(t) = \frac{t - r_j}{\alpha_{r_j}}$ if $J_j$ is active at time $t$ and $p_j \leq \alpha_{r_j}$; $\widehat{S_{j,\alpha,\beta}}(t) = \frac{t - r_j}{\beta_{r_j}}$ if $J_j$ is active at time $t$ and $p_j > \alpha_{r_j}$. We use $\widehat{S_{j,\alpha,\beta}}$ to denote the $\alpha$-$\beta$ adaptive relaxed-stretch at the completion of job $J_j$: $\widehat{S_{j,\alpha,\beta}} = \frac{C_j - r_j}{\alpha_{r_j}}$ if $p_j \leq \alpha_{r_j}$; $\widehat{S_{j,\alpha,\beta}} = \frac{C_j - r_j}{\beta_{r_j}}$ if $p_j > \alpha_{r_j}$. Fixing a schedule, we use *max-adaptive-relaxed-stretch*, denoted by $\widehat{S_{\alpha,\beta}}$ to represent the maximum $\alpha$-$\beta$ adaptive relaxed-stretch of all jobs: $\widehat{S_{\alpha,\beta}} = \max_{1 \leq j \leq n} \widehat{S_{j,\alpha,\beta}}$. We drop the subscripts $\alpha$ and $\beta$ from the notations and the prefix $\alpha$-$\beta$ when the context is clear for simplicity.

The classification of job $J_j$ now depends on the relationship between $p_j$ and $\alpha_{r_j}$.

**Definition E.2** ($\alpha$-$\beta$ $r$-short and $r$-long Jobs). Given parameters $\alpha$ and $\beta$, a job $J_j$ is said to be $r$-short if $p_j \leq \alpha_{r_j}$; a job $J_j$ is said to be $r$-long otherwise.

**Definition E.3** (Minimum max-adaptive-relaxed-stretch). We use $\widehat{S^*_{\alpha,\beta}}$ to denote the minimum max-adaptive-relaxed-stretch, given any $\alpha$ and $\beta$. The existence of this quantity is due to Theorem A.1. We drop the subscripts $\alpha$ and $\beta$ from the notations when the context is clear.

Define $\widehat{d_j}$ to be the deadline of completing job $J_j$ to achieve the minimum max-adaptive-relaxed-stretch, i.e., $\widehat{d_j} = r_j + \widehat{S^*} \cdot \alpha_{r_j}$ for $r$-short job $J_j$ and $\widehat{d_j} = r_j + \widehat{S^*} \cdot \beta_{r_j}$ for $r$-long job $J_j$. A schedule achieves the optimal max-adaptive-relaxed-stretch if every job $J_j$ completes no later than $\widehat{d_j}$. Define set $\widehat{P_j}(t) = \{J_x | \widehat{d_x} \leq \widehat{d_j}, J_x \text{ active at time } t\}$ and set $\widehat{F_j}(t) = \{J_x | \widehat{d_x} > \widehat{d_j}, J_x \text{ active at time } t\}$ to be the predecessors and followers of $J_j$ at time $t$ in terms of minimizing the max-adaptive-relaxed-stretch.

**Definition E.4** (Definition of $\oslash$-delay). Job $J_j$ is said to be $\oslash$-*delayed* by another job $J_k$ at time $t$, if the following four conditions are simultaneously met: (1) $J_j$ is not completed or not released at time $t$, (2) $J_k \in \widehat{F_j}(t)$, (3) $\widehat{P_j}(t) \neq \emptyset$, and (4) ARG is processing $J_k$ at time $t$ with $\max_{J_x \in \widehat{P_j}(t)} \widehat{S_x}(t) < \widehat{S_k}(t)$.

The symbol $\oslash$ is only to distinguish $\oslash$-*delay* from *delay*; it does not contain any other meaning. We first make the following observation.

*Observation* E.5. For any $x \leq y$, it holds (1) $\alpha_x \leq \beta_x$, (2) $\alpha_x \leq \alpha_y$, and (3) $\beta_x \leq \beta_y$.

**Lemma E.6** (No Job $\oslash$-delays a $r$-long Job). *A $r$-long job is never $\oslash$-delayed.*

*Proof.* Fix a $r$-long job $J_j$. Consider, for the proof by contradiction, another job $J_k \in \widehat{F_j}(t)$ $\oslash$-delays $J_j$ at some time $t$. Observe that $r_j < r_k$, since otherwise, we would have $\widehat{d_j} = r_j + \widehat{S^*} \cdot \beta_{r_j} \geq r_k + \widehat{S^*} \cdot \beta_{r_k} \geq \widehat{d_k}$. Thus, $J_j$ is active at time $t$. Job $J_k$ must be $r$-short, since otherwise, $\widehat{S_j}(t) = \frac{t - r_j}{\beta_{r_j}} > \frac{t - r_k}{\beta_{r_k}} = \widehat{S_k}(t)$. We have:

$$\widehat{S_j}(t) < \widehat{S_k}(t) \Rightarrow \frac{t - r_j}{\beta_{r_j}} < \frac{t - r_k}{\alpha_{r_k}} \Rightarrow \beta_{r_j} > \alpha_{r_k} \text{ and } t > \frac{r_k \cdot \beta_{r_j} - r_j \cdot \alpha_{r_k}}{\beta_{r_j} - \alpha_{r_k}} \geq r_j + \widehat{S^*} \cdot \beta_{r_j} = \widehat{d_j}$$

where the last inequality is due to $\widehat{d_k} > \widehat{d_j} \Rightarrow r_k + \widehat{S^*} \cdot \alpha_{r_k} > r_j + \widehat{S^*} \cdot \beta_{r_j}$. Thus, no job $\oslash$-delays $J_j$ before time $\widehat{d_j}$. However, in this case, job $J_j$ completes no later than $\widehat{d_j}$, which is earlier than $t$, contradicting the definition of $\oslash$-delay. $\square$

**Lemma E.7** (Bound on the Delay for a $r$-short Job). *A $r$-short job $J_j$ completes no later than* $r_j + 3 \cdot \widehat{S^*} \cdot \alpha_{r_j}$.

*Proof.* We claim that any $r$-short job can only be $\oslash$-delayed before $\widehat{d_j}$, i.e., some job $J_k \in \widehat{F_j}(t)$ $\oslash$-delays $J_j$ at time $t$ implies $t < \widehat{d_j}$. We show this via proof by contradiction. Fix a $r$-short job $J_j$.

Suppose another job $J_k \in \widehat{F_j}(t)$ $\oslash$-delays $J_j$ at some time $t \geq \widehat{d_j}$. Job $J_j$ must be active at time $t$. Observe that $r_j > r_k$, since otherwise, we would have $\widehat{d_j} = r_j + \widehat{S^*} \cdot \alpha_{r_j} \leq r_k + \widehat{S^*} \cdot \alpha_{r_k} \leq \widehat{d_k}$. Job $J_k$ must be $r$-long, since otherwise, $\widehat{d_k} = r_k + \widehat{S^*} \cdot \alpha_{r_k} < r_j + \widehat{S^*} \cdot \alpha_{r_j} = \widehat{d_j}$. Then, it follows $\widehat{d_k} > \widehat{d_j} \Rightarrow r_k + \widehat{S^*} \cdot \beta_{r_k} > r_j + \widehat{S^*} \cdot \alpha_{r_j} \Rightarrow \beta_{r_k} > \alpha_{r_j}$. We have:

$$\widehat{S_k}(t) > \widehat{S_j}(t) \Rightarrow \frac{t - r_k}{\beta_{r_k}} > \frac{t - r_j}{\alpha_{r_j}} \Rightarrow t < \frac{r_j \cdot \beta_{r_k} - r_k \cdot \alpha_{r_j}}{\beta_{r_k} - \alpha_{r_j}} < r_j + \widehat{S^*} \cdot \alpha_{r_j} = \widehat{d_j}$$

where the last inequality is due to $\widehat{d_k} > \widehat{d_j} \Rightarrow r_k + \widehat{S^*} \cdot \beta_{r_k} > r_j + \widehat{S^*} \cdot \alpha_{r_j}$. It follows $\widehat{d_j} \leq t < \widehat{d_j}$, contradicting that $J_k$ $\oslash$-delays job $J_j$ at time $t \geq \widehat{d_j}$.

Next, consider any $r$-short job $J_j$. Let $\tau_j$ denote the earliest time after which no job $\oslash$-delays $J_j$. Every job $J_x \in \widehat{P_j}(\tau_j)$ is $\oslash$-delayed at time $\tau_j$ by the job $J_l$ running at time instant $\tau_j$, since $\widehat{d_x} \leq \widehat{d_j} < \widehat{d_l}$. Thus, every job $J_x \in \widehat{P_j}(\tau_j)$ is $r$-short by Lemma E.6. It follows that, for any job $J_x \in \widehat{P_j}(\tau_j)$, $\widehat{d_x} \leq \widehat{d_j} \Rightarrow r_x + \widehat{S^*} \cdot \alpha_{r_x} \leq r_j + \widehat{S^*} \cdot \alpha_{r_j} \Rightarrow r_x \leq r_j$ and $\tau_t \leq \widehat{d_x} \Rightarrow r_x \geq \tau_j - \widehat{S^*} \cdot \alpha_{r_x} \geq \tau_j - \widehat{S^*} \cdot \alpha_{r_j}$. We bound the total amount of delay on $J_j$ by $\sum_{J_x \in \widehat{P_j}(\tau_j)} p_x^*$. Consider the optimal schedule (in minimizing adaptive-relaxed-stretch) constructed by EDF with $\widehat{S^*}$. We have two cases.

**Case 1** ($\tau_j < r_j$)**.** In the optimal schedule, all jobs in $\widehat{P_j}(\tau_j)$ are released no earlier than $\tau_j - \widehat{S^*} \cdot \alpha_{r_j}$ and no later than $\tau_j$ and completed no later than $\tau_j + \widehat{S^*} \cdot \alpha_y \leq \tau_j + \widehat{S^*} \cdot \alpha_{r_j}$, where $y$ denotes $\max_{J_x \in \widehat{P_j}(\tau_j)} r_x$. We have:

$$\sum_{J_x \in \widehat{P_j}(\tau_j)} p_x^* \leq \tau_j + \widehat{S^*} \cdot \alpha_{r_j} - (\tau_j - \widehat{S^*} \cdot \alpha_{r_j}) = 2 \cdot \widehat{S^*} \cdot \alpha_{r_j}$$

**Case 2** ($\tau_j \geq r_j$)**.** In the optimal schedule, all jobs in $\widehat{P_j}(\tau_j)$ (including job $J_j$) are released no earlier than $\tau_j - \widehat{S^*} \cdot \alpha_{r_j}$ and completed no later than $\widehat{d_j} = r_j + \widehat{S^*} \cdot \alpha_{r_j}$. We have:

$$\sum_{J_x \in \widehat{P_j}(\tau_j)} p_x^* \leq \widehat{d_j} - (\tau_j - \widehat{S^*} \cdot \alpha_{r_j}) = r_j + 2 \cdot \widehat{S^*} \cdot \alpha_{r_j} - \tau_j \leq 2 \cdot \widehat{S^*} \cdot \alpha_{r_j}$$

Thus, the completion time for a $r$-short job $J_j$ in ARG is at most $\widehat{d_j} + \sum_{J_x \in \widehat{P_j}(\tau_j)} p_x^* \leq r_j + 3 \cdot \widehat{S^*} \cdot \alpha_{r_j}$ $\qquad \square$

Let $\widehat{C_j^{EDF}}$ and $C_j^{ARG}$ be the completion time of $J_j$ in the schedule constructed by EDF with $\widehat{S^*}$ and ARG. Let $\widehat{S_j^*}$ and $\widehat{S_j^{ARG}}$ be the adaptive relaxed-stretch of $J_j$ in these two schedules.

**Lemma E.8.** *It holds $\widehat{S_j^{ARG}} \leq \widehat{S^*}$ for any $r$-long job $J_j$ and $\widehat{S_j^{ARG}} \leq 3 \cdot \widehat{S^*}$ for any $r$-short job $J_j$.*

*Proof.* Pick any job $J_j$. If $J_j$ is a $r$-long job, it follows by Lemma E.6 that:

$$C_j^{ARG} \leq \widehat{C_j^{EDF}} + \int_{\text{any job } \oslash\text{-delays } J_j \text{ at time } t} dt = \widehat{C_j^{EDF}} \Rightarrow \widehat{S_j^{ARG}} \leq \widehat{S_j^*} \leq \widehat{S^*}$$

If $J_j$ is a $r$-short job by Lemma E.7, it follows that:

$$\widehat{S_j^{ARG}} = \frac{C_j^{ARG} - r_j}{\alpha_{r_j}} \leq \frac{(r_j + 3 \cdot \widehat{S^*} \cdot \alpha_{r_j}) - r_j}{\alpha_{r_j}} = 3 \cdot \widehat{S^*} \qquad \square$$

**Lemma E.9.** $\widehat{S^*} \leq \eta \cdot S^*$, *with* $\alpha_t = \frac{\sqrt{p_{\min}^e \cdot p_{\max,t}}}{\sqrt{3}}$ *and* $\beta_t = \max\{p_{\max,t}, \alpha_t\}$.

*Proof.* Let $\widehat{S^{OPT}}$ be the max-adaptive-relaxed-stretch of the optimal schedule for the max-stretch problem. It follows $\widehat{S^*} \leq \widehat{S^{OPT}}$. Consider the optimal schedule with the minimum max-stretch.

Let $C_j^*$ and $\widehat{S_j^{OPT}}$ be the completion time of job $J_j$ and the adaptive relaxed-stretch of $J_j$ in this schedule. Pick any job $J_j$. If job $J_j$ is a $r$-short job, it follows $p_j \le \alpha_{r_j}$ and $\widehat{S_j^{OPT}} = \frac{C_j^* - r_j}{\alpha_{r_j}} \le \frac{C_j^* - r_j}{p_j} \le \eta \cdot \frac{C_j^* - r_j}{p_j^*} \le \eta \cdot S^*$. If job $J_j$ is a $r$-long job, it follows $p_j \le \max_{r_x \le r_j} p_x = p_{\max, r_j} = \beta_{r_j}$ and $\widehat{S_j^{OPT}} = \frac{C_j^* - r_j}{\beta_{r_j}} \le \frac{C_j^* - r_j}{p_j} \le \eta \cdot \frac{C_j^* - r_j}{p_j^*} \le \eta \cdot S^*$. Combining these two cases gives us $\widehat{S^*} \le \widehat{S^{OPT}} = \max_{1 \le j \le n} \widehat{S_j^{OPT}} \le \eta \cdot S^*$. $\qquad\square$

Let $S_j^{\mathsf{ARG}}$ and $\widehat{S_j^{\mathsf{ARG}}}$ be the stretch and adaptive relaxed-stretch of job $J_j$ in the execution of $\mathsf{ARG}$.

**Lemma E.10.** *It holds $S_j^{\mathsf{ARG}} \le \sqrt{3} \cdot \lambda^{0.5} \cdot \eta^{1.5} \cdot \sqrt{P} \cdot \widehat{S_j^{\mathsf{ARG}}}$ for any $r$-long job $J_j$ and $S_j^{\mathsf{ARG}} \le \frac{\lambda^{0.5} \cdot \eta^{0.5}}{\sqrt{3}} \cdot \sqrt{P} \cdot \widehat{S_j^{\mathsf{ARG}}}$ for any $r$-short job $J_j$, with $\alpha_t = \frac{\sqrt{p_{\min}^e \cdot p_{\max, t}}}{\sqrt{3}}$ and $\beta_t = \max\{p_{\max, t}, \alpha_t\}$.*

*Proof.* Pick any job $J_j$. If job $J_j$ is $r$-long, it follows $p_j > \alpha_{r_j} = \frac{\sqrt{p_{\min}^e \cdot p_{\max, r_j}}}{\sqrt{3}}$. We have:

$$S_j^{\mathsf{ARG}} = \frac{C_j^{\mathsf{ARG}} - r_j}{p_j^*} = \frac{C_j^{\mathsf{ARG}} - r_j}{\beta_{r_j}} \cdot \frac{\beta_{r_j}}{p_j^*} \le \widehat{S_j^{\mathsf{ARG}}} \cdot \eta \cdot \frac{\beta_{r_j}}{p_j} < \widehat{S_j^{\mathsf{ARG}}} \cdot \eta \cdot \sqrt{3} \cdot \frac{p_{\max, r_j}}{\sqrt{p_{\min}^e \cdot p_{\max, r_j}}}$$

$$\le \widehat{S_j^{\mathsf{ARG}}} \cdot \eta \cdot \sqrt{3} \cdot \sqrt{\frac{p_{\max, r_j}}{p_{\min}^e}} \le \widehat{S_j^{\mathsf{ARG}}} \cdot \eta \cdot \sqrt{3} \cdot \sqrt{\frac{\eta \cdot p_{\max}^*}{p_{\min}^*/\lambda}} = \sqrt{3} \cdot \lambda^{0.5} \cdot \eta^{1.5} \cdot \sqrt{P} \cdot \widehat{S_j^{\mathsf{ARG}}}$$

If job $J_j$ is $r$-short, we have:

$$S_j^{\mathsf{ARG}} \le \frac{C_j^{\mathsf{ARG}} - r_j}{p_{\min}^*} = \frac{C_j^{\mathsf{ARG}} - r_j}{\alpha_{r_j}} \cdot \frac{\alpha_{r_j}}{p_{\min}^*} \le \widehat{S_j^{\mathsf{ARG}}} \cdot \frac{\sqrt{\lambda \cdot p_{\min}^* \cdot \eta \cdot p_{\max}^*}}{\sqrt{3} \cdot p_{\min}^*} = \frac{\lambda^{0.5} \cdot \eta^{0.5}}{\sqrt{3}} \cdot \sqrt{P} \cdot \widehat{S_j^{\mathsf{ARG}}}$$

The derivation uses the observation: $p_{\max, t} = \max_{r_x \le t} p_x \le p_{\max} \le \eta \cdot p_{\max}^*$ for any $t \ge 0$. $\qquad\square$

**Theorem E.11** (Theorem 4.10 Restated). *Adaptive Relaxed-Greedy is $\sqrt{3} \cdot \lambda^{0.5} \cdot \eta^{2.5} \cdot \sqrt{P}$-competitive for online scheduling with predictions.*

*Proof for Theorem 4.10.* If job $J_j$ is $r$-long, we have, by Lemmas E.10, E.8, and E.9, that:

$$S_j^{\mathsf{ARG}} \le \sqrt{3} \cdot \lambda^{0.5} \cdot \eta^{1.5} \cdot \sqrt{P} \cdot \widehat{S_j^{\mathsf{ARG}}} \le \sqrt{3} \cdot \lambda^{0.5} \cdot \eta^{1.5} \cdot \sqrt{P} \cdot \widehat{S^*} \le \sqrt{3} \cdot \lambda^{0.5} \cdot \eta^{2.5} \cdot \sqrt{P} \cdot S^*$$

If job $J_j$ is $r$-short, we have, by Lemmas E.10, E.8, and E.9, that:

$$S_j^{\mathsf{ARG}} \le \frac{\lambda^{0.5} \cdot \eta^{0.5}}{\sqrt{3}} \cdot \sqrt{P} \cdot \widehat{S_j^{\mathsf{ARG}}} \le \frac{\lambda^{0.5} \cdot \eta^{0.5}}{\sqrt{3}} \cdot \sqrt{P} \cdot 3 \cdot \widehat{S^*} \le \sqrt{3} \cdot \lambda^{0.5} \cdot \eta^{1.5} \cdot \sqrt{P} \cdot S^*$$

Therefore, we have $S^{\mathsf{ARG}} \le \sqrt{3} \cdot \lambda^{0.5} \cdot \eta^{2.5} \cdot \sqrt{P} \cdot S^*$. $\mathsf{ARG}$ is $\sqrt{3} \cdot \lambda^{0.5} \cdot \eta^{2.5} \cdot \sqrt{P}$-competitive. $\qquad\square$

Algorithm $\mathsf{ARG}$ is $(n-1) \cdot P + 1$-robust. The robustness result is tight by the same construction used in Theorem 4.4 with $p_{\min}^e = 1$, where $\mathsf{ARG}$ sets $\alpha = \frac{\sqrt{p_1}}{\sqrt{3}}$ and $\beta = p_1$ at time 0, remains these parameters constants, and produces the same result as $\mathsf{RG}$.

For proving the run-time and space complexity, as well as the number of preemptions, we first show an equivalent version of Lemma C.11 for $\mathsf{ARG}$.

**Lemma E.12.** *A $r$-short job $J_j$ remains to have the largest adaptive relaxed-stretch among active jobs for all time $t \ge t'$ before its completion if it had the largest adaptive relaxed-stretch among active jobs at some time $t'$.*

*Proof.* Suppose job $J_j$ has the largest adaptive relaxed-stretch among active jobs at time $t'$. For any $r$-short job $J_k$ active at time $t'$, we have $\widehat{S_j}(t') \ge \widehat{S_k}(t') \Rightarrow \frac{t' - r_j}{\alpha_{r_j}} \ge \frac{t' - r_k}{\alpha_{r_k}} \Rightarrow r_j \le r_k$. Fix any time $t \ge t'$. Any $r$-short job $J_k$ active at time $t$ (including those released after $t'$) has

$r_j \leq r_k$ and $\alpha_{r_j} \leq \alpha_{r_k}$, thus, $\widehat{S_j}(t) \geq \widehat{S_k}(t)$. For any $r$-long job $J_k$ released after $t'$, we have $\frac{t-r_j}{\alpha_{r_j}} \geq \frac{t-r_k}{\beta_{r_k}} \Rightarrow \widehat{S_j}(t) \geq \widehat{S_k}(t)$, as $r_j \leq r_k$ and $\alpha_{r_j} \leq \beta_{r_j} \leq \beta_{r_k}$. For any $r$-long job $J_k$ active at time $t'$, we have:

$$\frac{t'-r_j}{\alpha_{r_j}} \geq \frac{t'-r_k}{\beta_{r_k}} \Rightarrow t' \geq \frac{\beta_{r_k} \cdot r_j - \alpha_{r_j} \cdot r_k}{\beta_{r_k} - \alpha_{r_j}} \Rightarrow t \geq \frac{\beta_{r_k} \cdot r_j - \alpha_{r_j} \cdot r_k}{\beta_{r_k} - \alpha_{r_j}} \Rightarrow \widehat{S_j}(t) \geq \widehat{S_k}(t)$$

Therefore, job $J_j$ remains to have the largest adaptive relaxed-stretch among active jobs at time $t$. $\quad\square$

**Theorem E.13** (Theorem 4.12 Restated). *Adaptive Relaxed-Greedy admits an implementation with $O(1)$ run-time to make any online decision and $O(n)$ space.*

*Proof for Theorem 4.12.* Similar to the proof for Theorem 4.6. Main two First-In-First-Out (FIFO) queues: *short queue* and *long queue*. The short queue contains $r$-short jobs in order of release time; the long queue contains $r$-long jobs in order of release time. The algorithm always processes the job with the largest adaptive relaxed-stretch at the front of the two queues and removes a job from the (front of the) queue when the job is completed. We show that one of the jobs at the front of the queues has the overall largest adaptive relaxed-stretch at any time $t$. To see this, it holds that $r_j \leq r_k \Rightarrow \frac{t-r_j}{\alpha_{r_j}} \geq \frac{t-r_k}{\alpha_{r_k}} \Rightarrow \widehat{S_j}(t) \geq \widehat{S_k}(t)$ for any $r$-short jobs $J_j, J_k$ and that $r_j \leq r_k \Rightarrow \frac{t-r_j}{\beta_{r_j}} \geq \frac{t-r_k}{\beta_{r_k}} \Rightarrow \widehat{S_j}(t) \geq \widehat{S_k}(t)$ for any $r$-long jobs $J_j, J_k$. Finally, as the algorithm maintains at most all jobs in memory and a constant number of support variables, the space required is at most $O(n)$. $\quad\square$

**Theorem E.14** (Theorem 4.13 Restated). *Adaptive Relaxed-Greedy generates a schedule with at most $n-1$ preemptions. Thus, the amortized number of preemptions per job is at most $1$.*

*Proof for Theorem 4.13.* Preemptions are triggered only by JobSwitch. Observe that JobSwitch is not triggered while a $r$-short job is being processed by Lemma E.12. A $r$-long job may be preempted while being processed by $r$-short jobs; these $r$-short jobs, however, will not be preempted by Lemma E.12. Thus, the number of preemptions is bounded by the total number of $r$-short jobs, which is at most $n-1$. Note that no preemptions occur if all jobs have the same job size prediction $p_{\max}$. As a result, the amortized number of preemptions per job is at most $\frac{n-1}{n} \leq 1$. $\quad\square$

## F    MISSING PROOFS FOR SECTION 4.4

We prove that PRG achieves an $O(\lambda^{0.5} \cdot \varphi^{0.5} \cdot \eta \cdot \max\{\eta, \varphi\} \cdot \sqrt{P})$ competitive ratio. With static $\alpha$ and $\beta$, we (re-)use the definitions, lemmas, and notations from Section 4.2.

**Theorem F.1** (Theorem 4.14 Restated). *Predictive Relaxed-Greedy is $\max\{\frac{3}{\mu}, \mu\} \cdot \lambda^{0.5} \cdot \varphi^{0.5} \cdot \eta \cdot \max\{\eta, \varphi\} \cdot \sqrt{P}$-competitive for online scheduling with predictions.*

*Proof for Theorem 4.14.* Consider the execution of PRG. Pick any job $J_j$. If job $J_j$ is a long job, we have:

$$S_j^{\mathsf{PRG}} = \frac{C_j^{\mathsf{PRG}} - r_j}{\beta} \cdot \frac{\beta}{p_j^*} \leq \widetilde{S_j^{\mathsf{PRG}}} \cdot \eta \cdot \frac{\beta}{p_j} < \widetilde{S_j^{\mathsf{PRG}}} \cdot \eta \cdot \mu \cdot \frac{p_{\max}^e}{\sqrt{p_{\min}^e \cdot p_{\max}^e}} \leq \widetilde{S_j^{\mathsf{PRG}}} \cdot \eta \cdot \mu \cdot \sqrt{\frac{p_{\max}^e}{p_{\min}^e}}$$

$$\leq \widetilde{S_j^{\mathsf{PRG}}} \cdot \eta \cdot \mu \cdot \sqrt{\frac{\varphi \cdot p_{\max}^*}{p_{\min}^*/\lambda}} \leq \mu \cdot \lambda^{0.5} \cdot \varphi^{0.5} \cdot \eta \cdot \sqrt{P} \cdot \widetilde{S^*} \quad \textit{(By Lemma C.6)}$$

If job $J_j$ is a short job, we have:

$$S_j^{\mathsf{PRG}} \leq \frac{C_j^{\mathsf{PRG}} - r_j}{p_{\min}^*} \leq \widetilde{S_j^{\mathsf{PRG}}} \cdot \frac{\sqrt{\lambda \cdot p_{\min}^* \cdot \varphi \cdot p_{\max}^*}}{\mu \cdot p_{\min}^*} \leq \frac{3}{\mu} \cdot \lambda^{0.5} \cdot \varphi^{0.5} \cdot \sqrt{P} \cdot \widetilde{S^*} \quad \textit{(By Lemma C.6)}$$

We then bound $\widetilde{S^*}$ by $S^*$ through $\widetilde{S^{OPT}}$. Pick any job $J_j$. It follows $\widetilde{S_j^{OPT}} = \frac{C_j^* - r_j}{\alpha} \leq \frac{C_j^* - r_j}{p_j} \leq \eta \cdot \frac{C_j^* - r_j}{p_j^*} \leq \eta \cdot S^*$, if job $J_j$ is a short job. It follows $\widetilde{S_j^{OPT}} = \frac{C_j^* - r_j}{\beta} \leq \varphi \cdot \frac{C_j^* - r_j}{p_{\max}^*} \leq \varphi \cdot \frac{C_j^* - r_j}{p_j^*} \leq \varphi \cdot S^*$,

if job $J_j$ is a long job. Therefore, we have $\widetilde{S^*} \le \widetilde{S^{OPT}} = \max_{1 \le j \le n} \widetilde{S_j^{OPT}} \le \max\{\eta, \varphi\} \cdot S^*$, and, thus, $S^{\mathsf{PRG}} = \max_{1 \le j \le n} S_j^{\mathsf{PRG}} \le \max\{\frac{3}{\mu}, \mu\} \cdot \lambda^{0.5} \cdot \varphi^{0.5} \cdot \eta \cdot \max\{\eta, \varphi\} \cdot \sqrt{P} \cdot S^*$. $\qquad\square$

Algorithm $\mathsf{PRG}$ is $(n-1) \cdot P + 1$-robust. The robustness result for $\mathsf{PRG}$ is tight by the same construction used in Theorem 4.4 with $p_{\min}^e = 1$ and $p_{\max}^e = p_1$.

## G  MISSING PROOFS FOR SECTION 5

We show that $\mathsf{RG}^x$ is $O(\eta^{2+2x} \cdot P^{1-x})$-competitive.

**Theorem G.1** (Theorem 5.1 Restated). *$\mathsf{RG}^x$ ($0 \le x \le \frac{1}{2}$) is $\sqrt{3} \cdot \eta^{2+2x} \cdot P^{1-x}$-competitive for online scheduling with predictions.*

*Proof for Theorem 5.1.* Consider the execution of $\mathsf{RG}^x$. Pick any job $J_j$. If job $J_j$ is a long job, we have:

$$S_j^{\mathsf{RG}^x} \le \widetilde{S_j^{\mathsf{RG}^x}} \cdot \eta \cdot \frac{\beta}{p_j} < \widetilde{S_j^{\mathsf{RG}^x}} \cdot \eta \cdot \sqrt{3} \cdot \frac{p_{\max}}{p_{\min}^x \cdot p_{\max}^{1-x}} \le \sqrt{3} \cdot \eta^{1+2x} \cdot P^x \cdot \widetilde{S_j^{\mathsf{RG}^x}}$$

$$\le \sqrt{3} \cdot \eta^{1+2x} \cdot P^x \cdot \widetilde{S^*} \le \sqrt{3} \cdot \eta^{2+2x} \cdot P^x \cdot S^* \quad \text{(By Lemmas C.6 and C.7)}$$

If job $J_j$ is a short job, we have, by Lemmas C.6 and C.7, that:

$$S_j^{\mathsf{RG}^x} \le \widetilde{S_j^{\mathsf{RG}^x}} \cdot \frac{\alpha}{p_{\min}^*} \le \widetilde{S_j^{\mathsf{RG}^x}} \cdot \frac{p_{\min}^x \cdot p_{\max}^{1-x}}{\sqrt{3} \cdot p_{\min}^*} \le \frac{\eta \cdot P^{1-x}}{\sqrt{3}} \cdot \widetilde{S_j^{\mathsf{RG}^x}} \le \sqrt{3} \cdot \eta^2 \cdot P^{1-x} \cdot S^*$$

Therefore, we have $S^{\mathsf{RG}^x} = \max_{1 \le j \le n} S_j^{\mathsf{RG}^x} \le \sqrt{3} \cdot \eta^{2+2x} \cdot P^{1-x} \cdot S^*$. $\qquad\square$

**Theorem G.2** (Theorem 5.2 Restated). *$\mathsf{RG}^0$ (with $\alpha = p_{\max}, \beta = p_{\max}$), or $\mathsf{FIFO}$, is $P$-competitive.*

*Proof for Theorem 5.2.* We first show $\mathsf{RG}^0$ minimizes $\max_{1 \le j \le n} C_j - r_j$ (max-flow). If $\omega = \max_{1 \le j \le n} C_j - r_j$ (existence of $\omega$ is due to Theorem A.1), set $d_j = r_j + \omega$ for all $1 \le j \le n$ and run $\mathsf{EDF}$ with $d_j$. The resulting schedule, which is identical to the schedule generated by $\mathsf{RG}^0$, achieves the minimum max-flow. Let $J_k$ be the job with the largest flow in the schedule that minimizes max-stretch (e.g., $C_k^* - r_k = \max_{1 \le j \le n} C_j^* - r_j$). Consider the execution of $\mathsf{RG}^0$. Pick any job $J_j$. We have:

$$S_j^{\mathsf{RG}^0} = \frac{C_j^{\mathsf{RG}^0} - r_j}{p_j^*} \le \frac{\max_{1 \le q \le n} C_q^{\mathsf{RG}^0} - r_q}{p_j^*} \le \frac{\max_{1 \le q \le n} C_q^* - r_q}{p_j^*} = \frac{C_k^* - r_k}{p_k^*} \cdot \frac{p_k^*}{p_j^*} \le S^* \cdot P$$

Therefore, we have $S^{\mathsf{RG}^0} = \max_{1 \le j \le n} S_j^{\mathsf{RG}^0} \le S^* \cdot P$. $\qquad\square$

## H  CONSTRUCTION OF THE TRADE-OFF ALGORITHM FOR SECTION 5

We introduce parameters $\theta = (a_1, a_2, b_1, b_2, \mu)$ and parametrize $\alpha = \frac{p_{\min}^{a_1} \cdot p_{\max}^{a_2}}{\mu}$ and $\beta = p_{\min}^{b_1} \cdot p_{\max}^{b_2}$ in the definition of relaxed-stretch. This section shows the construction of $\mathsf{RG}^x$, the trade-off between consistency and smoothness under this parametrization.

For Lemmas C.4, C.5, and C.6 to hold, we must ensure $\alpha < \beta$. For Lemma C.7 to hold, we must ensure $p_j \le \beta$ for all $1 \le j \le n$. We will enforce these two relations after we derive the others we desire. For now, we assume that these relations hold. Run $\mathsf{RG}$ with the parametrized $\alpha$ and $\beta$. Pick any job $J_j$. If $J_j$ is a long job, it holds that $p_j > \alpha$, and we have:

$$S_j^{\mathsf{RG}} = \frac{C_j^{\mathsf{RG}} - r_j}{p_j^*} = \frac{C_j^{\mathsf{RG}} - r_j}{\beta} \cdot \frac{\beta}{p_j^*} \le \widetilde{S_j^{\mathsf{RG}}} \cdot \eta \cdot \frac{\beta}{p_j} < \widetilde{S_j^{\mathsf{RG}}} \cdot \eta \cdot \mu \cdot \frac{p_{\min}^{b_1} \cdot p_{\max}^{b_2}}{p_{\min}^{a_1} \cdot p_{\max}^{a_2}}$$

$$= \widetilde{S_j^{\mathsf{RG}}} \cdot \mu \cdot \eta \cdot \frac{p_{\max}^{b_2-a_2}}{p_{\min}^{a_1-b_1}} \le \mu \cdot \eta \cdot \frac{p_{\max}^{b_2-a_2}}{p_{\min}^{a_1-b_1}} \cdot \widetilde{S^*} \le \mu \cdot \eta^2 \cdot \frac{p_{\max}^{b_2-a_2}}{p_{\min}^{a_1-b_1}} \cdot S^* \quad \text{(By Lemma C.6 and C.7)}$$

If $J_j$ is a short job, we have:

$$S_j^{\mathsf{RG}} = \frac{C_j^{\mathsf{RG}} - r_j}{p_j^*} \le \frac{C_j^{\mathsf{RG}} - r_j}{p_{\min}^*} = \frac{C_j^{\mathsf{RG}} - r_j}{\alpha} \cdot \frac{\alpha}{p_{\min}^*} = \widetilde{S_j^{\mathsf{RG}}} \cdot \frac{p_{\min}^{a_1} \cdot p_{\max}^{a_2}}{\mu \cdot p_{\min}^*}$$

$$\le \frac{3}{\mu} \cdot \frac{p_{\min}^{a_1} \cdot p_{\max}^{a_2}}{p_{\min}^*} \cdot \widetilde{S^*} \le \frac{3}{\mu} \cdot \eta \cdot \frac{p_{\min}^{a_1} \cdot p_{\max}^{a_2}}{p_{\min}^*} \cdot S^* \quad (\text{By Lemma C.6 and C.7})$$

$$\le \frac{3}{\mu} \cdot \eta^{1+|a_1|} \cdot \frac{p_{\max}^{a_2}}{(p_{\min}^*)^{1-a_1}} \cdot S^*$$

where the last inequality is due to $p_{\min}^{a_1} \le \eta^{|a_1|} \cdot (p_{\min}^*)^{a_1}$, as $p_{\min}^{a_1} \le (\eta \cdot p_{\min}^*)^{a_1} = \eta^{|a_1|} \cdot (p_{\min}^*)^{a_1}$ if $a_1 \ge 0$ and $p_{\min}^{a_1} = (\frac{1}{p_{\min}})^{-a_1} \le (\frac{\eta}{p_{\min}^*})^{-a_1} = \eta^{|a_1|} \cdot (p_{\min}^*)^{a_1}$ if $a_1 < 0$.

The constant term to appear in the competitive ratio is $\max\{\mu, \frac{3}{\mu}\}$, which is at least $\sqrt{3}$ (the minimum is achieved at $\mu = \sqrt{3}$). Therefore, we choose $\mu = \sqrt{3}$ for an optimal constant term.

As we aim to relate the competitive ratio to $P$, the maximum ratio of any two job sizes, we enforce the power of $p_{\max}$ to equal that of $p_{\min}$ (or $p_{\min}^*$) on the denominator. Thus, we set:

$$a_1 + a_2 = b_1 + b_2 = 1$$

To enforce $\alpha < \beta$, or $\frac{\alpha}{\beta} = \frac{p_{\min}^{a_1-b_1} \cdot p_{\max}^{a_2-b_2}}{\sqrt{3}} = \frac{1}{\sqrt{3}} \cdot (\frac{p_{\min}}{p_{\max}})^{a_1-b_1} < 1$, we set:

$$a_1 \ge b_1$$

To enforce $p_j \le \beta$ for all $1 \le j \le n$, or $p_{\max} \le p_{\min}^{b_1} \cdot p_{\max}^{b_2} = p_{\min}^{b_1} \cdot p_{\max}^{1-b_1} \iff (\frac{p_{\max}}{p_{\min}})^{b_1} \le 1$, we set:

$$b_1 \le 0$$

With these constraints, the results (i.e., Lemmas) used in this section can hold.

We have (with $a_1 + a_2 = b_1 + b_2$ and $a_1 \ge b_1$):

$$S_j^{\mathsf{RG}} \le \sqrt{3} \cdot \eta^2 \cdot \frac{p_{\max}^{b_2-a_2}}{p_{\min}^{a_1-b_1}} \cdot S^* = \sqrt{3} \cdot \eta^2 \cdot (\frac{p_{\max}}{p_{\min}})^{a_1-b_1} \cdot S^* \le \sqrt{3} \cdot \eta^2 \cdot (\frac{\eta \cdot p_{\max}^*}{p_{\min}^*/\eta})^{a_1-b_1} \cdot S^*$$

$$= \sqrt{3} \cdot \eta^{2+2a_1-2b_1} \cdot P^{a_1-b_1} \cdot S^*$$

for $J_j$ being a long job. We have (with $a_2 = 1 - a_1$):

$$S_j^{\mathsf{RG}} \le \sqrt{3} \cdot \eta^{1+|a_1|} \cdot (\frac{p_{\max}}{p_{\min}^*})^{1-a_1} \cdot S^*$$

for $J_j$ being a short job.

Observe that $\mathsf{FIFO}$ is $P$-competitive by Theorem 5.2. We set the power of $P$ and that of $\frac{p_{\max}}{p_{\min}^*}$ (which will relate to $P$) to be at most 1, as $\mathsf{FIFO}$ will otherwise dominate our algorithm. We enforce:

$$a_1 - b_1 \le 1 \Rightarrow a_1 \le 1 + b_1 \le 1$$

and

$$a_1 \ge 0$$

Fix any $0 \le a_1 \le 1$, decreasing $b_1$ increases both the power of $\eta$ and $P$ in term $\sqrt{3} \cdot \eta^{2+2a_1-2b_1} \cdot P^{a_1-b_1} \cdot S^*$. We choose $b_1 = 0$ (and therefore $b_2 = 1$) for the optimal (minimal) competitive ratio.

With $\theta = (a_1, 1 - a_1, 0, 1, \sqrt{3})$, we have:

$$S_j^{\mathsf{RG}} \le \sqrt{3} \cdot \eta^{2+2a_1} \cdot P^{a_1} \cdot S^*$$

for $J_j$ being a long job, and:

$$S_j^{\mathsf{RG}} \le \sqrt{3} \cdot \eta^{1+a_1} \cdot (\frac{p_{\max}}{p_{\min}^*})^{1-a_1} \cdot S^* \le \sqrt{3} \cdot \eta^{1+a_1} \cdot (\frac{\eta \cdot p_{\max}^*}{p_{\min}^*})^{1-a_1} \cdot S^* = \sqrt{3} \cdot \eta^2 \cdot P^{1-a_1} \cdot S^*$$

for $J_j$ being a short job. Therefore, we have $S^{\mathsf{RG}} = \max_{1 \leq j \leq n} S_j^{\mathsf{RG}} \leq \sqrt{3} \cdot \eta^{2+2a_1} \cdot P^{\max\{a_1, 1-a_1\}} \cdot S^*$. For the competitive ratio, $\mathsf{RG}$ with $a_1 = \frac{1}{2} - \Delta$ dominates that with $a_1 = \frac{1}{2} + \Delta$ for any $0 < \Delta \leq \frac{1}{2}$. Therefore, we restrict:

$$0 \leq a_1 \leq \frac{1}{2}$$

The competitive ratio of $\mathsf{RG}$ (with $\theta = (a_1, 1-a_1, 0, 1, \sqrt{3})$ and $0 \leq a_1 \leq \frac{1}{2}$) is $\sqrt{3} \cdot \eta^{2+2a_1} \cdot P^{1-a_1}$. Replacing $a_1$ by $x$ ($0 \leq x \leq \frac{1}{2}$), we construct $\mathsf{RG}^x$ with $\alpha = \frac{p_{\min}^x \cdot p_{\max}^{1-x}}{\sqrt{3}}$ and $\beta = p_{\max}$.

## I    VISUALIZATION OF CONSISTENCY-SMOOTHNESS TRADE-OFFS

The trade-off between consistency and smoothness is visualized in Figure 7.

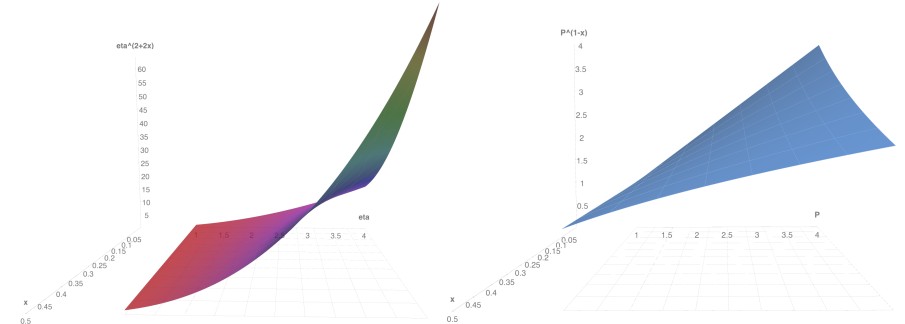

Figure 7: Trade-off between consistency and smoothness. The graphs show how the hyper-parameter $x$ (on the $x$-axis) changes the competitive ratio as a function of $\eta$ (left curve $(x, \eta, \eta^{2+2x})$) and $P$ (right curve $(x, P, P^{1-x})$), i.e., the projection on plane $x = a$ ($0 \leq a \leq \frac{1}{2}$) represents the competitive ratio as a function of $\eta$ and $P$.

## J    MISSING PROOFS FOR SECTION 6

**Theorem J.1** (Theorem 6.1 Restated). *Any algorithm is at best $\frac{\epsilon}{1-\epsilon} \cdot n + 1$-competitive, if given a $(1 - \epsilon)$-speed machine, for any $0 < \epsilon < 1$.*

*Proof for Theorem 6.1.* Construct $n$ jobs: $p_j^* = 1, r_j = j - 1$ for all $1 \leq j \leq n$. Clearly, the optimal schedule with a unit-speed machine has a max-stretch of 1. Run any algorithm with a $(1 - \epsilon)$-speed machine. Let the last completed job be $J_k$. Clearly, we have $C_k \geq \frac{n}{1-\epsilon}$ and $r_k \leq n - 1$. Therefore, the max-stretch achieved is at least $S_k = \frac{C_k - r_k}{1} \geq \frac{n}{1-\epsilon} - (n - 1)$, which is $\frac{\epsilon}{1-\epsilon} \cdot n + 1$ times the optimum. $\square$

We prove Lemma 6.3, which claims the monotonicity of the algorithms mentioned in this paper. We first show that the $\mathsf{RG}$-based algorithms ($\mathsf{RG}$, $\mathsf{ARG}$, $\mathsf{PRG}$, $\mathsf{RG}^x$) are monotonic. Our proof shows that (1) any $\mathsf{RG}$-based algorithm has *time-varying priority execution* for all problem inputs, and (2) any algorithm with time-varying priority execution for all problem inputs is monotonic. We define time-varying priority execution as follows.

**Definition J.2** (Time-varying Priority Execution). Let $A$ be an algorithm and $I$ a problem input. We say $A$ has a time-varying priority execution for $I$ if there exist a partitioning of time $0 = t_0 \leq t_1 \leq t_2 \leq ... \leq t_P$ and a mapping $F$ from time partitions to total orderings of jobs (i.e., $F(t_x, t_{x+1})$ is an order of all jobs) such that $A$ always processes the active job that has the highest priority indicated by $F(t_x, t_{x+1})$ (jobs at the front of $F(t_x, t_{x+1})$ have higher priorities) during $(t_x, t_{x+1})$ for all $0 \leq x \leq P$, where $t_{P+1}$ represents $\infty$.

Here, a problem input differs from a problem instance. A problem input reveals the job size prediction to an algorithm when a job is released, i.e., problem input includes release times and job size predictions. A problem instance is all the data (release times, job sizes, and job size predictions). Thus, a job set can create different problem instances by attaching different job size predictions. A

time-varying priority execution processes jobs in a (time-varying) order that depends solely on the problem input, not the job sizes. This is key to proving monotonicity.

**Lemma J.3.** *Any* RG*-based algorithm has time-varying priority execution for all problem inputs.*

*Proof.* Consider any RG-based algorithm $A$ and any problem input $I$. Suppose $I$ include $n$ jobs $(J_1, ..., J_n)$ where job size prediction for $J_j$ is $p_j$ and release time is $r_j$. Let $y_j = \alpha$ if $p_j \leq \alpha$ and $y_j = \beta$ if $p_j > \alpha$ for all $1 \leq j \leq n$ in the case of static $\alpha$ and $\beta$. In the case of time-varying $\alpha$ and $\beta$ (for ARG), let $y_j = \alpha_{r_j}$ if $p_j \leq \alpha_{r_j}$ and $y_j = \beta_{r_j}$ if $p_j > \alpha_{r_j}$. The relaxed-stretch (or adaptive-relaxed-stretch in ARG) for $J_j$ at the time $t$ can be uniformly written as $\frac{t-r_j}{y_j}$ for any RG-based algorithm. Let $0 = t_0 \leq t_1 \leq t_2 \leq ... \leq t_P$ be the sorted elements in set $\{\frac{y_i \cdot r_j - y_j \cdot r_i}{y_i - y_j} \mid 1 \leq i < j \leq n, y_i \neq y_j, \frac{y_i \cdot r_j - y_j \cdot r_i}{y_i - y_j} \geq 0\} \cup \{0\}$, where every element corresponds to the solution to equation $\frac{t-r_i}{y_i} = \frac{t-r_j}{y_j}$ with variable $t$ for some $i, j$. Recall that the algorithm runs the active job $J_j$ with the largest $\frac{t-r_j}{y_j}$ at any time $t$ (with tie-breaking by smaller job index first). Thus, to conclude a time-varying priority execution, we only need to show that the relative priority between jobs (in terms of $\frac{t-r_j}{y_j}$) for the entire job set remains the same in every interval $(t_x, t_{x+1})$. Suppose, for proof by contradiction, that there are two jobs $J_a$ and $J_b$ with $\frac{t'-r_a}{y_a} > \frac{t'-r_b}{y_b}$ and $\frac{t''-r_a}{y_a} < \frac{t''-r_b}{y_b}$ with $t' < t''$ and $t', t'' \in (t_x, t_{x+1})$ for some $x$. Then, we have:

$$t_x < t' < \frac{y_a \cdot r_b - y_b \cdot r_a}{y_a - y_b} < t'' < t_{x+1}$$

which contradicts the construction of the time partitioning $0 = t_0 \leq t_1 \leq t_2 \leq ... \leq t_P$. $\square$

**Lemma J.4.** *Any algorithm with time-varying priority execution for all problem inputs is monotonic.*

*Proof.* Consider any algorithm $A$ with time-varying priority execution for all problem inputs, any two instances with identical input $I$ and actual job sizes $(p_1^*, ..., p_n^*)$ and $(p_1^{*'}, ..., p_n^{*'})$ such that $p_j^* \leq p_j^{*'}$ for all $j$. Let $S^{(1)}$ and $S^{(2)}$ be the max-stretch obtained by $A$ on these two instances, and we denote the schedules by $\sigma_1$ and $\sigma_2$. To show $S^{(1)} \leq S^{(2)}$, we show $C_j^{(1)} \leq C_j^{(2)}$ for all $1 \leq j \leq n$, where $C_j^{(1)}$ and $C_j^{(2)}$ denote the completion time for job $J_j$ in the two schedules. Let $p_j^{(1)}(t)$ and $p_j^{(2)}(t)$ denote the remaining job size of job $J_j$ at time $t$ in the two schedules, i.e., $p_j^*$ minus the total amount of processing allocated to job $J_j$ up to time $t$. To show $C_j^{(1)} \leq C_j^{(2)}$, we show that algorithm $A$ maintains $p_j^{(1)}(t) \leq p_j^{(2)}(t)$ for all $1 \leq j \leq n$, $t \geq 0$. Let $0 = t_0 \leq t_1 \leq ... \leq t_P$ be the time partition (for algorithm $A$ running against $I$) with a mapping $F$ such that $A$ always processes the active job that has the highest priority indicated by $F(t_x, t_{x+1})$.

We show $p_j^{(1)}(t) \leq p_j^{(2)}(t)$ for all $1 \leq j \leq n$, $t < t_x$, $x \geq 1$ via induction on $x$ (we slightly abuse the notation to let $t_{P+1}$ represent $\infty$). The base case is to consider the interval $[t_0, t_1)$. Initially, $p_j^{(1)}(0) = p_j^* \leq p_j^{*'} = p_j^{(2)}(0)$. Fix any time $t \in (t_0, t_1)$. The ordering $F(t_0, t_1)$ defines the job priority in $(t_0, t_1)$. Schedules $\sigma_1$ and $\sigma_2$ are processing (either) the same or different jobs at time $t$. If they are processing the same job $J_s$, it follows $p_j^{(1)}(t) = 0 = p_j^{(2)}(t)$ for every job $J_j$ with a priority higher than $J_s$; $p_j^{(1)}(t) = p_j^{(1)}(0) \leq p_j^{(2)}(0) = p_j^{(2)}(t)$ for every job $J_j$ with a priority lower than $J_s$. It also follows $p_s^{(1)}(t) \leq p_s^{(2)}(t)$, as $p_j^* \leq p_j^{*'}$ for every job $J_j$ with a higher priority than $J_s$. If they are processing different jobs $J_a$ (in $\sigma_1$) and $J_b$ (in $\sigma_2$), it follows that $J_a$ has a lower priority than $J_b$ in $F(t_0, t_1)$ and $J_b$ has been completed in $\sigma_1$ at time $t$. Therefore, any job $J_j$ with a priority higher than $J_a$ has been completed in $\sigma_1$, i.e., $p_j^{(1)}(t) = 0 \leq p_j^{(2)}(t)$; any job $J_j$ with a priority no higher than $J_a$ has not been processed in $\sigma_2$, i.e., $p_j^{(1)}(t) \leq p_j^* \leq p_j^{*'} = p_j^{(2)}(t)$. The arguments hold for every $t \in (t_0, t_1)$. For the inductive step, suppose $p_j^{(1)}(t) \leq p_j^{(2)}(t)$ for all $1 \leq j \leq n$, $t < t_k$ for some $k \geq 1$. Consider the interval $[t_k, t_{k+1})$. By inductive hypothesis, we have $p_j^{(1)}(t_k) \leq p_j^{(2)}(t_k)$ for all $1 \leq j \leq n$. The inductive step uses a generalized version of the arguments to show the base case. Fix any time $t \in (t_k, t_{k+1})$. The ordering $F(t_k, t_{k+1})$ defines the job priority in $(t_k, t_{k+1})$. If $\sigma_1$ and $\sigma_2$ are processing the same job $J_s$, it follows $p_j^{(1)}(t) = 0 = p_j^{(2)}(t)$ for every job $J_j$ with a

priority higher than $J_s$; $p_j^{(1)}(t) = p_j^{(1)}(t_k) \leq p_j^{(2)}(t_k) = p_j^{(2)}(t)$ for every job $J_j$ with a priority lower than $J_s$; $p_s^{(1)}(t) \leq p_s^{(2)}(t)$, as $p_j^{(1)}(t_k) \leq p_j^{(2)}(t_k)$ for every job $J_j$ with a higher priority than $J_s$. If they are processing different jobs $J_a$ (in $\sigma_1$) and $J_b$ (in $\sigma_2$), it follows that $p_j^{(1)}(t) = 0 \leq p_j^{(2)}(t)$ for any job $J_j$ with a priority higher than $J_a$ and $p_j^{(1)}(t) \leq p_j^{(1)}(t_k) \leq p_j^{(2)}(t_k) = p_j^{(2)}(t)$ for any job $J_j$ with a priority no higher than $J_a$. $\qquad \square$

**Lemma J.5** (Lemma 6.3 Restated). *All algorithms mentioned in this paper (RG, ARG, PRG, RG$^x$, and RR) are monotonic.*

*Proof for Lemma 6.3.* Any RG-based algorithm (RG, ARG, PRG, and RG$^x$) is monotonic by Lemmas J.3 and J.4. It is trivial to see that RR is monotonic. $\qquad \square$

**Theorem J.6** (Theorem 6.4 Restated). *Given a monotonic algorithm $A$ with competitive ratio $c_A$, one can obtain a $(1+\epsilon)$-speed $\min\{c_A, \frac{n}{\epsilon}\}$-competitive (RR-augmented) algorithm.*

*Proof for Theorem 6.4.* With a $(1+\epsilon)$-speed machine, we allocate one-unit speed to algorithm $A$ and $\epsilon$-unit speed to RR. That is, for every time unit, both algorithms run in parallel: algorithm $A$ is run at a rate of $\frac{1}{1+\epsilon}$, and RR is run at a rate of $\frac{\epsilon}{1+\epsilon}$. Let the completion time for $J_j$ be $C_j$ in this schedule.

When running $A$ standalone on a unit-speed machine, the completion time $C_j^A$ satisfies $\max_{1 \leq j \leq n} \frac{C_j^A - r_j}{p_j^*} \leq c_A \cdot S^*$, where $S^*$ is the optimal max-stretch, and $c_A$ is the competitive ratio of algorithm $A$. Similarly, for RR running standalone with $\epsilon$-speed, we have $\max_{1 \leq j \leq n} \frac{C_j^R - r_j}{p_j^*} \leq \frac{n}{\epsilon} \cdot S^*$, where $C_j^R$ is the completion time in this schedule.

Due to the monotonicity of algorithms $A$ and RR, we have:

$$\max_{1 \leq j \leq n} \frac{C_j - r_j}{p_j^*} \leq \max_{1 \leq j \leq n} \frac{\min\{C_j^A, C_j^R\} - r_j}{p_j^*} \leq \max_{1 \leq j \leq n} \min\{c_A \cdot S^*, \frac{n}{\epsilon} \cdot S^*\} = \min\{c_A, \frac{n}{\epsilon}\} \cdot S^*$$

Thus, the combined algorithm achieves a competitive ratio of $\min\{c_A, \frac{n}{\epsilon}\}$. $\qquad \square$

## J.1 Optimality of RG$^+$

The proof of Theorem B.1 can be extended to show that "no deterministic algorithm can achieve a competitive ratio better than $\frac{n}{1+\epsilon}$ for online non-clairvoyant max-stretch scheduling with a machine of speed $(1+\epsilon)$, even if all jobs are released at time 0." The idea of the extension is as follows.

Consider $n$ identical jobs all with job size 1 to be released at time 0. Run any algorithm $A$ against this set of jobs. By time 1, let $J_1$ be the job with the least amount of the processed size $l_1$. We have $l_1 \leq \frac{1+\epsilon}{n}$ by the pigeonhole principle. We can then construct an instance with $n$ jobs, all released at time 0, where the job sizes are defined as $p_1^* = \frac{1+\epsilon}{n} + \lambda$ and the rest geometrically increasing sizes, i.e., $p_2^* = \psi - p_1^*, p_3^* = \psi^2 - \psi, p_4^* = \psi^3 - \psi^2, ..., p_n^* = \psi^{n-1} - \psi^{n-2}$, where $\psi$ is a large constant and $\lambda$ a small constant. In this setup, the optimal max-stretch using a unit-speed machine approaches 1, while the max-stretch achieved by algorithm $A$ with a faster machine is at least approximately $\frac{n}{1+\epsilon}$. This establishes a lower bound for the $(1+\epsilon)$-speed competitive ratio as $\frac{n}{1+\epsilon}$. Therefore, algorithm RG$^+$ achieves an asymptotically optimal robustness under speed augmentation.

## K Job Size Distributions in Experiments

We aim to replicate real-world data size distributions in our synthetic datasets. Figure 8 shows the job size distributions in the real-world datasets (Google Google (2019), Alibaba Alibaba (2023), and Azure Cortez et al. (2017)). Job sizes generally follow an exponential distribution, except for Google's data, which has more randomness in the right tail. With the maximum job size ratio $P$, we set $p_j^*$ to $\max\{1, -\log X \cdot \mathcal{A}\}$ where $X$ is a random variable with $X \sim \mathcal{U}(0,1)$ and $\mathcal{A}$ a scaling

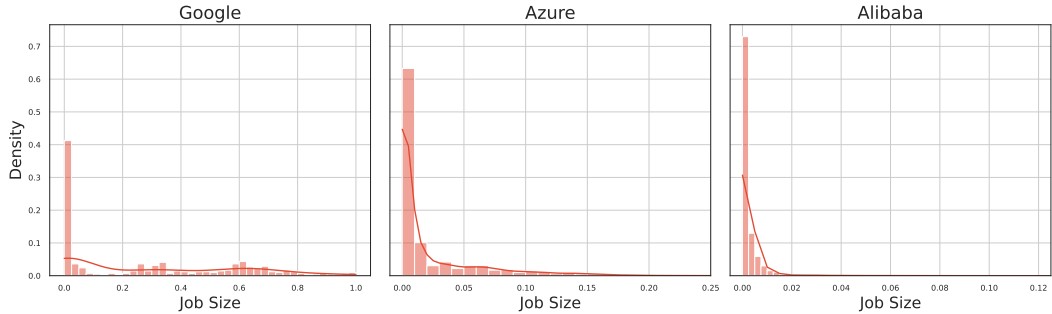

Figure 8: Real-world Job Size Distribution.

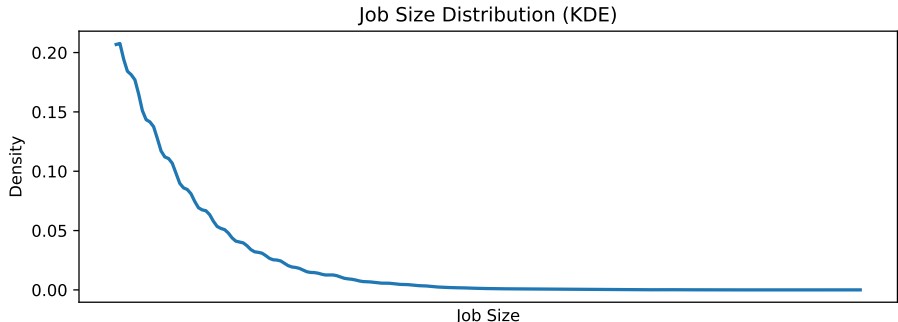

Figure 9: Synthetic Job Size Distribution

factor ensuring $p_j^* \leq P$. Figure 9 shows the Kernel Density Estimation (KDE) of the job sizes in the synthetic datasets, showing that the synthetic job sizes follow an exponential distribution aligned with the real-world applications.

## L EXPERIMENTS FOR VARIANCE OF STRETCH

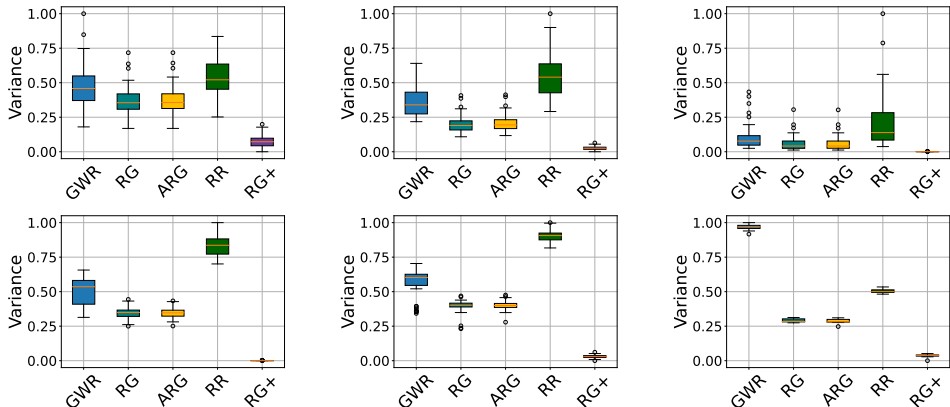

Figure 10: The variance of stretch with increasing jobs. Each box represents the distribution of the normalized variance scores. The number of jobs is set to 1000 (Row 1 Column 1), 2000 (R1C2), 3000 (R1C3), 4500 (R2C1), 6000 (R2C2), and 10000 (R2C3). Jobs are released in time $[0, 5000]$ uniformly at random. The job sizes are drawn from $[1, 10]$. The prediction error is set to 1 (i.e., $p_j = p_j^*$).

This section presents the experimental results for GWR, RR, RG, ARG, and RG$^+$ in minimizing the *variance of stretch*, defined as:

$$\text{variance of stretch} = \frac{1}{n} \sum_{j=1}^{n} (S_j - \overline{S})^2$$

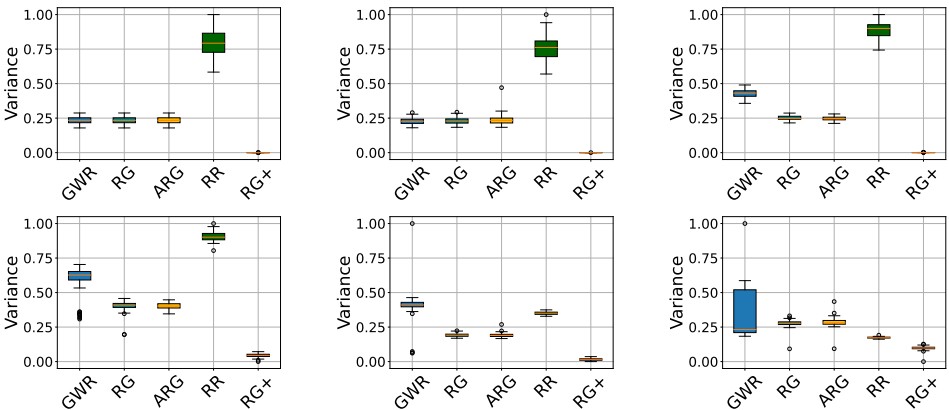

Figure 11: The variance of stretch with an increasing $P$. The job sizes are drawn from interval $[1, P]$, where $P$ is set to 1 (R1C1), 2 (R1C2), 5 (R1C3), 10 (R2C1), 20 (R2C2), and 40 (R2C3). The number of jobs is fixed at 4500. Jobs are released in time $[0, 5000]$ uniformly at random. The prediction error is set to 1.

where $\overline{S} = \frac{1}{n}\sum_{j=1}^{n} S_j$. The variance of stretch measures fairness from the view of jobs as if they are compared to each other. We run the experiments on the synthetic datasets under the same settings as Section 7. In presenting the results, we apply a min-max normalization to the variance scores, i.e., dividing the variance by the difference between the maximum and minimum variance of all problem instances under each setting.

Figures 10 and Figure 11 present the normalized variance score under increasing $n$ and $P$ with perfect predictions. The performance ordering of these algorithms generally aligns with that in the max-stretch minimization: a lower max-stretch performance ratio corresponds to a lower variance score. $RG^{+}$ shows its superior performance under all the settings. This suggests that $RG^{+}$ ensures fairness in resource allocation measured in the stretch space. In contrast, RR suffers the consistent worst variance scores. This worsened performance is the cost of ignoring the job size information in fair scheduling.

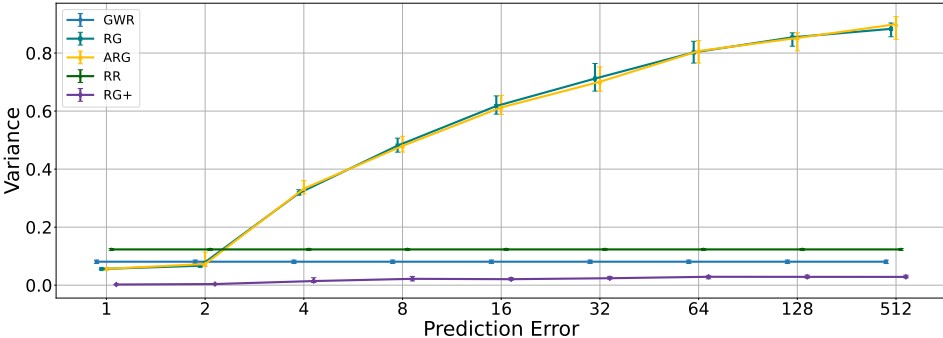

Figure 12: The variance of stretch with an increasing prediction error $\eta$. The prediction error increases from 1 to 512. The number of jobs is fixed at 4500. Jobs are released in time $[0, 5000]$ uniformly at random. The job sizes are drawn from $[1, 10]$.

Figure 12 presents the results under an increasing $\eta$. The variance scores of GWR and RR are constants, as their executions are irrelevant to job size predictions. The variance score of RG and ARG increases with an increasing prediction error, showing a weakening fairness guarantee with worsening predictions. $RG^{+}$ demonstrates its robustness in minimizing the variance of stretch under imperfect predictions: the variance score increases slowly and is capped as $\eta$ increases. Overall, the distinction in the variance scores is evident: $RG^{+}$ outperforms the rest, followed by RG and ARG, and then GWR, with RR the worst.

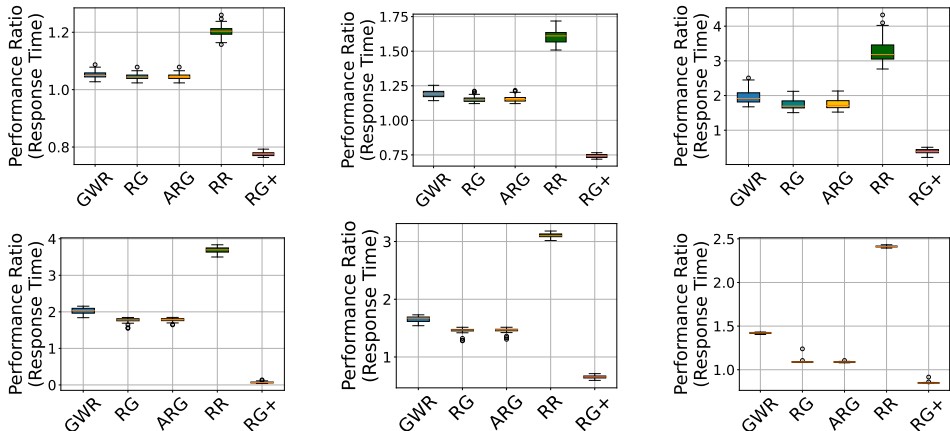

Figure 13: Performance ratios for the mean response time with increasing jobs. Each box represents the distribution of the performance ratios. The number of jobs is set to 1000 (Row 1 Column 1), 2000 (R1C2), 3000 (R1C3), 4500 (R2C1), 6000 (R2C2), and 10000 (R2C3). Jobs are released in time $[0, 5000]$ uniformly at random. The job sizes are drawn from $[1, 10]$. The prediction error is set to 1 (i.e., $p_j = p_j^*$).

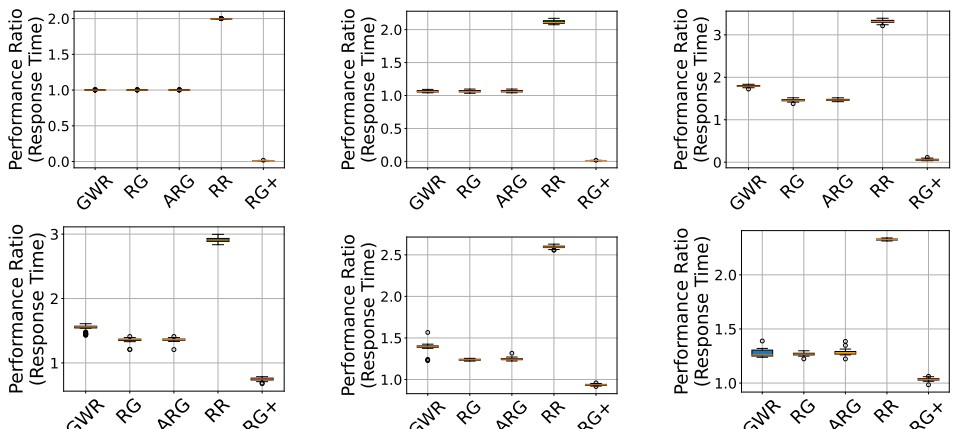

Figure 14: Performance ratios for the mean response time with an increasing $P$. The job sizes are drawn from interval $[1, P]$, where $P$ is set to 1 (R1C1), 2 (R1C2), 5 (R1C3), 10 (R2C1), 20 (R2C2), and 40 (R2C3). The number of jobs is fixed at 4500. Jobs are released in time $[0, 5000]$ uniformly at random. The prediction error is set to 1.

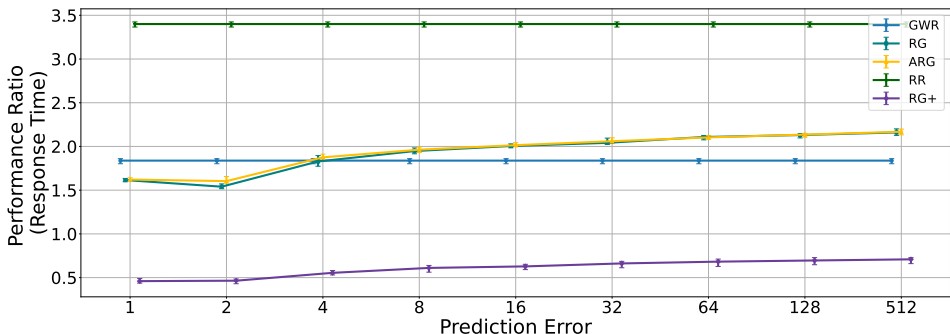

Figure 15: Performance ratios for the mean response time with an increasing prediction error $\eta$. The prediction error increases from 1 to 512. The number of jobs is fixed at 4500. Jobs are released in time $[0, 5000]$ uniformly at random. The job sizes are drawn from $[1, 10]$.

## M  EXPERIMENTS FOR RESPONSE TIME

This section presents the experimental results for GWR, RR, RG, ARG, and RG$^+$ in minimizing the *mean response time*, defined as:

$$\text{mean response time} = \frac{1}{n} \sum_{j=1}^{n} C_j - r_j$$

The mean response time is used to measure the efficiency of scheduling. A lower response time indicates lower waiting time and an improved quality of service. While our proposed algorithms ensure fairness of scheduling, we are interested in whether this comes with a cost in efficiency. We compare the mean response time of our algorithms with that of GWR and RR. We measure the performance ratio for the response time, defined as the ratio between the mean response time obtained by an algorithm and the optimal value given a problem instance. Shortest Remaining Processing Time First (SRPT) is used to compute the optimal mean response time Baker (1974). We run the experiments on the synthetic datasets under the same settings as Section 7.

Figures 13 and 14 present the performance ratio for the mean response time under increasing $n$ and $P$ with perfect predictions. RR has the worst performance, as active jobs mutually delay each other when they constantly share the resource equally. The performance ordering of the other algorithms aligns with that in the max-stretch minimization. It suggests that reducing the max-stretch does not come with a high cost in response time. While RG achieves outstanding performance in minimizing response time, its augmented variant, RG$^+$, emerges as the champion again.

Figure 15 presents the results under an increasing $\eta$. The performance ratios of RG and ARG increase marginally with an increasing prediction error, showing their robustness in minimizing response time with imperfect predictions. In addition, the performance ratio stays close to that of GWR, even with badly wrong predictions. Though the performance ratio increases as that of RG, RG$^+$ shows its superior performance by its winning performance ratio and the robustness to the prediction error. Overall, RG$^+$ performs well in minimizing response time while having a strong guarantee for fairness.

## N  EXPERIMENTS ON REAL-WORLD DATASETS

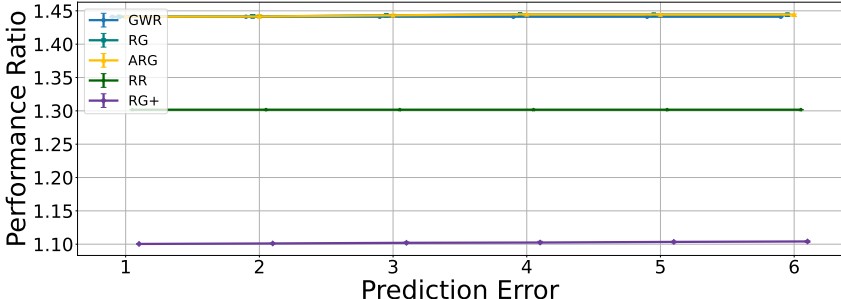

Figure 16: Performance ratios with an increasing prediction error on real-world trace-log data from Google Cloud. The prediction error increases from 1 to 6.

Figure 16, 18, and 17 presents the results on the real-world datasets. The data are server trace logs collected from Google, Azure, and Alibaba Cloud. Each dataset contains the arrival time and job size of the jobs submitted to the cloud. We generate job size predictions with a prediction error from 1 to 6: with the prediction error $\eta$ and job size $p_j^*$, we set $p_j \leftarrow p_j^* \cdot \exp(Y)$ with $Y \sim \mathcal{U}(-\log \eta, \log \eta)$. On Azure and Alibaba datasets, RG and ARG, under a low prediction error, are comparable to GWR, with RR the worst. The performance of RG and ARG gradually becomes worse than RR as the prediction error increases. RG$^+$ consistently outperforms the others. The overall performance of all algorithms on Alibaba and Auzre datasets aligns with the results on the synthetic datasets. On the Google dataset, the performance ratios are not sensitive to the prediction error due to the distribution of job size being closer to uniform than exponential. RG$^+$ consistently achieves the best performance. This time, however, RR outperforms RG, ARG, and GWR due to a high maximum job size ratio $P$

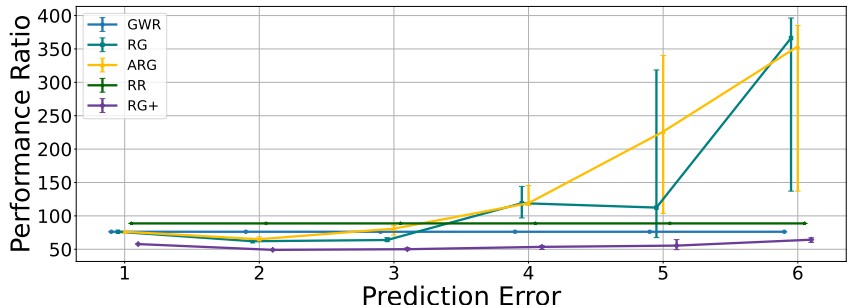

Figure 17: Performance ratios with an increasing prediction error on real-world trace-log data from Azure Cloud. The prediction error increases from 1 to 6.

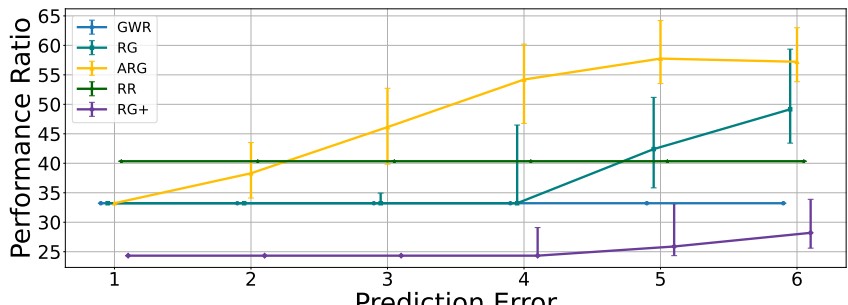

Figure 18: Performance ratios with an increasing prediction error on real-world trace-log data from Alibaba Cloud. The prediction error increases from 1 to 6.

in the Google dataset (compared to the others). RR's performance does not worsen with an increasing $P$, but the others do.

## O    ADDITIONAL EXPERIMENTS WITH A UNIT-SPEED RG$^+$

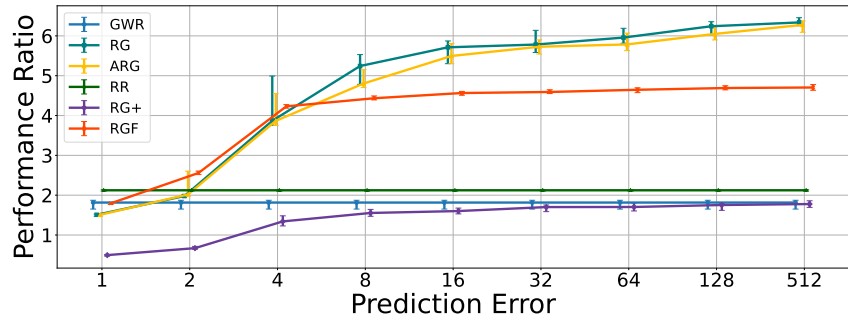

Figure 19: Performance ratios for the max-stretch with an increasing prediction error $\eta$. The prediction error increases from 1 to 512. The number of jobs is fixed at 4500. Jobs are released in time $[0, 5000]$ uniformly at random. The job sizes are drawn from $[1, 10]$.

We performed additional experiments where we set the speed of RG$^+$ to be the same as the others. Figures 19, 20, and 21 presents the results for max-stretch, variance of stretch, and response time. The label RGF denotes the RG$^+$ running with a unit-speed machine, where $0.3$-speed is allocated to RR and the rest $0.7$-speed to RG.

The performance of RGF is not as good as RG$^+$ due to the slower machine, but the performance generally stays bounded in between RG and RR for all metrics. This is as expected. The intuition is that when RG performs well on an instance, some portion of the processing is shared to RR, thus decreasing the overall performance; when RR performs well, again, some portion is shared to RG,

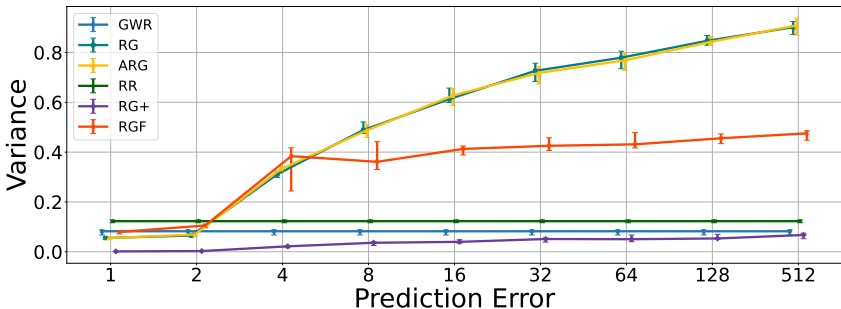

Figure 20: The variance of stretch with an increasing prediction error $\eta$. The prediction error increases from 1 to 512. The number of jobs is fixed at 4500. Jobs are released in time $[0, 5000]$ uniformly at random. The job sizes are drawn from $[1, 10]$.

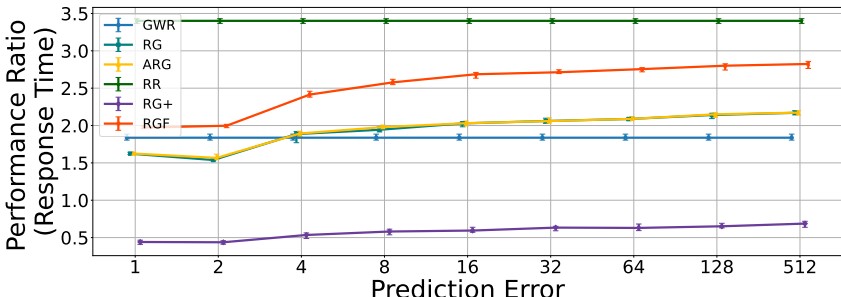

Figure 21: Performance ratios for the mean response time with an increasing prediction error $\eta$. The prediction error increases from 1 to 512. The number of jobs is fixed at 4500. Jobs are released in time $[0, 5000]$ uniformly at random. The job sizes are drawn from $[1, 10]$.

thus decreasing the overall performance. In return, however, the robustness is persevered, making the performance relatively stable under varying prediction errors.

