# OpenReview forum: "Competitive Fair Scheduling with Predictions"
_ICLR.cc/2025/Conference — ICLR 2025 Poster_

### Official Review · Reviewer_KufQ · 2024-10-21

**Soundness:** 3
**Presentation:** 2
**Contribution:** 3
**Rating:** 6
**Confidence:** 4

**Summary:**

The paper studies the scheduling problem of preemptively minimizing the maximum stretch on a single machine. It assumes that both job arrivals and processing times are uncertain.  This problem can be interpreted from the perspective of fairness, as an algorithm needs to give every job at least a k-fraction of the processor on average to achieve a max-stretch of at most k.

For this online problem strong lower bounds (n = number of jobs) on the competitive ratio exist, which is the worst-case ratio between an algorithm's max-stretch and an optimum's max-stretch. Thus, the authors apply the learning-augmented framework to this online problem. That is, they assume that predictions on the unknown processing times are given to an algorithm when a job arrives, and present bounds on the competitive ratio depending on the quality of these predictions. Here, the quality \eta is measured by the worst case ration between predicted and actual processing times. In the clairvoyant online setting, where an algorithm is oblivious to future job releases but jobs arrive with known processing times, a O(\sqrt(P)) competitive algorithm is known, which is asymptotically almost best-possible. Learning-augmented algorithms are a recently popular technique for beyond worst case analysis, and have become an established subfield in the intersection of algorithm theory and machine learning.

The main results of the present paper can be summarized as follows:
- An algorithm (RELAXED-GREEDY) that uses predictions on the processing times but assumes to know the maximum and minimum prediction at time 0 with a competitive ratio of at most O(\eta^3 \sqrt{P}). That is, if the prediction error is small, the algorithm achieve a performance close that of the best-known algorithm for the clairvoyant setting.
- Variants of the previous algorithms that do not need the assumption of knowing the maximum and minimum prediction value in advance, but instead receive a prediction on these values. The performance guarantees then depend also on the quality of the predictions on these aggregated values.
- A further variation of the previous algorithm in which the user can control the trade-off between the consistency bound (the bound assuming the predictions are correct) and the error-dependent bound using a hyperparameter.
- A general method to additionally achieve a robustness in O(n) using a processor that is (1+\eps) times faster than the optimum's machine (speed augmentation)
- An empirical evaluation of the proposed algorithms.

**Strengths:**

- The paper seems to be the first to consider max-stretch in the non-clairvoyant setting. The results are a nice package and a solid contribution to the area of learning-augmented algorithms. Thus, it generally also fits well to ICLR.
- The paper addresses the crucial assumptions of knowing the minimum and maximum processing time in advance, which was made by previous works. They show that relaxing these assumptions to prediction information only loses small factors in the competitive ratio.
- Empirical experiments indicate that their algorithms perform well on real-world data sets.

**Weaknesses:**

There are two questions that may influence the set of results, and thus, the overall contribution. Thus, I currently value them slightly negatively, but I am open to change this depending on the rebuttal.

Besides that, I think the main weakness is the presentation. I have the feeling that the abstract and the initial parts of the introduction are filled with too many technical details, and are not approachable for the broader ML community. In particular, I am missing a better introduction and highlighting of learning-augmented algorithms and a stronger motivation and connection to machine learning (as ICLR is an ML conference). I think a slightly more informal introduction could strongly increase the readability.
Some other examples where I got confused:
- The paragraph on "prediction errors" also contains the definition of the prediction model, and moreover distinguishes between several different models, which are hard to grasp at this point in the writeup.
- Line 87: You mention "preferential round robin", but it is unclear what that is or in which reference one can find it.
- Line 89: In point 4 of the list of contributions, you only mention that you conduct experiments, but not outline any interesting findings.
- Line 94: Why is this assumption unrealistic? In practice?
- Line 397: "This shows the necessity of resource augmentation". This is a strong claim, and I am not sure if it is correct. Just because some technique does not work, this does not imply that one immediately needs strong assumptions such as speed augmentation.

**Questions:**

- In line 86 you claim that your algorithm achieves "optimal robustness of O(n) with (1+\eps)-speed", but I could not find a proof for this claim or a reference to related work. In particular, Theorem 3.5 only considers a lower bound regarding a unit speed processor. It is not obvious to me why this should still hold under speed augmentation, specifically because for other objectives such as total flow time, speed augmentation suffices to achieve a constant competitive ratio [Kalyanasundaram and Pruhs 2000]. I think this is an important question to address, because if a constant competitive ratio is possible under speed augmentation, in my opinion, the result of Theorem 6.4 is way less interesting.
- Line 418-423: It is unclear in the main part why monotonicity is important. I figured out that it is used in Line 1637 for the proof of Theorem J.6. I do not understand why this property is even necessary for the proof. The proof of Theorem J.6 is not very detailed, so it is hard to understand why monotonicity is important. I think it is not necessary to obtain this result: You run your main algorithm on the speed-1 machine and RR on the speed-\eps machine, but both work on copies of the actual instance. Whenever one of the algorithms finishes a job which is still running in the other algorithm, the actual job can complete in the actual schedule. However, we can still make the other algorithm continue working on the job because it works on its independent copy, and the actual processing time is known now. Thus, the overall bound is still the minimum of the two bound of the algorithms, and no algorithm gets "distracted" by a completion of a job in the other algorithm.


Additional remarks:
- The idea of a distortion prediction error has first been considered by Azar, Leonardi and Touitou (STOC 21). I would at least add this reference when the error is defined in line 48.
- Line 60: I think the definition of smoothness and robustness is equivalent. I think smoothness should be explained differently.
- Personally, I think that stating bounds in the related work section is not helpful. The considered problems are very different and the variables are not explained. Instead, I would prefer to have more references to related work on algorithms (scheduling) with predictions.
- If you keep the results on the offline problem, I would at least mention in the results section. Otherwise, this comes quite unexpected in the paper, and it is not clear (in the beginning of section 3) what the benefit of this result in the context of the results mentioned earlier is.
- Line 207: the maximum is not defined properly

---

> ### Author Response · Authors · 2024-11-16
>
> We appreciate the reviewer's constructive comments, insightful questions, and positive feedback regarding our work. We are grateful for the opportunity to clarify the points raised. Below, we provide detailed responses to your questions. *We will address additional concerns/weaknesses regarding the presentation in separate threads with a revised manuscript.*
>
> ### **Q1. Speed augmentation**
>
> We appreciate the reviewer's question regarding the claimed optimal competitive ratio under speed augmentation. To provide a comprehensive response, we will first clarify the model we used and then discuss the optimality of our results.
>
> **Clarification of the model**
>
> In our paper, the competitive ratio under speed augmentation is defined as the worst-case ratio between the objective value achieved by an algorithm with a faster machine and that of an optimal algorithm using a unit-speed machine. This is stated on Line 65 of our paper and follows the standard definition used by Kalyanasundaram and Pruhs (2000) in the work, "Speed is as powerful as clairvoyance."
>
> Regarding your example with total flow time, it is true that speed augmentation can result in a constant competitive ratio if the algorithm is equipped with a faster machine while the optimal algorithm remains restricted to unit speed. The key point is that speed augmentation benefits only the algorithm to analyze. If both the algorithm and the optimal algorithm were equally sped up, the (worst-case) relative performance would remain unchanged, as the problem instance could simply be rescaled by shrinking release times or sizing up job sizes proportionally.
>
> In our case, we show that $\textsf{RG}^+$ is $O(n)$-competitive with a faster machine, which aligns with the asymptotically optimal competitive ratio for non-clairvoyant scheduling (as established in Theorem 3.5). This is analogous to achieving a constant competitive ratio for flow time scheduling under speed augmentation, demonstrating that our result has similar theoretical merit. In the following response, we elaborate on the optimality of this result.
>
> **Optimality**
>
> While not explicitly detailed in our paper, the proof of Theorem 3.5 (see Appendix, Theorem B.1) can be extended to show that "no deterministic algorithm can achieve a competitive ratio better than $\frac{n}{1 + \epsilon}$ for online non-clairvoyant max-stretch scheduling with a machine of speed $(1 + \epsilon)$, even if all jobs are released at time 0." The intuition behind this extension is as follows.
>
> Consider $n$ identical jobs all with job size $1$ to be released at time $0$. Run any algorithm $A$ against this set of jobs. By time $1$, let $J_1$ be the job with the least amount of the processed size $l_1$. We have $l_1 \leq \frac{1 + \epsilon}{n}$ by the pigeonhole principle. We can then construct an instance with $n$ jobs, all released at time 0, where the job sizes are defined as $p_1^* = \frac{1 + \epsilon}{n} + \lambda$ and the rest geometrically increasing sizes, i.e., $p^*_2 = \psi - p_1^*$, $p^*_3 = \psi^2 - \psi$, $p^*_4 = \psi^3 - \psi^2$, ..., $p^*_n = \psi^{n-1} - \psi^{n-2}$, where $\psi$ is a large constant and $\lambda$ a small constant. In this setup, the optimal max-stretch using a unit-speed machine approaches 1, while the max-stretch achieved by algorithm $A$ with a faster machine is at least approximately $\frac{n}{1 + \epsilon}$. This establishes a lower bound for the $(1 + \epsilon)$-speed competitive ratio as $\frac{n}{1 + \epsilon}$. For a detailed formalization, please refer to the proof of Theorem B.1 and observe how this extension relates to the original argument.
>
> **Conclusion**
>
> In summary, our analysis demonstrates that the asymptotically optimal competitive ratio under $(1 + \epsilon)$-speed augmentation is $O(n)$, which is achieved by our proposed $\textsf{RG}^+$ algorithm. We hope this clarification addresses your question and illustrates the optimality of our approach.

---

> ### Author Response · Authors · 2024-11-16
>
> ### **Q2. Monotonicity**
>
> We appreciate the reviewer's observation regarding the role of monotonicity in the proof of Theorem 6.4 (Theorem J.6 in the appendix). We acknowledge that this may not have been clearly explained in our original submission, and we will revise the paper to include a more detailed proof. Below, we outline the argument in greater detail.
>
> With a $(1 + \epsilon)$-speed machine, we allocate one-unit speed to algorithm $A$ and $\epsilon$-unit speed to RR. That is, for every time unit, both algorithms run in parallel: algorithm $A$ is run at a rate of $\frac{1}{1 + \epsilon}$, and RR is run at a rate of $\frac{\epsilon}{1 + \epsilon}$. Let the completion time for $J_j$ be $C_j$ in this schedule.
>
> When running $A$ standalone on a unit-speed machine, the completion time $C_j^A$ satisfies $\max_{1 \leq j \leq n}{\frac{C_j^A - r_j}{p_j^*}} \leq c_A \cdot S^*$, where $S^*$ is the optimal max-stretch, and $c_A$ is the competitive ratio of algorithm $A$. Similarly, for $\textsf{RR}$ running standalone with $\epsilon$-speed, we have $\max_{1 \leq j \leq n}{\frac{C_j^R - r_j}{p_j^*}} \leq \frac{n}{\epsilon} \cdot S^*$, where $C_j^R$ is the completion time in this schedule.
>
> Due to the monotonicity of algorithms $A$ and $\textsf{RR}$, we have:
> $$ \max_{1 \leq j \leq n}{\frac{C_j - r_j}{p_j^*}} \leq \max_{1 \leq j \leq n}{\frac{\min \\{ C_j^A, C_j^R \\} - r_j}{p_j^*}} \leq \max_{1 \leq j \leq n}{\min \\{ c_A \cdot S^*, \frac{n}{\epsilon} \cdot S^* \\}} = \min \\{ c_A, \frac{n}{\epsilon} \\} \cdot S^*$$
>
> Thus, the combined algorithm achieves a competitive ratio of $\min \\{ c_A, \frac{n}{\epsilon} \\}$.
>
> While we acknowledge the reviewer's suggested "redundancy argument" is valid, the monotonicity property is still crucial in practice. It ensures that when two algorithms collaborate, the combined schedule will not degrade the max-stretch objective.
>
> To illustrate why monotonicity is important, consider a scenario with three jobs $J_1, J_2, J_3$ (available at time $0$) with job size $2, 2, 10$. Suppose algorithm $A_1$ has a strategy that starts with running $J_1$ and runs $J_3$ instead of $J_2$ if $J_1$ finishes strictly before time $2$. Otherwise, it proceeds with $J_2$ next. Without any external help, this leads to the schedule $\sigma_1$, which processes $J_1, J_2, J_3$ sequentially, achieving a max-stretch of 2. Suppose another algorithm $A_2$ assists by running $J_1$ with a separate unit-speed machine at time $0$ until $J_1$ is finished (when $J_1$ finishes, $A_2$ does not run any other job). This earlier completion causes algorithm $A_1$ to run $J_3$ at time $1$ when $J_1$ completes, resulting in schedule $\sigma_2$ where $J_3$ completes at time 11 and $J_2$ at time $13$, leading to a max-stretch of 6.5. In this case, the collaboration worsens the max-stretch compared to the standalone schedule $\sigma_1$.
>
> Monotonicity ensures that such counterintuitive outcomes do not occur, as it guarantees that additional resources or assistance can only improve (or at least not degrade) the objective value. Thus, monotonicity is essential to ensure the robustness of scheduling algorithms when they are combined or supplemented with additional resources.
>
> We appreciate the reviewer's insightful comments and will incorporate these clarifications into the revised manuscript to strengthen the exposition.

---

> ### Author Response · Authors · 2024-11-24
> **Responses to weaknesses**
>
> **Revision of the abstract and introduction**
>
> We appreciate the reviewer’s comments on the presentation. We have revised the abstract and the initial parts of the introduction to make the paper more approachable for the broader ML community.
>
> We added the following text as the first sentence of the abstract:
>
> *Beyond the worst-case analysis of algorithms, the recent learning-augmented framework considers that an algorithm can leverage possibly imperfect predictions about the unknown variables to make decisions to have guarantees tied to the prediction quality.*
>
> We also added the following as the first paragraph of the introduction:
>
> *In traditional online scheduling, algorithms are designed to have worst-case guarantees under incomplete information. However, this conservative approach often results in suboptimal performance, especially compared to where full information is available. Given the rapid advancements in Machine Learning (ML) predictive models, the need for worst-case guarantees under incomplete information may be too pessimistic. Beyond worst-case analysis, recent researchers have explored learning-augmented algorithms to leverage predictions to improve decision-making. With ML models providing increasingly accurate predictions of unknown variables, these algorithms are robust and have an improved performance.*
>
> **Clarify the "prediction errors" section**
>
> Thanks for pointing out the non-clarity of the "prediction errors" section. We have revised it to begin with the following clarifying sentence:
>
> *We consider several types of predictions: job size predictions provided at each job's arrival, an upfront prediction for the minimum job size, and an upfront prediction for the maximum job size.*
>
>
> **"Preferential round robin" at Line 87**
>
> Thanks for pointing out the missing reference for "preferential round robin". We added the following reference when we refer to it.
>
> *Manish Purohit, Zoya Svitkina, and Ravi Kumar. Improving online algorithms via ml predictions. In Advances in Neural Information Processing Systems, 2018.*
>
>
> **Findings from the experiments**
>
> Thank you for highlighting the need to include key findings from our experiments. We have revised the text in the contribution list to address this point as follows.
>
> ***Extensive experimental validation:*** *We conduct experiments on synthetic and real-world datasets to evaluate the performance of these algorithms. The experimental results validate our theoretical analysis and demonstrate that $\textsf{RG}^+$ consistently outperforms other algorithms in practical scenarios, showing its robustness and effectiveness.*
>
>
> **Line 94: Why is this assumption unrealistic? In practice?**
>
> For clarification, this assumption implies that the scheduler has precise knowledge of the minimum and maximum job sizes it will encounter during the upcoming operating period (e.g., a day). While it may be feasible to establish an upper bound on job size by imposing a hard constraint on the maximum allowable processing time in the system, this is different from knowing the exact minimum or maximum job sizes of an upcoming stream of jobs. In practice, determining such precise values is challenging.
>
>
> **"Necessity" of resource augmentation**
>
> We appreciate the reviewer’s comment on the use of the term "necessity" in the name of Theorem 6.1. We acknowledge that this wording may have been too strong and potentially misleading. For clarification, our intention was not to imply that speed augmentation is a strictly necessary condition (in the mathematical sense) for achieving robustness of $O(n)$.
>
> Instead, our point was to emphasize that existing well-established techniques, i.e., Preferential Round Robin (PRR), are insufficient for achieving sound robustness in our setting. Therefore, we propose speed augmentation as a solution. To avoid confusion, we will change the name of Theorem 6.1 from "Necessity of Resource Augmentation" to "Motivation for Resource Augmentation."

---

> ### Author Response · Authors · 2024-11-24
> **Responses to additional remarks**
>
> **Adding reference of Azar, Leonardi and Touitou (STOC 21)**
>
> Thanks for suggesting the reference to Azar, Leonardi and Touitou (STOC 21). Their work similarly defines the error as the multiplicative gap between the exact values and predictions, though in a slightly different form. We have now included this reference in the revision where the prediction error is defined (Line 60).
>
> **Smoothness and Robustness**
>
> Thanks for pointing out the need to clarify the distinction between smoothness and robustness. In our work, smoothness refers to the competitive ratio as a function of the prediction error $\eta$, while robustness is specifically the competitive ratio in the worst-case scenario when $\eta \rightarrow \infty$. In other words, robustness is defined as robustness = $\lim_{\eta \rightarrow \infty}{\text{smoothness}}$. To address this comment, we have added the following explanation at the end of the section introducing consistency, smoothness, and robustness to make this distinction clearer for readers:
>
> *The relationship between smoothness and robustness can be expressed as robustness = $\lim_{\eta \rightarrow \infty}{\text{smoothness}}$.*
>
> **Related work**
>
> Thanks for your valuable feedback on how to enhance the related work section. We have revised this section to ensure all variables are defined and that the reviewed works represent recent advancements in scheduling algorithms with predictions. The updated version now reads as follows:
>
> *We review recent advances in competitive online scheduling with predictions. For single-machine static scheduling to minimize total completion time, an $O(\min \\{ \frac{1}{\lambda} (1 + \frac{2\eta}{n}), \frac{2}{1 - \lambda} \\})$-competitive algorithm exists (Purohit et al., 2018), where $\eta$ is the additive error defined as $\eta = \sum_{j = 1}^{n}{|p_j - p^{\ast}_j|}$, and $ \lambda $ is a user-defined parameter. Follow-up works analyze robustness-consistency trade-offs (Wei and Zhang, 2020) and extend the results to dynamic scheduling and parallel machines (Bampis et al., 2022). For unrelated machine scheduling to minimize makespan, a deterministic $O(\min \\{ \frac{\log \eta \log m}{\log \log m}, \log m \\}) $-competitive algorithm is developed (Li and Xian, 2021), building on (Lattanzi et al., 2020). Here, the predictions are for machine loads, $ \eta $ is the multiplicative error of the predictions, and $ m $ is the number of machines. For single-machine scheduling to minimize weighted response time, an $O(\min \\{ \mu^3 \log (\mu P), \mu^3 \log (\mu D) \\})$-competitive algorithm is developed (Azar et al., 2021), where $ \mu $ is the multiplicative error for job size predictions and $D$ is the maximum density ratio. For parallel identical machine scheduling, an $O(\min \\{ \mu P, \mu \log \mu + \mu \log \frac{n}{m} \\})$-competitive algorithm is developed to minimize the mean response time, along with an $O(\mu^2)$-competitive non-migrative algorithm for minimizing mean stretch (Azar et al., 2022). Our problem is a special case of minimizing maximum weighted response time, but prior works under the learning-augmented framework focus on min-sum objective, while our work focuses on min-max. Results on prediction error metrics (Im et al., 2021), multiple predictions (Dinitz et al., 2022), machine speed predictions (Balkanski et al., 2023), and stochastic scheduling (Merlis et al., 2023) are also available.*
>
> **Including the offline result in the contributions section**
>
> Thanks for your suggestion to highlight the offline result in the contributions section. We have revised the first item in the contributions section to include this result. The updated version now reads:
>
> *We present an exact offline algorithm with a runtime of $O(n^2 \log n)$, where $n$ is the number of jobs.*
>
> **Line 207: clarification on the maximum**
>
> Thanks for your comment regarding the maximum. We are unsure if this refers to Line 207, as there is no reference to a "maximum" there. Perhaps the reviewer intended to reference Line 307 instead? If so, we are happy to clarify.
>
> In Line 307, $p_{\max, t}$ is defined as the maximum job size prediction observed by the algorithm up to and including time $t$, i.e., $p_{\max, t} = \max_{r_j \leq t}{p_j}$. This means that $p_{\max}$ is not static in ARG but instead varies over time as the algorithm receives new predictions.

---

> ### Author Response · Authors · 2024-11-24
> **Revisions based on the questions**
>
> **Optimality of $\textsf{RG}^+$**
>
> We have added Subsection J.1 in the Appendix to discuss the optimality of the robustness achieved by $\textsf{RG}^+$ under speed augmentation (see Appendix, P32).
>
> **Refining the proof of Theorem 6.4**
>
> We have refined the proof of Theorem 6.4 (Theorem J.6 in Appendix, P32) to highlight the use of monotonicity explicitly.

---

> ### Author Response · Authors · 2024-11-24
>
> We sincerely hope that our responses and revisions have addressed your concerns. We would be truly grateful if you could kindly consider revisiting your evaluation and, if appropriate, raising your score.

---

> > ### Comment · Reviewer_KufQ · 2024-11-25
> >
> > The authors addressed my comments and questions. I slightly raised my score.

---

> > > ### Author Response · Authors · 2024-11-26
> > > **Appreciate the reviewer's support**
> > >
> > > We greatly appreciate the reviewer's support and the decision to raise the score.

---

### Official Review · Reviewer_YoQS · 2024-11-01

**Soundness:** 3
**Presentation:** 3
**Contribution:** 3
**Rating:** 6
**Confidence:** 4

**Summary:**

The authors consider the minimum max-stretch scheduling problem under the learning augmented framework. Jobs $J_j$ $1\leq j \leq n$ arrive sequentially. Job $J_j$ arrives at time $r_j$, has compute time $p_j^*$ and is released at time $C_j$. The goal is to minimize the max stretch, $\max_{j \in [n]} \frac{C_j-r_j}{p^*_j}$.

The authors present several algorithms:
- An offline algorithm achieving optimal max-stretch in poly-time, improving upon previous literature where a poly-time approximation algorithm was known.
- A family of relaxed-greedy algorithms to schedule online under different types of available predictions.
- An algorithm that explores the smoothness consistency trade-off
- An algorithm for the resource augmented setting

**Strengths:**

The results presented in the paper are quite nice and turn in an array of contributions. They address a relevant problem in scheduling under a currently popular and pertinent lense, algorithm with predictions.

The paper is not very well written but remains relatively easy to parse.

**Weaknesses:**

The paper lacks insights into the results and techniques. It is currently a list of theorems, and it takes some time to figure out how they are related to each other and what the actual contribution is. Listed below is a list of elements on which the paper could improve.

On offline scheduling:
- The first theorem (Thm 3.1) and the first paragraph contradict each other. The Thm says the algorithm is poly time in $n$, the paragraph says it is not.
- Earliest deadline first is referenced without being defined (defined only in the appendix).
- The paragraph states that there are infinite candidates for the max-stretch. That is not a convincing argument: with $r_{max}$ the max release time, $U=\frac{r_{max}+\sum_j p_j}{\min_j p_j}$ is an upper bound on the max-stretch. Then there are $U/\epsilon$ candidates for $\epsilon$ approximations of the max stretch.
- The nice contribution of Thm 3.2 is Lemma A.1.: instead of doing a binary search over uniformly spread candidate max stretch, the authors find a list of candidates that contains the true max stretch. This entails that the proposed algorithm is optimal instead of approximately optimal. This is a nice contribution, but it takes reading the Appendix to figure it out.

On Relaxed-Greedy:
- Streamline the pseudo-code of relaxed greedy. In its current form, it does not help the reader. Maybe every subroutine does not need to be detailed. (e.g. JobRelease() takes 4 lines but is fairly useless)
- The main algorithmic idea is the reduction to two job-size, where the criteria for job length classification depends on $p_{min}$ and $p_{max}$. It would be helpful to get this insight earlier: it would help the reader understand the algorithm. It would also highlight why $p_{min}$ and $p_{max}$ play a special role.
- Thm 4.6 and 4.7: there is too many theorems in the paper, it is hard to tell what's important and what's not. Those two could easily become propositions.
- Definitions 4.8 and 4.9 are clutter: it is the same as the previous ones with $\alpha$ replaced with $\alpha_{r_j}$. If there is a difference I missed it should be highlighted, otherwise, I suggest dropping them. I am also not convinced by the use of vocabulary $r$-long.

On $RG^x$:
It should be said earlier in the text that FIFO is O(P) competitive and that $RG^x$ interpolates between FIFO and the proposed algorithms.



On notations:
- Maybe replace $p_j$ with $\hat{p}_j$ and $p^*_j$ with $p_j$

**Questions:**

Is the algorithm in Thm 3.1 poly time or not?

---

> ### Author Response · Authors · 2024-11-16
>
> We appreciate the reviewer's constructive comments, insightful questions, and positive feedback regarding our work. We are grateful for the opportunity to clarify the points raised. Below, we provide detailed responses to your questions. *We will address additional concerns/weaknesses regarding the presentation in separate threads with a revised manuscript.*
>
> ### **Q1 (and W1). Algorithm complexity in Thm 3.1**
>
> We appreciate the opportunity to clarify the complexity of the algorithm in Theorem 3.1. This question is also related to the first point raised in the weaknesses section of the review.
>
> For clarification, the algorithm in Theorem 3.1 is polynomial time in $n$ and $\frac{1}{\epsilon}$, as stated in the Theorem. We acknowledge that the wording in the first paragraph following Theorem 3.1 may have caused confusion, and we will revise it to make this clearer. Our intention was to convey that while the algorithm is polynomial in terms of the number of jobs $n$ and the precision parameter $\epsilon$, it is not purely polynomial in the input size $n$. This is because, traditionally, finding the exact minimum max-stretch was considered to involve an infinite search space, thus requiring a predefined precision $\epsilon$ to discretize the search space, and, therefore, an algorithm would need to account for this precision, which explains the dependence on $\frac{1}{\epsilon}$.
>
> Our work overcomes this infinite search space issue by showing that the exact minimum max-stretch can be found within a list of candidates of size polynomial in $n$, as you correctly noted. This eliminates the need for an infinite search and ensures that our algorithm is exact and is polynomial purely in $n$.
>
> We appreciate the reviewer's recognition of this contribution and will revise the relevant sections to prevent any ambiguity for readers.

---

> ### Author Response · Authors · 2024-11-24
> **Responses to weaknesses**
>
> ***Improving the presentation of the paper***
>
> We appreciate the reviewer's constructive comments and suggestions on improving the presentation of the paper. We have revised the paper to highlight the contributions and insights earlier. Below, we respond to the reviewer's comments and detail the relevant changes to the paper according to the reviewer's suggestions.
>
> **On offline scheduling**
>
> - We have revised the first item in the contributions section to include this result. The updated version now reads:
>
> *We present an exact offline algorithm with a runtime of $O(n^2 \log n)$, where $n$ is the number of jobs.*
>
> - We also clarified the first paragraph for the offline scheduling section. The revised text reads:
>
> *While the algorithm is polynomial in terms of the number of jobs $n$ and the precision parameter $\epsilon$, it is not purely polynomial in the input size $n$. This is because, traditionally, finding the exact minimum max-stretch was considered to involve an infinite search space, thus requiring a predefined precision $\epsilon$ to discretize the search space.*
>
> **On Relaxed-Greedy**
>
> - We have streamlined the pseudo-code of Relaxed-Greedy. The functions relaxedStretch and JobRelease now take fewer lines.
>
> - To improve clarity, we have highlighted at the very start of Section 4.1 (Algorithm Overview) that the key algorithmic idea is the reduction to the two-job-size case. The section begins as follows:
>
> *The key algorithmic idea is the reduction to the case with two job sizes.*
>
> - We have changed Theorems 4.6, 4.7, 4.12, and 4.13 to Propositions.
>
> - For clarification, Definitions 4.8 and 4.9 differ from the previous ones, though they appear similar. In Adaptive Relaxed-Greedy (ARG), the algorithm does not know the maximum job size upfront; it learns this value dynamically as it observes job size predictions. Consequently, the values of $\alpha$ and $\beta$ change over time. This makes the classification of jobs non-stationary. One of the key technical contributions of ARG is ensuring that a job's classification remains fixed based on the value of $\alpha_{r_j}$ at its arrival time. For example, jobs with job size prediction 5 are classified as *long* at arrival and may no longer be classified as *long* later as the algorithm may encounter larger job size predictions, say, 50. To emphasize that a job's classification depends on the arrival time and the past job size predictions and is not intrinsic to the job's characteristics, we use the terms *$r$-long* and *$r$-short* instead of the static terms *long* and *short*.
>
> **On $\textsf{RG}^x$**
>
> We have highlighted the summarized result for $\textsf{RG}^x$ in the first paragraph of Section 5. The revised text now reads:
>
> *We show that $\textsf{RG}^x$ interpolates between the $\textsf{First Come First Serve}$ (denoted by $\textsf{FIFO}$) algorithm and $\textsf{RG}$, where $\textsf{FIFO}$ is $O(P)$-competitive and $\textsf{RG}$ is $O(\eta^{3} \cdot \sqrt{P})$-competitive.*
>
> **On notations**
>
> The notations are designed to align with the existing literature on similar topics. For instance, see Zhao, Li, and Zomaya (RTSS 2022). Symbols with a star denote the exact values not directly observable by the algorithm, such as the actual job size or the optimal max-stretch ($p^*_j$ and $S^*$). In contrast, symbols without a star represent quantities the algorithm can directly observe, such as the release time $r_j$ and job size prediction $p_j$ (when the job arrives).

---

> ### Author Response · Authors · 2024-11-24
>
> We sincerely hope that our responses and revisions have addressed your concerns. We would be truly grateful if you could kindly consider revisiting your evaluation and, if appropriate, raising your score.

---

> > ### Comment · Reviewer_YoQS · 2024-11-25
> > **Commitment to improve clarity of the paper**
> >
> > Overall, I do not have major concerns; my questions have been addressed appropriately.
> >
> > There seems to be a consensus among reviewers that the paper had to be improved to include more insights into the techniques and proofs.
> >
> > The revised version has improved along that dimension, so I have slightly raised my score in response. However, I believe there is still room for improvement, and I encourage the authors to go through the paper again and try to clarify further.

---

> ### Author Response · Authors · 2024-11-26
> **Improving the clarity of the paper**
>
> We appreciate the reviewer’s suggestions for further clarifying the technical insights in our paper. In response, we have revised the manuscript again to enhance the presentation of key technical contributions in the following sections:
>
> **Offline scheduling**
>
> We have added the following text to Section 3.1 to emphasize the key insight of the offline algorithm:
>
> *The insight is identifying an $O(n^2)$-size list of candidates that contains the optimal max-stretch.*
>
> **Non-clairvoyant lower bound**
>
> We have revised Section 3.3 to summarize the construction of the adversary in the proof of the non-clairvoyant lower bound:
>
> *The key construction of the adversary is to force the algorithm to allocate no more than a $\frac{1}{n}$ share of processing at some point for the smallest-sized job, resulting in a stretch of $n$ for that job. Subsequently, the rest of the jobs are assigned geometrically increasing sizes, ensuring that the optimum max-stretch is approximately $1$.*
>
> **Innovation of heterogeneous predictions**
>
> To clarify the innovation in the use of heterogeneous predictions, we have added the following text to Section 4.3:
>
> *One innovation of $\textsf{ARG}$ is the use of heterogeneous predictions, i.e., predictions for different types of quantities, in learning-augmented algorithms. Our analysis reveals that these different predictions impact the algorithm’s performance differently.*
>
> We greatly appreciate the reviewer’s support and the decision to raise the score. We hope these additional revisions further improve the clarity and impact of the paper. Should there be any remaining areas for improvement, we are happy to address them promptly. We would also be grateful for further support from the reviewer in light of these clarifications.

---

### Official Review · Reviewer_xyBg · 2024-11-03

**Soundness:** 3
**Presentation:** 3
**Contribution:** 3
**Rating:** 8
**Confidence:** 3

**Summary:**

The authors consider the problem of preemptively scheduling jobs on a single machine to minimize the maximum stretch, where the stretch of a job is the response time of the job normalized by its processing time. The jobs arrive online and the processing times are unknown and are only revealed once a job is completed, i.e., jobs need to be processed non-clairvoyantly.

Since the problem admits strong adversarial lower bounds, the authors consider learning-augmented algorithms that initially have access to predictions on the minimum and maximum processing time over the jobs and, for each arriving job, receive a prediction of the jobs processing time upon arrival.

The authors give algorithms that achieve a consistency (competitive ratio for accurate predictions) that matches the best-known clairvoyant algorithm, and are smooth w.r.t. a prediction error. Furthermore, the authors show that well established standard techniques are not enough to achieve the best-possible robustness unless the algorithm has access to a faster machine (speed-augmentation).
Besides this, the authors also give a polynomial-time algorithm for the corresponding offline problem and empirical experiments for the presented algorithms.

**Strengths:**

# Strengths:

* The learning-augmented results given in the paper are quite complete and answer most of the questions that one could have (consistency, smoothness, discussion of the robustness).
* The algorithmic ideas are presented in a nice way by iteratively removing more assumptions from Section 4.2 to 4.4.
* I like that the authors also resolve the complexity of the offline problem with Theorem 3.2. The idea to reduce the problem to deadline scheduling (minimizing the maximum lateness) by finding the “correct” deadlines via binary search over the set of line intersections is very elegant.
* The fact that Preferential Round  Robin (Theorem 6.1) does not work is a nice separation from many existing results on learning-augmented algorithms for scheduling problems.
* The experiments seem well-done.

**Weaknesses:**

# Weaknesses:

* The main part of the paper does not really highlight the techniques that are used to prove the theoretical results, apart from the discussion on the overestimation of job sizes at the beginning of Section 4.1. Since the algorithmic ideas are similar to the Algorithm by Bender et al. (2002), it would have been nice to highlight the differences and challenges in the proof that are caused by using predictions instead of precise processing times. The way it is written now, it is hard to estimate the technical contributions.
* The appendix could use one more iteration of proofreading. Some examples:
  * The authors state on page 14 that EDF is optimal for minimizing the maximum lateness, but use it already on page 13.
  * The phrasing of Theorem A.1 does not seem to quite match the statement that you actually show. For example: If you plug-in the processing time vector $(p_1^*,...,p_n^*)$ in Theorem A.1, then the theorem says that there exists a schedule with minimum max-stretch $\max_{1 \le j \le n} \frac{C_j-r_j}{p_j}$. However, this holds directly by definition of the problem. For this reason, Lemma A.3 does not seem to be implied by the statement of Theorem A.1, but by the proof of the theorem, which indeed implies the lemma.  Otherwise, the approach of the authors to prove the offline result makes sense.
  * There are several repetitions already on the first two pages of the appendix.

* Apart from the first point, the presentation in the main part is mostly fine with the following small exceptions:
  * Lines 038-040: You state that job sizes are typically estimated using ML.  Do you mean in other theoretical learning-augmented results or in practice? If you mean the latter, it would be nice to have a reference for this.
  * Line 046: I think there are learning-augmented results on scheduling problems with additive errors. Maybe clarify that this refers to your error.
  * Lines 094-096: I agree that the assumption of knowing the precise processing times may be unrealistic. However, it is a bit strange to use the adversary as an argument since having an adversary might not be a very practical assumption as well.
  * Related work/offline scheduling: Would be nice to reference  some flow-time (response-time) results and briefly discuss the differences. (As your offline problem seems to be a special case of minimizing the maximum weighted response time, such a discussion seems relevant.)
  * Maybe mention a bit earlier that preemption is allowed.
 * Experiments: I am not sure if we learn something by comparing the speed-augmented RG+ to the other algorithms running at normal speed. It seems like an unfair comparison. Maybe it would be more interesting to also run RG+ on normal speed to empirically analyze the impact of Theorem 6.1 on real-world instances.

**Questions:**

# Questions:

1. More of a comment than a question: It is interesting to me that your algorithms nearly do not use the predicted processing times and essentially only use them for the classification whether a job is big or small. Maybe you can prove similar results (with different error measures) if the predictions are p_max, p_min and, for each job, a bit that predicts the job to be either small or big.
2. Theorem 6.1 shows that the blackbox combination of two algorithms will not benefit from the predictions. However, this does not exclude the possibility of a different technique to achieve an improved consistency and a robustness of O(n)? If it does not exclude that possibility, then calling Theorem 6.1 the “Necessity of Resource Augmentation” might be misleading.
3. Experiments: I assume that GWR gets access to the precise processing times? Any idea on why the competitive ratio does not increase with the prediction error in Figure 6?
4. Just to make sure I understand since I was a bit confused reading it: Section 4.3 still assumes that $p_\min^e$ is given up-front but, in contrast to Section 4.2, it might be wrong even with respect to the predicted job sizes, i.e., $p_\min^e \not= \min_j p_j$ might hold. Is this right?

---

> ### Author Response · Authors · 2024-11-16
>
> We appreciate the reviewer's constructive comments, insightful questions, and positive feedback regarding our work. We are grateful for the opportunity to clarify the points raised. Below, we provide detailed responses to your questions. *We will address additional concerns regarding technical contributions and presentation in separate threads with a revised manuscript.*
>
> ### **Q1. Regression vs. classification**
>
> We appreciate the reviewer’s observation regarding the use of predictions in our algorithm, which may appear to rely primarily on classifying jobs as "big" or "small." However, it is important to note that this classification is non-stationary, meaning that whether a job is classified as big or small is not intrinsic but instead relative to the parameter $\alpha$.
>
> In particular, the Adaptive Relaxed-Greedy (ARG) algorithm (Line 310) continuously updates $\alpha$ based on observed jobs, making it a function of time. As a result, a job classified as big upon arrival might be classified as small if it arrives at a different point in time or if a different set of jobs arrives beforehand. This dynamic classification depends heavily on the sequence of job arrivals, making it challenging to model this as a straightforward classification problem.
>
> Even if a classification model were to be designed, it would require access to the current value of $\alpha$ as part of its input. Additionally, such a classification model would likely need to internally predict job sizes (i.e., perform a regression) before applying the classification layer. Thus, while it may appear that our algorithm relies on classification, it fundamentally leverages the numerical predictions about the job sizes to make decisions.
>
> ### **Q2. Clarification on the use of the term "necessity"**
>
> We appreciate the reviewer's comment on the use of the term "necessity" in the name of Theorem 6.1. We acknowledge that this wording may have been too strong and potentially misleading. For clarification, our intention was not to imply that speed augmentation is a strictly necessary condition (in the mathematical sense) for achieving robustness of $O(n)$.
>
> Instead, our point was to emphasize that existing well-established techniques, i.e., Preferential Round Robin (PRR), are insufficient for achieving sound robustness in our setting. Therefore, we propose speed augmentation as a solution. To avoid confusion, we will change the name of Theorem 6.1 from "Necessity of Resource Augmentation" to "Motivation for Resource Augmentation."
>
> ### **Q3. Experimental results regarding GWR**
>
> Yes, you are correct that GWR has access to the exact processing times, as it is a clairvoyant algorithm. This explains why its competitive ratio does not change with increasing prediction errors in Figure 6. Since GWR always operates with the actual job sizes, it is unaffected by changes in predictions. Similarly, the Round Robin (RR) algorithm does not rely on job size information, so its performance also remains unchanged regardless of the prediction error.
>
> ### **Q4. Clarification on $p^e_{\min}$ in Section 4.3**
>
> The reviewer's understanding is correct. In Section 4.3, we assume that a prediction for the minimum job size, denoted as $p^e_{\min}$, is provided up-front. This prediction may not be correct, so $p_{\min}^e \neq \min_j p_j^*$ and may not even be consistent with the following job size predictions, so $p_{\min}^e \neq \min_j p_j$ might also hold.
>
> For clarification, the prediction $p^e_{\min}$ might (very likely) be obtained from a separate model, potentially one trained on historical data, to estimate the smallest job size expected to arrive over a certain period (e.g., daily). This model may operate independently from the real-time job size prediction model that uses the job's characteristics to predict its job size. As a result, the prediction may not be perfect, and inconsistencies can occur between the predicted minimum job size and the (real-time) predictions for job size.
>
> Despite this potential inconsistency, our analysis remains valid. The prediction errors for different types of quantities (minimum job size to appear vs individual job sizes) impact the performance guarantee independently (Line 300). We will clarify this distinction in the revised manuscript to prevent any confusion for readers.

---

> ### Author Response · Authors · 2024-11-24
> **Responses to weaknesses**
>
> We appreciate the reviewer's constructive comments and suggestions on improving the presentation of the paper. We have revised the paper to highlight the key technical contributions of Section 4. Below, we respond to the reviewer's comments and detail the relevant changes to the paper according to the reviewer's suggestions.
>
> **Technical contributions of Section 4**
>
> We highlight the technical contributions of Section 4 as follows.
>
> **1.** Relaxing assumptions on job sizes
>
> Previous algorithms assume that the minimum job size is known to be 1 and that the maximum job size is also known upfront. Relaxing these assumptions using predictions is non-trivial. Our approach includes:
>
> - Dynamically adjusting the predicted maximum job size $p_{\max}$ as new jobs are observed, as implemented in the Adpative Relaxed-Greedy (ARG) algorithm (original submission Line 304).
>
> - Using upfront predictions for minimum and maximum job sizes, as employed in the Predictive Relaxed-Greedy (PRG) algorithm (Line 347).
>
> In both approaches, we set $\alpha$ as the geometric mean of the predicted minimum and maximum job sizes, which serves as the boundary for classifying jobs as small or large. Our analysis shows that such methods allow the performance to tie to the prediction errors for these quantities (original submission Line 352).
>
> **2.** Dynamically classifying jobs with changing $\alpha$
>
> A key technical challenge in ARG is classifying jobs when $\alpha$ is continuously updated over time ($\alpha_t$ , original submission Line 309). Should all jobs be reclassified upon each update? It turned out that dynamically changing the classification of jobs might work, but the proof for its competitive ratio is difficult to establish (we did not find proof). Our solution overcomes this issue by ensuring that once a job is classified at its arrival time, based on $\alpha_{r_j}$, it remains fixed in that classification. For example, if job $J_j$ is classified as $r$-long, it remains a long job, regardless of future updates of $\alpha$. This enables the competitive analysis for ARG (Lemma E.6 to Theorem E.11).
>
> We acknowledge that this subtle yet significant technical challenge may not be immediately apparent in the main text. We have made the following revisions to clarify these technical contributions in Section 4.
>
> We have added the following paragraph at the very beginning of Section 4.2 Relaxed Greedy.
>
> *One of the assumptions for previous algorithms is that the minimum job size is known to be 1 and that the maximum job size is also known upfront. We relax these assumptions by setting a threshold $\alpha$ as the geometric mean of the predicted minimum and maximum job sizes, which serves as the boundary for classifying jobs as small or large. Our analysis shows that such methods allow the performance to tie to the prediction errors for these quantities.*
>
> We also have added the following paragraph at the start of Section 4.3 Adaptive Relaxed Greedy.
>
> *A key technical challenge in ARG is classifying jobs when $\alpha$ is continuously updated over time. Dynamically changing the classification of jobs might work, but the proof for its competitive ratio is difficult to establish (we did not find proof). Our solution overcomes this issue by ensuring that once a job is classified at its arrival time, based on $\alpha_{r_j}$, it remains unchanged in that classification. For example, if job $J_j$ is classified as $r$-long, it remains a long job, regardless of future updates of $\alpha$. This enables the competitive analysis for ARG.*
>
>
> **Offline scheduling in the Appendix**
>
> Thanks for pointing out a few improvements in the Appendix for offline scheduling. We have revised the Appendix accordingly. Below details our responses to the comments and the revisions.
>
> - We have moved the following text to the start of Appendix A.
>
> *In the analysis of this section, we use the optimality of $\textsf{EDF}$: $\textsf{EDF}$ minimizes the maximum lateness of the jobs and always finds a feasible schedule if one exists.*
>
> - Thanks for going through the details of the proof for Theorem A.1 and Lemma A.3. We feel that, though the max-stretch minimization problem asks to minimize the max-stretch, it does not obviously (to us) that the problem implies that a global max-stretch always exists unless proved by Theorem A.1, which fills a (small) gap in the theory for max-stretch minimization.
>
> - Thanks for pointing out repetitions in the construction of set $\mathbf{S}$ in the Appendix A. We have revised the text to delete the repetitions.

---

> ### Author Response · Authors · 2024-11-24
> **Responses to weaknesses (cont.)**
>
> **Responses to the comments related to the presentation**
>
> - We mean that job sizes are typically estimated using ML in practice. See a comprehensive survey on this field:
>
> *Amiri and Mohammad-Khanli, "Survey on prediction models of applications for resources provisioning in cloud," J. Netw. Comput. Appl., 2017.*
>
> We have included this reference in the revision (Line 49).
>
> - We have clarified in the related works that some existing works on learning-augmented scheduling use additive error, and some use multiplicative error (See the Related work section in the revision).
>
> - Thanks for pointing out the argument about the adversary. For clarification, the adversary argument was to support that knowing in advance the exact minimum and maximum job sizes for the following stream of jobs is hard (not to support knowing the exact job sizes per arrival). Here, the adversary is not in the sense that an attacker intentionally alters extreme job sizes to send to the system, but that it is natural that there will be varying levels of extreme jobs (small or large) from now and then that "unintentionally" makes fair scheduling hard. We have revised the text as follows to clarify this point.
>
> *The assumption is unrealistic, particularly when the extreme job sizes vary from time to time.*
>
> **Related work**
>
> Thanks for suggesting references for some flow-time (response-time) results. We have revised the related work section to include the recent results on learning-augmented algorithms for response time scheduling and mean stretch. We also discussed its relevance to our work as follows.
>
> *Our problem is a special case of minimizing maximum weighted response time, but prior works under the learning-augmented framework focus on min-sum objective, while our work focuses on min-max.*
>
> **Preemption**
>
> Thanks for suggesting clarification of preemption earlier. We have mentioned the preemption in the first sentence of the second paragraph in the Introduction in the revision, which reads as follows.
>
> *Scheduling jobs that arrive over time on a single machine (preemption allowed) is a fundamental challenge in computing resource management, particularly when aiming to minimize the maximum stretch.*
>
> **Speed of $\textsf{RG}^+$ in the experiments**
>
> Thanks for the question regarding the speed of $\textsf{RG}^+$ in the experiments. In theory, we have shown the competitive ratio under speed augmentation. It is thus expected to verify how the speed translates to the performance in the experiments. We can learn how much performance we can gain with a slightly faster machine, which justifies the development of faster processors. Having a perfectly fair comparison setting for these algorithms is hard due to their slightly different execution requirements. For example, it is also unfair to the RG family when compared to GWR, which has access to the exact job size and the exact maximum job size in the job set upfront — so it can make (accurate) job classification. However, we are still interested in comparing against GWR because we want to see how much performance we lose when having imperfect information.
>
> We acknowledge that running $\textsf{RG}^+$ with the same speed as the others has the merit of showing the algorithm's robustness even without speed augmentation. We therefore expanded the experiments to include this result, which we added at the end of the Appendix (See Appendix O on P37-38). The observation is as follows.
>
> The performance of $\textsf{RGF}$ ($\textsf{RG}^+$ running with a unit-speed machine) is not as good as $\textsf{RG}^+$ due to the slower machine, but the performance generally stays bounded in between $\textsf{RG}$ and $\textsf{RR}$ for all metrics. This is as expected. The intuition is that when $\textsf{RG}$ performs well on an instance, some portion of the processing is shared to $\textsf{RR}$, thus decreasing the overall performance; when $\textsf{RR}$ performs well, again, some portion is shared to $\textsf{RG}$, thus decreasing the overall performance. In return, however, the robustness is persevered, making the performance relatively stable under varying prediction errors.

---

> ### Author Response · Authors · 2024-11-24
>
> We sincerely hope that our responses and revisions have addressed your concerns. We would be truly grateful if you could kindly consider revisiting your evaluation and, if appropriate, raising your score.

---

> > ### Comment · Reviewer_xyBg · 2024-11-26
> >
> > Thank you to the authors for the well-crafted response. All my questions have been answered and most of my concerns have been addressed. In particular, the authors have added two very helpful paragraphs to section 4 to highlight the technical contributions of their approach.
> >
> > I have increased my presentation score and my contribution score from fair to good. I will also increase my overall score from 6 to 8 (the 7 score does not seem to be available). I am a little bit conflicted about the increase, but my assessment of the paper is closer to an 8 than a 6, so I will go with the higher score.

---

> > > ### Author Response · Authors · 2024-11-26
> > > **Appreciate the reviewer's support**
> > >
> > > We greatly appreciate the decision to raise the presentation and contribution scores and the overall score to 8. Your recognition of the paper means a great deal to us. Thank you for your kind acknowledgment and for considering the paper closer to an 8. We deeply value your assessment and support.

---

### Official Review · Reviewer_Lc8Q · 2024-11-04

**Soundness:** 3
**Presentation:** 3
**Contribution:** 2
**Rating:** 6
**Confidence:** 3

**Summary:**

This paper studies an online scheduling problem where the objective is the notion of max-stretch, which is designed to balance fairness over jobs. Here, the stretch of a job is defined as the ratio of its response time and the job size, and the "max-stretch" is the maximum stretch of jobs. There exist some previous works on both offline and online versions of the scheduling problem, but they assume exact knowledge of job sizes. This paper considers the setting where job sizes are not precisely given while some predictions about the values are available. The paper studies several types of prediction models, for which nearly tight algorithms are developed.

**Strengths:**

- The paper provides a comprehensive study of the max-stretch scheduling, not only considering the problem with predictions but also the offline and online problem with exact job sizes (Section 3).
- This paper considers several settings of different prediction models, concerning job sizes, the maximum job size, and the minimum job size. For all these settings, efficient algorithms are developed with a rigorous theoretical analysis.
- Trade-offs between consistency and smoothness are also analyzed, and how to improve robustness via resource augmentation is addressed.
- Numerical results with real-world datasets are given.

**Weaknesses:**

While the paper gives a comprehensive analysis of the online scheduling problem with predictions, the concept of scheduling with predictions is not entirely new. As very briefly summarized in the related work section, various works exist for online scheduling with predictions. One may argue that this paper extends the literature to the max-stretch scheduling.

**Questions:**

What is the novel component of this work compared to the previous works on online scheduling with predictions?

---

> ### Author Response · Authors · 2024-11-16
>
> We appreciate the reviewer’s constructive comments and positive remarks on the strengths of our paper. We understand that the reviewer's primary concern lies in identifying the novel contributions of our work in online scheduling with predictions. We want to address this concern by highlighting the novel components of our paper that distinguish it from prior work:
>
> **1. Introducing fairness objectives in learning-augmented scheduling**
>
> Prior research on scheduling with predictions has primarily focused on optimizing efficiency-related objectives such as total completion time and total flow time; our work is the first to extend this paradigm to fairness. We address the max-stretch metric to ensure equitable resource allocation among jobs, demonstrating that learning-augmented schedulers can be leveraged to enhance fairness, not just efficiency. This shift in focus underscores the broader applicability of learning-augmented algorithms in optimizing diverse objectives beyond traditional efficiency metrics.
>
> **2. Incorporating heterogeneous predictions for enhanced guarantees**
>
> Unlike previous studies that rely on homogeneous predictions (i.e., predictions for a single type of parameter), our work introduces the novel concept of using heterogeneous predictions to improve scheduling performance. Specifically, our Predictive Relaxed-Greedy (PRG) algorithm incorporates predictions for job sizes and predictions for the minimum and maximum job sizes to arrive in the future. Our analysis reveals that these different types of predictions have different impacts on the algorithm’s performance, leading to a competitive ratio of $O(\lambda^{0.5} \cdot \varphi^{0.5} \cdot \eta \cdot \max \\{ \eta, \varphi \\} \cdot \sqrt{P})$, where $\eta$ is the prediction error for job sizes, $\lambda$ for the minimum job size prediction, and $\varphi$ for the maximum job size prediction. This approach is novel in leveraging diverse predictions for improved scheduling performance.
>
>
> **3. Consistency-smoothness trade-off in scheduling**
>
> Most existing work on learning-augmented scheduling has concentrated on the trade-off between consistency (performance under perfect predictions) and robustness (performance under worst-case predictions). In contrast, our work is the first to introduce and analyze the consistency-smoothness trade-off. With the algorithm $\textsf{RG}^x$, we demonstrate that sacrificing some level of consistency can yield gains in smoothness, resulting in a competitive ratio with a better functional dependence on the prediction error. This contribution expands the theoretical landscape of learning-augmented algorithms, offering a new perspective on balancing prediction quality with algorithmic performance.
>
> **4. Bounding robustness via resource augmentation**
>
> In addressing the challenge of robustness, we identify that the widely-used Preferential Round Robin (PRR) technique, effective in many scheduling problems, does not extend to our problem of minimizing max-stretch. We overcome this limitation by introducing a new method based on speed augmentation, which achieves optimal robustness. Our RR-augmented $\textsf{RG}^+$ algorithm achieves a competitive ratio of $\min \\{ \sqrt{3} \cdot \eta^3 \cdot \sqrt{P}, \frac{n}{\epsilon} \\}$ with a $(1 + \epsilon)$-speed augmentation. This framework optimizes robustness for our specific setting and applies to other scheduling problems, opening new avenues for research.

---

> ### Author Response · Authors · 2024-11-24
>
> We sincerely hope that our responses and revisions have addressed your concerns. We would be truly grateful if you could kindly consider revisiting your evaluation and, if appropriate, raising your score.

---

### Official Review · Reviewer_k4zw · 2024-11-08

**Soundness:** 4
**Presentation:** 4
**Contribution:** 4
**Rating:** 8
**Confidence:** 4

**Summary:**

This paper uses the consistency-robustness framework to study the problem of learning-augmented job scheduling in multiple settings. The results are provided under multiple settings as functions of different notions of prediction errors, trade-offs between consistency-robustness, and resource augmentation settings. The paper then evaluates the performance of the proposed algorithms using both synthetic and multiple real data traces.

**Strengths:**

-- This is a very well-written paper with a comprehensive set of results both on the theoretical side and experiments.

-- The theoretical results sound solid and consider multiple settings that are common to consider in the analysis of such problems.

-- The experiments are great additions to the paper, especially since the author reports results for both synthetic and real traces.

**Weaknesses:**

-- Overall, this paper presents a good set of results. However, the authors do not provide insights on the additional challenges with respect to closely relevant prior work and how their techniques and algorithms differ from existing results. The current version of the introduction only states the problem, introduces the notions and preliminaries, and states the technical contributions. There is a big missing part on the challenges to achieve such results and, in doing so, what type of techniques are required.

-- Generally, the paper does not provide predictions based on lower bounds and notes on the Pareto-optimality of the provided consistency-robustness tradeoff.

**Questions:**

The answer to both comments on weaknesses will be great.

---

> ### Author Response · Authors · 2024-11-16
>
> We sincerely thank the reviewer for the positive feedback, insightful comments, and strong support for our work. We are grateful for the opportunity to address the points you raised. Below, we provide detailed responses to your questions and concerns. *We will also incorporate these insights into a revised manuscript to improve its clarity.*
>
> ### **Q1. How our techniques differ from existing results**
>
> We appreciate the reviewer's interest in understanding the additional challenges we addressed compared to prior work and how our techniques differ. Below, we highlight the specific challenges and our solutions:
>
> **1. Relaxing assumptions on job sizes**
>
> Previous algorithms assume that the minimum job size is known to be 1 and that the maximum job size is also known upfront. Relaxing these assumptions using predictions is non-trivial. Our approach includes:
>
> - Dynamically adjusting the predicted maximum job size $p_{\max}$ as new jobs are observed, as implemented in the Adpative Relaxed-Greedy (ARG) algorithm (Line 304).
>
> - Using upfront predictions for minimum and maximum job sizes, as employed in the Predictive Relaxed-Greedy (PRG) algorithm (Line 347).
>
> In both approaches, we set $\alpha$ as the geometric mean of the predicted minimum and maximum job sizes, which serves as the boundary for classifying jobs as small or large. Our analysis shows that such methods allow the performance to tie to the prediction errors for these quantities (Line 352).
>
> **2. Dynamically classifying jobs with changing $\alpha$**
>
> A key technical challenge in ARG is classifying jobs when $\alpha$ is continuously updated over time ($\alpha_t$ , Line 309). Should all jobs be reclassified upon each update? It turned out that dynamically changing the classification of jobs might work, but the proof for its competitive ratio is difficult to establish (we did not find proof). Our solution overcomes this issue by ensuring that once a job is classified at its arrival time, based on $\alpha_{r_j}$, it remains fixed in that classification. For example, if job $J_j$ is classified as $r$-long, it remains a long job, regardless of future updates of $\alpha$. This enables the competitive analysis for ARG (Lemma E.6 to Theorem E.11).
>
> We acknowledge that this subtle yet significant technical challenge may not be immediately apparent in the main text, and we will add more detailed discussions in the Appendix to highlight it.
>
>
> **3. Consistency-smoothness trade-off**
>
> Exploring how consistency and smoothness can be traded off in our scheduling problem is non-trivial. We discovered that parameterizing $\alpha$ as a weighted geometric mean of the predicted minimum and maximum job sizes allows us to interpolate between the theoretical guarantees of FIFO and RG algorithms. This leads to our design of $\textsf{RG}^x$ (Line 370).
>
> **4. Addressing the robustness issue**
>
> The RG family of algorithms suffers from poor robustness when predictions are arbitrarily bad. The challenge is that traditional approaches like Preferential Round Robin (PRR) do not bound robustness for max-stretch. We proved (Theorem 6.1) that combining algorithms, as long as each uses a slower-than-unit-speed machine, fails to achieve bounded robustness while maintaining consistency.
>
> Our solution was to introduce resource augmentation, which achieves bounded robustness while preserving consistency. We showed that this approach also leads to an asymptotically optimal robustness (see Theorems 6.4 and 3.5).
>
>
> ### **Q2. Pareto-optimality of the provided trade-off**
>
> We appreciate the reviewer's interest in the Pareto-optimality of the consistency-smoothness trade-off in Section 5. We acknowledge that while we have not formally proven Pareto-optimality in this work, we recognize it as an important direction for future research.
>
> However, we established that our method achieved the best possible trade-off within our current framework (see Appendix H). This result provides a benchmark for future researchers to explore alternative approaches if they aim to surpass what $\textsf{RG}^x$ achieves. While it may not be as strong as proving full Pareto-optimality, we believe this contribution is still valuable as it establishes the current limits of our method and sets the stage for further advancements.

---

> ### Author Response · Authors · 2024-11-24
> **Responses to weaknesses**
>
> **Clarifications of the technical challenges**
>
> Thanks for your suggestion to clarify the technical challenges we address in our work. We have revised the paper accordingly to include the following explanations:
>
> We have added the following paragraph at the very beginning of Section 4.2 (Relaxed Greedy).
>
> *One of the assumptions for previous algorithms is that the minimum job size is known to be 1 and that the maximum job size is also known upfront. We relax these assumptions by setting a threshold $\alpha$ as the geometric mean of the predicted minimum and maximum job sizes, which serves as the boundary for classifying jobs as small or large. Our analysis shows that such methods allow the performance to tie to the prediction errors for these quantities.*
>
> We also have added the following paragraph at the start of Section 4.3 (Adaptive Relaxed Greedy).
>
> *A key technical challenge in ARG is classifying jobs when $\alpha$ is continuously updated over time. Dynamically changing the classification of jobs might work, but the proof for its competitive ratio is difficult to establish (we did not find proof). Our solution overcomes this issue by ensuring that once a job is classified at its arrival time, based on $\alpha_{r_j}$, it remains unchanged in that classification. For example, if job $J_j$ is classified as $r$-long, it remains a long job, regardless of future updates of $\alpha$. This enables the competitive analysis for ARG.*
>
> **Pareto-optimality as future work**
>
> Additionally, we have included a note in the Conclusion section highlighting Pareto-optimality for the consistency-smoothness trade-off as an avenue for future research. The added text reads:
>
> *While we have shown that our construction of $\textsf{RG}^x$ achieves the best possible trade-off for consistency and smoothness under this method, we have not demonstrated the Pareto-optimality of the trade-off. Establishing this remains an interesting direction for future work.*

---

> > ### Comment · Reviewer_k4zw · 2024-11-26
> >
> > Thanks for the response, and I appreciate the author's effort.

---

> > > ### Author Response · Authors · 2024-11-26
> > > **Appreciate the reviewer's support**
> > >
> > > We sincerely appreciate the reviewer's support in the original assessment of our paper. Your recognition of our work means a lot to us and has been highly encouraging throughout the response period. Thank you for your kind comments and valuable feedback. We deeply value your evaluation and support.

---

### Author Response · Authors · 2024-11-24
**To All Reviewers**

We have addressed all the questions and weaknesses raised in the reviews and have revised the paper accordingly.

We look forward to your feedback and are happy to make any further improvements before the deadline.

Thank you for your time and consideration!

---

### Meta-Review · Area_Chair_2FZj · 2024-12-22

**Metareview:**

This paper considers learning-augmented scheduling problems, which are both really important in practice and quite interesting in theory.

All reviewers and I think this paper is interesting and brings valuable contribution to the field, hence we happily recommend acceptance.

**Additional Comments On Reviewer Discussion:**

Reviewers and I all enjoyed that paper. No discussion needed

---

### Decision · Program_Chairs · 2025-01-22

Accept (Poster)